# Conditional Generative Models are Sufficient to Sample from Any Causal Effect Estimand

**Md Musfiqur Rahman** *
Purdue University

**Matt Jordan** *
University of Texas at Austin

**Murat Kocaoglu**
Purdue University

## Abstract

Causal inference from observational data plays critical role in many applications in trustworthy machine learning. While sound and complete algorithms exist to compute causal effects, many of them assume access to conditional likelihoods, which is difficult to estimate for high-dimensional (particularly image) data. Researchers have alleviated this issue by simulating causal relations with neural models. However, when we have high-dimensional variables in the causal graph along with some unobserved confounders, no existing work can effectively sample from the un/conditional interventional distributions. In this work, we show how to sample from any identifiable interventional distribution given an arbitrary causal graph through a sequence of push-forward computations of conditional generative models, such as diffusion models. Our proposed algorithm follows the recursive steps of the existing likelihood-based identification algorithms to train a set of feed-forward models, and connect them in a specific way to sample from the desired distribution. We conduct experiments on a Colored MNIST dataset having both the treatment ($X$) and the target variables ($Y$) as images and sample from $P(y|\mathrm{do}(x))$. Our algorithm also enables us to conduct a causal analysis to evaluate spurious correlations among input features of generative models pre-trained on the CelebA dataset. Finally, we generate high-dimensional interventional samples from the MIMIC-CXR dataset involving text and image variables.

## 1 Introduction

Causal inference has recently attracted significant attention in machine learning (ML) due to its application in fairness, invariant prediction, and explainability [60, 62, 49]. Even though existing ML models show notable predictive performance by optimizing the likelihood of the training data, they are prone to failure when the covariate distribution changes in the test domain. Consider the medical scenario in Fig. 1a with the causal order: Xray($X$)→Diagnosis($S$)→Report($R$) representing the true data-generating mechanisms. Suppose a practitioner observes only $X$ to make a high-level intermediate diagnosis $S$ that contains sufficient information about the patient. The prescription report($R$) is written only based on the diagnosis (thus $X \not\to R$). Since data are collected from different hospitals locations ($H$), $H$ acts as an unobserved common cause for both $X$ and $R$, i.e., $X \leftrightarrow R$ (ex: correlation between x-ray artifacts and report writing style). The task is "x-ray to report" generation. One might train an ML model to directly learn a mapping $f : \mathcal{X} \to \mathcal{R}$, with maximum likelihood estimation (MLE) [7, 10] mimicking the conditional distribution $P(r|x)$. However, since $H$ is an unobserved common cause between $X$ and $R$; $H$ has some influence on $P(r|x)$ [9]. Thus, if the model is deployed in a new location, its MLE-based prediction accuracy may drop since $P(r|x)$ shifts in that location. On the other hand, if we can remove the location bias $X \leftrightarrow R$ with an intervention on the x-ray variable (do($x$)), the x-ray to report generation would be invariant to domain shifts. Thus, to obtain such generalization, we need to perform causal interventions in high-dimension.

---

* Equal contribution. Correspondence to rahman89@purdue.edu

38th Conference on Neural Information Processing Systems (NeurIPS 2024).

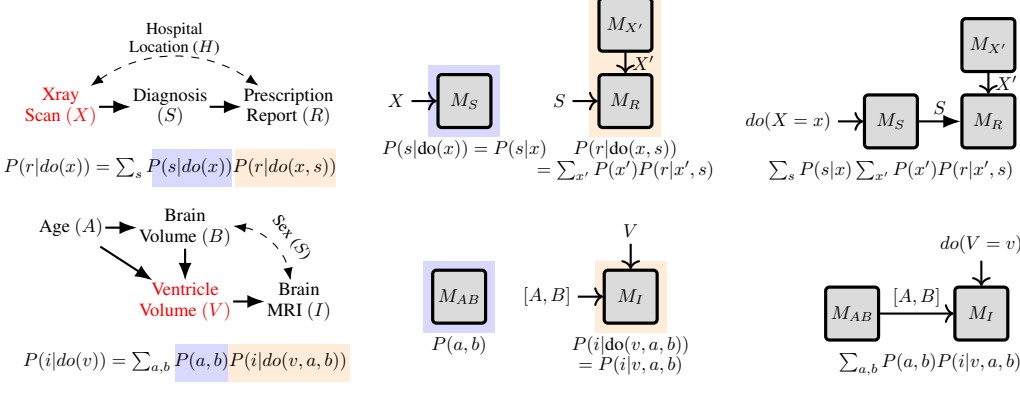

(a) ML model failure scenario  (b) Train conditional models.  (c) Merge and sample ancestrally.

Figure 1: (Top: x-ray to report generation task) (a) $do(X = x)$ removes the $X \leftrightarrow R$ bias and makes the generation of $R$ domain invariant. $P(r|do(x))$ is factorized into c-factors and (b) conditional models ($\{M_{V_k}\}_k$) are trained for each factor (shown as boxes). (c) The intervened value $X = x$ is propagated through the merged network and samples from the $P(r|do(x))$ are generated.

Structural causal models (SCM) [38] enable a data-driven approach to estimate interventional distributions [43, 30]. Given the qualitative causal relations, summarized in a *causal graph*, we now have a complete understanding of which causal effects/queries (ex: $P(r|do(x))$) can be uniquely identified from the observational distribution and which require further assumptions or experimental data [2, 23, 28, 37, 46, 50]. More precisely, if all conditional probability tables are available, sound and complete identification algorithms [51, 46] can perform exact inference to estimate causal effects or [4, 5, 25] can sample from the interventional distribution, using a combination of marginalization and product operators applied to those conditional distributions. However, such approaches struggle to deal with high-dimensional variables. In Fig. 1a, we could intervene on x-ray $X$ and estimate its effect on the report $R$ as, $P(r|do(x)) = \sum_s P(s|x) \sum_{x'} P(x')P(r|x', s)$, i.e., as functions of the observational distribution ($X, X'$: independent instances of the same variable). However, the second and third terms in this expression require marginalization over the "*X-ray*" variable. Exact Bayesian inference methods used for calculating conditional distributions are infeasible for high-dimensional variables since marginalization over their non-parametric distributions is generally intractable [6].

Deep generative models with variational inference methods approximate the intractable marginalization and can sample from such high-dimensional distributions [22, 47, 14]. Recent works such as Xia et al. [59], Chao et al. [11], Rahman and Kocaoglu [40] employ deep generative models to match joint distribution of the system by learning the conditional generation of each variable from its causal parents. Nonetheless, it is highly non-trivial for these works to mimic any arbitrary causal model with high-dimensional variables, specially when there are unobserved confounders in the (semi-Markovian) causal model. Consider the $X \leftrightarrow R$ relation in Fig. 1a where $R$ and $X$ are correlated through unobserved hospital location. To learn the joint distribution $P(x, r)$, the above approaches need to synchronously train their generative models. For that purpose, Xia et al. [59], Rahman and Kocaoglu [40] train two GAN networks concurrently by feeding the same prior noise. However, it is nontrivial to design a loss function for the joint distribution balancing multiple high-dimensional variables making it challenging for the discriminator to detect true/false sampled pairs. Thus, the high-dimensional intervention problem still requires a more effective approach.

In this paper, we propose a novel algorithm ID-GEN that can *utilize any (conditional) generative models (such as GANs or diffusion models) to perform high-dimensional interventional sampling in the presence of latent confounders.* For this purpose, we resort to the sound and complete identification algorithm [50, 46] and design our algorithm on top of its structure to sample from any identifiable causal query which may have an arbitrarily complex probabilistic expression (ex: Eq. 1). More precisely, given a causal graph, training data, and a causal query, our algorithm *i)* follows the recursive trace of the ID algorithm to factorize the query *ii)* trains a set of conditional models for each factor, *iii)* connects them to build a neural network called sampling network and generate interventional samples from this network. For example, to sample from $P(r|do(x))$ for the frontdoor graph in Fig. 1 (top), we *i)* utilize ID to obtain the factors: $P(s|do(x))$ and $P(r|do(x, s))$, *ii)* train conditional models $\{M_S\}, \{M_{X'}, M_R\}$ for the two factors (Fig. 1b), *iii)* merge all models based on input-output (Fig.1c). Sampling according to this network's topological order would produce samples

from $P(r|\text{do}(x))$. Similarly, for the backdoor graph (bottom), we train conditional models $\{M_{A,B}\}$ and $\{M_I\}$ to learn $P(a,b)$ and $P(i|v,a,b)$ respectively and merge them to sample from $P(i|do(v))$. To the best of our knowledge, we are the first to show that conditional generative/feedforward models are sufficient to sample from any identifiable causal effect estimand. Our contributions are as follows:

- We propose a recursive algorithm called ID-GEN that trains a set of conditional generative models on observational data to sample from high-dimensional interventional distributions. We are the first to use diffusion models as conditional models for semi-Markovian SCMs.

- We show that ID-GEN is sound and complete, establishing that conditional generative models are sufficient to sample from any identifiable interventional and conditional interventional query. The latter type are especially challenging for existing GAN-based causal models.

- We demonstrate ID-GEN's performance on three datasets containing image and text variables. First, we perform image intervention with diffusion models for the Colored-MNIST experiment. Next, we show our application in trustworthy AI through quantifying spurious correlations in pre-trained models for the CelebA dataset. Finally, we make the report to X-ray generation task interpretable and domain invariant based on the MIMIC-CXR dataset.

## 2 Background

**Structural causal model**, (SCM) [36] is a tuple $\mathcal{M} = (G = (\mathbf{V}, \mathcal{E}), \mathcal{N}, \mathcal{U}, \mathcal{F}, P(.))$. $\mathbf{V} = \{V_1, ..., V_n\}$ is a set of observed variables in the system. $\mathcal{N}$ is a set of independent exogenous random variables where $N_i \in \mathcal{N}$ affects $V_i$ and $\mathcal{U}$ is a set of unobserved confounders each affecting any two observed variables (for $> 2$ check Appendix C.7). This refers to the **semi-Markovian causal model**. A set of deterministic functions $\mathcal{F} = \{f_{V_1}, f_{V_2}, .., f_{V_n}\}$ determines the value of each variable $V_i$ from other observed and unobserved variables as $V_i = f_i(Pa_i, N_i, U_{S_i})$, where $Pa_i \subset \mathbf{V}$ (parents), $N_i \in \mathcal{N}$ (randomness) and $U_{S_i} \subset \mathcal{U}$ (**latent confounders**) for some $S_i$. $\mathcal{P}(.)$ is a product probability distribution over $\mathcal{N}$ and $\mathcal{U}$ and projects a **joint distribution** $\mathcal{P}_\mathbf{V}$ over the set of actions $\mathbf{V}$ representing their likelihood.

An SCM $\mathcal{M}$, induces an **acyclic directed mixed graph** (ADMG) $G = (\mathbf{V}, \mathcal{E})$ containing nodes for each variable $V_i \in \mathbf{V}$. For each $V_i = f_i(Pa_i, N_i, U_{S_i})$, $Pa_i \subset \mathbf{V}$, we add an edge $V_j \rightarrow V_i \in \mathcal{E}, \forall V_j \in Pa_i$. Thus, $Pa_i(V_i)$ becomes the parent nodes in $G$. $G$ has a **bi-directed edge**, $V_i \leftrightarrow V_j \in \mathcal{E}$ between $V_i$ and $V_j$ if and only if they share a latent confounder. If a path $V_i \rightarrow \ldots \rightarrow V_j$ exists, then $V_i$ is an ancestor of $V_j$, i.e., $V_i = An(V_j)_G$. An **intervention** $\text{do}(x)$ replaces the structural function $f_x$ with $X = x$ and in other structural functions where $X$ occurs. The distribution induced on the observed variables $V$ after such an intervention is represented as $\mathcal{P}_x(v)$ or $\mathcal{P}(v|\text{do}(x))$. Graphically, it is represented by $G_{\overline{X}}$ where incoming edges to $X$ are removed (marked red). With a slight abuse of notation, we will use $P(y)$ for both the numerical value $\mathbb{P}(Y = y)$ and the probability distribution $[P(y)]_y$, depending on the context. An example for the latter is:"*Let $Y$ be sampled from $P(y)$*". Also, $P_x(y)$ refers to the **interventional distribution** for all $x, y$. Given an ADMG $G$, a maximal subset of nodes where any two nodes are connected by bidirected paths is called a **c-component** $C(G)$. For any $S \in C(G)$, $P(S|\text{do}(\mathbf{V} \setminus S))$ is called a c-factor. We assume that we have access to the ADMG through some causal structure learning algorithm and expert knowledge. **Classifier-free diffusion guidance [20]** Let $(\mathbf{v}, \mathbf{c}) \sim P(\mathbf{v}, \mathbf{c})$ be the data distribution and $\mathbf{z} = \{\mathbf{z}_\lambda | \lambda \in [\lambda_{min}, \lambda_{max}]\}$ for $\lambda_{min} < \lambda_{max} \in \mathbb{R}$. We corrupt the data as $\mathbf{z}_\lambda = \alpha_\lambda \mathbf{x} + \sigma_\lambda \epsilon$ and optimize the denoising model by taking the gradient step on $\nabla_\theta ||\epsilon_\theta(\mathbf{z}_\lambda, \mathbf{c}) - \epsilon||^2$. Given that variables $\mathbf{V}$ are connected as a directed acyclic graph and we have diffusion models trained to learn the distributions $P(v_i|pa(v_i))$, we can perform **ancestral sampling** from the joint distribution, $P(\mathbf{v}) = \prod_{V_i \in \mathbf{V}} P(v_i|pa(v_i))$ by making one pass through each model in the topological order while sampling from the conditional distributions [6]. We choose classifier-free diffusion as our conditional model, but the choice changes based on the application. We use $M_V(c)$ and a square node as notation.

## 3 ID-GEN: generative model-based interventional sampling

Given a causal graph $G$, dataset $\mathcal{D} \sim P(\mathbf{v})$, our objective is to generate high-dimensional interventional samples from a query $P(\mathbf{y}|\text{do}(\mathbf{x}))$ or a conditional query $P(\mathbf{y}|\text{do}(\mathbf{x}), \mathbf{z})$. ID-GEN builds upon the recursive structure of the identification algorithm [46] to train necessary conditional models. Thus, we first discuss its connection with us and show the challenges it faces if deployed for sampling.

## 3.1 Identification algorithm (ID) and challenges with high-dimensional sampling

Shpitser and Pearl [46] propose a recursive algorithm (Algorithm 6) for estimating an interventional distribution $P_\mathbf{x}(\mathbf{y})$ given access to all probability tables. At any recursion level, it enters one of its four recursive steps: 2, 3, 4, 7 and three base case steps: 1, 5, 6 . Below, we discuss them in detail.

**Step 1** occurs when the intervention set $\mathbf{X}$ is empty in $P_\mathbf{x}(\mathbf{y})$. The effect of $\mathbf{X} = \emptyset$ on $\mathbf{Y}$ is its marginal $P(\mathbf{y})$ which is returned as output. **Step 2** checks if there exists any non-ancestor variable of $\mathbf{Y}$ in the intervention set $\mathbf{X}$. Such variables in the graph do not have any causal effect on $\mathbf{Y}$. Thus, it is safe to drop them. In **Step 3**, it searches for a set $\mathbf{W}$ in $G$, which does not effect $\mathbf{Y}$ assuming that $\mathbf{X}$ has already been intervened on. Thus, it can include $\mathbf{W}$ as an additional intervention set: $\mathbf{X} = \mathbf{X} \cup \mathbf{W}$. An intervention on $\mathbf{W}$ implies deleting its incoming edges, which simplifies the problem in the future. **Step 4** is the most important line and is executed when there are multiple c-components in the subgraph $G \setminus \mathbf{X}$. It factorizes (decomposes) the problem of estimating $P_\mathbf{x}(\mathbf{y})$ into estimating c-factors (subproblems) and performs recursive calls for each c-factor. Base case **Step 5** returns fail for non-identifiable queries. Base case **Step 6** asserts that when $\mathbf{X}$ does not have a bi-directed edge with the rest of the nodes in $S$ and $S$ consists of a single c-component, intervening on $\mathbf{X}$ is equivalent to conditioning on $\mathbf{X}$. Thus, ID can now solve $P_\mathbf{x}(\mathbf{y})$ as $\sum_{s \setminus \mathbf{y}} \prod_{i | V_i \in S} P(v_i | v_\pi^{(i-1)})$ and return as output. **Step 7** occurs when the variables in $\mathbf{X}$ can be partitioned into two sets: one having bi-directed edges to other variables ($S'$) in the graph and one (defined as $X_Z$) with no bi-directed edges to $S'$. In that case, evaluating $P_\mathbf{x}(\mathbf{y})$ from $P(\mathbf{V})$ is equivalent to first obtaining $P'(\mathbf{V}) := P_{\mathbf{x}_Z}(\mathbf{V})$ and then evaluating $P_{\mathbf{x} \setminus \mathbf{x}_z}(Y)$ from $P'(\mathbf{V})$. Hence, $P_{\mathbf{x}_Z}(\mathbf{V})$ is first calculated as $\prod_{\{i | V_i \in S'\}} P(V_i | V_\pi^{(i-1)} \cap S', v_\pi^{(i-1)} \setminus S')$ and then passed to the next recursive call for $\text{do}(\mathbf{x} \setminus \mathbf{x}_z)$ to be applied. One major issue of ID is that it requires probability tables and thus cannot be applied for high-dimensional sampling. Suppose we naively design an algorithm that follows ID's recursive steps and trains a generative model for every factor it encounters and samples from it. This algorithm would not know which of these factors to learn and sample first, leading to a deadlock as shown in Ex C.1. ID-GEN solves such issue by avoiding direct sampling and building a sampling network.

**Definition 3.1** (Sampling network, $\mathcal{H}$)**.** A collection of feedforward models $\{M_{V_i}\}_{\forall i}$ for a set of variables $\mathbf{V} = \{V_i\}_{\forall i}$ is said to form a **sampling network**, $\mathcal{H}$, if the directed graph obtained by connecting each $M_{V_i}$ to $M_{An(V_i)_G}$ via incoming edges according to some conditional distribution, is acyclic. Two sampling networks $\mathcal{H}_i, \mathcal{H}_r$ can be merged into a larger network $\mathcal{H}$.

## 3.2 Recursive training of ID-GEN and interventional sampling

Similar to ID's recursive structure, ID-GEN has 7 steps (Algorithm 1). However, to deal with high-dimensional variables, we call three new functions: i) Algorithm 2:`ConditionalGMs(.)` inside steps 1 and 6 where we train diffusion models or other conditional models to learn conditional distributions, ii) Algorithm 3:`MergeNetwork(.)` inside step 4 to merge the conditional models, and iii) Algorithm 4:`Update(.)` inside step 7 to train models that can apply part of the interventions and update the training dataset for next recursive calls. We initiate with ID-GEN ($\mathbf{Y}, \mathbf{X}, G, \mathcal{D}, \hat{\mathbf{X}} = \emptyset, \hat{G} = G$). Along with the given inputs $\mathbf{Y}, \mathbf{X}, G, \mathcal{D}$, ID-GEN maintains two extra parameters $\hat{\mathbf{X}}, \hat{G}$ to keep track of the interventions performed. During the top-down phase, ID-GEN updates its parameters: by i) removing interventions from the intervention set $\mathbf{X}$, and ii) updating the training dataset $\mathcal{D}, \hat{\mathbf{X}}$ and the causal graph $G, \hat{G}$ according to the interventions. At any level of recursion, an ID-GEN call returns a sampling network $\mathcal{H}$ (DAG of a set of trained models) trained on the dataset $\mathcal{D}$ to learn conditional distributions according to $\hat{\mathbf{X}}, G, \hat{G}$. After the recursion ends, we can generate samples from $P_\mathbf{x}(\mathbf{V}), \mathbf{Y} \subseteq \mathbf{V}$, by ancestral sampling on $\mathcal{H}$. See a recursion tree in Appendix C.4.

**Base Case: Step 1:** ID-GEN enters step 1 if the intervention $\mathbf{X}$ is empty. For $\mathbf{X} = \emptyset$, we have, $P_\mathbf{x}(\mathbf{y}) = P(\mathbf{y}) = \sum_{\mathbf{v} \setminus \mathbf{y}} P(\mathbf{v}) = \sum_{\mathbf{v} \setminus \mathbf{y}} \prod_{V_i \in \mathbf{V}} P(v_i | v_\pi^{(i-1)})$ which is suitable for ancestral sampling. To train models that can collectively sample from this distribution, we call Algorithm 2:`ConditionalGMs(.)`. Here, we train each model $M_{V_i}, \forall V_i \in \mathbf{V}$ using $V_\pi^{(i-1)}$, (i.e., variables that are located earlier in the topological order $\pi$) as inputs to match $P(v_i | v_\pi^{(i-1)})$. Note that $\hat{\mathbf{X}}$ contains the values that were intervened in previous recursion levels and $\hat{G}$ is the graph at the current level that contains $\hat{\mathbf{X}}$ with its incoming edges cut. Since we want our conditional models to generate samples consistent with the values of $\hat{\mathbf{X}}$, we consider the topological order of $\hat{G}$ while using

$V_\pi^{(i-1)}$ as inputs so that $\hat{\mathbf{X}}$ are also fed as input while training. After training, we connect the trained models according to their input-outputs to build a sampling network $\mathcal{H}$ and return it (Alg 2:lines 1-6). Note that when all variables in $\mathbf{Y}$ are low dimensional, we can also learn a single model $M$ to sample $P(\mathbf{Y})$. However, for high-dimensional variables, matching such joint distributions is non-trivial [40].

**Step 2 & 3**: We follow the same steps of the ID algorithm as discussed in Section 3.1.

**Step 4 and Merge sampling Networks:** Our goal is to train models that can sample from $P_{\mathbf{x}}(\mathbf{y})$ which unfortunately is not straightforward. This step allows us to decompose our problem into sub-problems and we can train models to sample from the c-factors of $P_{\mathbf{x}}(\mathbf{y})$'s factorization. The next challenge is to connect these models consistently to sample from $P_{\mathbf{x}}(\mathbf{y})$. More precisely, if we remove $\mathbf{X}$ from $G$ and the graph splits into multiple c-components (variables in each component connected with $\leftrightarrow$) (Alg 1:line 11), we can apply c-component fac-

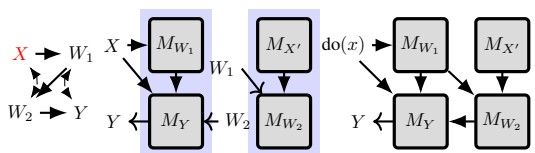

Figure 2: $\leftrightarrow$:Unobserved. Left blue samples from $P_{x,w_2}(w_1, y) = P(w_1|x)\ P(y|x, w_1, w_2)$. Right blue samples from $P_{x,w_1}(w_2) = \sum_{x'} P(x')$ $P(w_2|x', w_1)$. Joint network samples from $P_x(y)$.

torization (Lemma D.7, [51]) to factorize $P_{\mathbf{x}}(\mathbf{y})$ as $\sum_{\mathbf{v}\backslash(\mathbf{y}\cup\mathbf{x})} P_{\mathbf{v}\backslash s_1}(s_1)\dots P_{\mathbf{v}\backslash s_n}(s_n)$ where each $\{S_k\}_k$ is the c-factor corresponding to each c-component. To obtain trained models for each of these c-factors, we perform the next recursive calls: ID-GEN ($\mathbf{Y} = S_i, \mathbf{X} = \mathbf{V} \setminus S_i, G, \mathcal{D}, \hat{\mathbf{X}}, \hat{G}$). When these recursive calls return a sampling network $H_i$ for each $P_{\mathbf{v}\backslash s_i}(s_i)$, we can wire them based on their input-output to build a single sampling network $\mathcal{H}$. According to Theorem D.21 and D.22, $\mathcal{H}$ now can sample from $P_{\mathbf{x}}(\mathbf{y})$.

We call Algorithm 3: `MergeNetwork(.)` to connect all sampling networks $\{\mathcal{H}_i\}_{\forall_i}$. Here, each $\mathcal{H}_i$ is a set of trained conditional models $\{M_{V_j}\}_j$ connected to each other as a DAG. If a sampling network $\mathcal{H}_i$ contains an empty node $M_{V_j} = \emptyset$ without any conditional model and some other sampling network $\mathcal{H}_r$ generates this variable $V_j$ with its node $M_{V_k}$, i.e., $V_j = V_k$, then we combine $M_{V_j}$ and $M_{V_k}$ into the same node to build a connection between $\mathcal{H}_i$ and $\mathcal{H}_r$ (lines 3-6). Intuitively, due to the c-factorization at this step, the variables intervened in one sampling network might be generated from models in another network. We connect two networks to continue the ancestral sampling sequence. Fig. 2 shows an example of this step where $P_x(y)$ is factorized into c-factors as $P_x(y) = \sum_{w_1,w_2} P_{x,w_1}(w_2)P_{x,w_2}(w_1, y)$. For the c-component $\{W_1, Y\}$, ID-GEN first obtains $P_{x,w_2}(w_1, y) = P(w_1|x)P(y|x, w_1, w_2)$, and trains conditional models $M_{W_1}$ and $M_Y$ for these conditional distributions. Similarly, for $\{W_2\}$, we have $P_{x,w_1}(w_2) = \sum_{x'} P(x')P(w_2|x', w_1)$ and we train $M_{X'}$ and $M_{W_2}$. Finally, we merge these networks based on inputs-outputs to build a single sampling network and perform ancestral sampling on it to sample from $P_x(y), \forall x$.

**Base Case: Step 5:** We follow the step 5 of the ID algorithm as discussed in Section 3.1.

**Base Case: Step 6:** We enter Step 6 if $G \setminus \mathbf{X}$ is a single c-component $S$, and $\mathbf{X}$ is located outside the c-component. This situation allows us to replace the intervention on $\mathbf{X}$ by conditioning on $\mathbf{X}$: $P_{\mathbf{x}}(\mathbf{y}) = \sum_{s\backslash \mathbf{y}} \prod_{V_i \in S} P(v_i|v_\pi^{(i-1)})$. This step is similar to step 1, except that now we have a non-empty intervention set, i.e., $\mathbf{X} \neq \emptyset$. Here, we consider the topological order of $\hat{G}$ and $V_\pi^{(i-1)}$ contains both $\mathbf{X}$ and $\hat{\mathbf{X}}$. We call Algorithm 2:`ConditionalGMs(.)` which trains multiple conditional models to learn the above distribution. More precisely, we utilize classifier-free diffusion guidance for conditional training of each $M_{V_i}$ by taking the gradient step on $\nabla_\theta ||\epsilon_\theta(\mathbf{z}_\lambda^i, v_\pi^{(i-1)}) - \epsilon||^2$. Here, $z_\lambda^i$ is the noisy version of $V_i$ at time step $\lambda$ during the forward process and $v_\pi^{(i-1)}$ is the condition (see Background). Finally, we connect the input-output of these diffusion models according to the topological order to build a sampling network and return it as output. Note that for any specific conditional distribution, if we have access to a pre-trained models that can sample from it, we can directly plug it in the network instead of training it from scratch (motivated from [40]).

**Step 7:** Here, ID-GEN partitions $\mathbf{X}$ into two sets: one is applied in the current step to update the training dataset and other parameters, and the other is kept for future steps. It performs this step if i) $G \setminus \mathbf{X}$ is a single c-component $S$ and ii) $S$ is a sub-graph of a larger c-component $S'$ in the whole graph $G$, i.e, $(S = C(G \setminus \mathbf{X})) \subset (S' \in C(G))$. For example, in Fig. 3, for $P_{w_1,w_2,x}(y)$, we have $S = G \setminus \{W_1, W_2, X\} = \{Y\}, S' = \{W_1, X, Y\}$. In this step, we call Algorithm 4: `Update(.)`

**Algorithm 1** ID-GEN ($\mathbf{Y}, \mathbf{X}, G, \mathcal{D}, \hat{\mathbf{X}}, \hat{G}$)

1: **Input:** target $\mathbf{Y}$, to be intervened $\mathbf{X}$, intervened variables at step 7s $\hat{\mathbf{X}}$, causal graph $G$ without $\hat{\mathbf{X}}$, causal graph $\hat{G}$ with $\hat{\mathbf{X}}$ having no parents, training data $\mathcal{D}[\hat{G}]$ sampled from observed distribution $P(\mathbf{V})$.

2: **Output:** A sampling network of trained models.
3: **if** $\mathbf{X} = \emptyset$ **then** {Step 1}
4:     **Return** ConditionalGMs($\mathbf{Y}, \mathbf{X} = \emptyset, G, \mathcal{D}, \hat{\mathbf{X}}, \hat{G}$)
5: **if** $\mathbf{V} \setminus An(\mathbf{Y})_G \neq \emptyset$ **then** {Step 2}
6:     **Return** ID-GEN($\mathbf{Y}, \mathbf{X} \cap An(\mathbf{Y})_G, G_{An(\mathbf{Y})}, \hat{\mathbf{X}},$ $\hat{G}_{An(\mathbf{Y})}, \mathcal{D}' = \mathcal{D}[An(\mathbf{Y})_G]$ )
7: Let $\mathbf{W} = (\mathbf{V} \setminus \mathbf{X}) \setminus An(\mathbf{Y})_{G_{\overline{\mathbf{X}}}}$ {Step 3}
8: **if** $\mathbf{W} \neq \emptyset$ **then**
9:     **Return** ID-GEN($\mathbf{Y}, \mathbf{X} = \mathbf{X} \cup \mathbf{W}, G, \hat{\mathbf{X}}, \hat{G}, \mathcal{D}$)
10: **if** $C(G \setminus \mathbf{X}) = \{S_1, \dots, S_k\}$ **then** {Step 4}
11:     **for** each $S_i \in C(G \setminus \mathbf{X}) = \{S_1, \dots, S_k\}$ **do**
12:         $\mathcal{H}_i$=ID-GEN($S_i, \mathbf{X} = \mathbf{V} \setminus S_i, G, \hat{\mathbf{X}}, \hat{G}, \mathcal{D}$)
13:     **Return** MergeNetwork($\{\mathcal{H}_i\}_{\forall i}$)
14: **if** $C(G \setminus X) = \{S\}$ **then**
15:     **if** $C(G) = \{G\}$ **then** {Step 5}
16:         throw FAIL
17:     **if** $S \in C(G)$ **then** {Step 6}
18:         **Return** ConditionalGMs($S, \mathbf{X}, G, \mathcal{D}, \hat{\mathbf{X}}, \hat{G}$)
19:     **if** ($\exists S'$) such that $S \subset S' \in C(G)$ **then** {S7}
20:         **Return** ID-GEN(Update($S', \mathbf{X}, G, \mathcal{D}, \hat{\mathbf{X}}, \hat{G}$))

---

**Algorithm 2** ConditionalGMs($\mathbf{Y}, \mathbf{X}, G, \mathcal{D}, \hat{\mathbf{X}}, \hat{G}$)

1: **for** each $V_i \in \{\mathbf{X} \cup \hat{\mathbf{X}}\}$ **do**
2:     Add node $(V_i, \emptyset)$ to $\mathcal{H}$ {Initialized $\mathcal{H} = \emptyset$}
3: **for** each $V_i \in \mathbf{Y}$ in the topological order $\pi_{\hat{G}}$ **do**
4:     Let $M_{V_i}$ be a model trained on $\mathcal{D}[V_i, V_\pi^{(i-1)}]$ such that $M_{V_i}(V_\pi^{(i-1)}) \sim P(v_i | v_\pi^{(i-1)})$
5:     Add node $(V_i, M_{V_i})$ to $\mathcal{H}$
6:     Add edge $V_j \to V_i$ to $\mathcal{H}$ for all $V_j \in V_\pi^{(i-1)}$
7: **Return** $\mathcal{H}$.

---

**Algorithm 3** MergeNetwork($\{\mathcal{H}_i\}_{\forall i}$)

1: **Input:** Set of sampling networks $\{\mathcal{H}_i\}_{\forall i}$.
2: **Output:** A connected DAG sampling network $\mathcal{H}$.
3: **for** $H_i \in \{\mathcal{H}_i\}_{\forall i}$ **do**
4:     **for** $M_{V_j} \in \mathcal{H}_i$ **do**
5:         **if** $M_{V_j} = \emptyset$ and $\exists M_{V_k} \in \mathcal{H}_r, \forall r$ such that $V_j = V_k$ and $M_{V_k} \neq \emptyset$ **then**
6:             $M_{V_j} = M_{V_k}$
7: **Return** $\mathcal{H} = \{\mathcal{H}_i\}_{\forall i}$ {All $\mathcal{H}_i$ are connected.}

---

**Algorithm 4** Update($S', \mathbf{X}, G, \mathcal{D}, \hat{\mathbf{X}}, \hat{G}$)

1: $\mathbf{X}_Z = \mathbf{X} \setminus S'$
2: $\mathcal{H} = $ ConditionalGMs($S', \mathbf{X}_Z, G, \mathcal{D}, \hat{\mathbf{X}}, \hat{G}$)
3: $\mathcal{D}' \sim \mathcal{H}(\mathbf{X}_Z, \hat{\mathbf{X}})$;     $\hat{\mathbf{X}} = \hat{\mathbf{X}} \cup \mathbf{X_Z}$
4: **Return** $\mathbf{Y}, \mathbf{X} \cap S', G_{S'}, \mathcal{D}'[\hat{\mathbf{X}}, S'], \hat{\mathbf{X}}, \hat{G}_{\{S', \overline{\hat{\mathbf{X}}}\}}$

---

which utilizes the larger c-component $S'$ to partition the intervention set $\mathbf{X}$ into one set contained within $S'$, i.e., $\mathbf{X} \cap S'$, and another set not contained in $S'$, i.e., $\mathbf{X}_Z = \mathbf{X} \setminus S'$. Evaluating $P_{\mathbf{x}}(\mathbf{y})$ from $P(\mathbf{v})$ is equivalent to evaluating $P_{\mathbf{x} \cap s'}(\mathbf{y})$ from $P'(\mathbf{v})$ where $P'(\mathbf{v}) \coloneqq P_{\mathbf{x}_z}(\mathbf{v})$ is the joint distribution. Hence, we first perform $\text{do}(\mathbf{X}_Z)$ to update the dataset as $\mathcal{D}'$. Next, we shift our goal of sampling from $P_{\mathbf{x}}(\mathbf{y})$ in $G$ with training dataset $\mathcal{D} \sim P(\mathbf{V})$ to sampling from $P_{\mathbf{x} \cap s'}(\mathbf{y})$ in $\hat{G}_{\{S', \overline{\hat{\mathbf{X}}}\}}$ with training data $\mathcal{D}' \sim P_{\mathbf{x}_z}(\mathbf{v})$ in the next recursive calls. To generate dataset $\mathcal{D}' \sim P_{\mathbf{x}_z}(\mathbf{v})$, we call ConditionalGMs(.) and use the returned network to sample $\mathcal{D}'$ (lines 2-3).

Note that given access to probability tables, the ID can use any specific value $\mathbf{X}_Z = \mathbf{x}_z$ to calculate $P_{\mathbf{x}_z}(\mathbf{v})$ to get the correct estimation of $P_{\mathbf{x}}(\mathbf{y})$ (Verma constraint [54, 46]). In our case, if we use a specific value $\mathbf{x}_z$ to sample the training dataset $\mathcal{D}' \sim P_{\mathbf{x}_z}(\mathbf{v})$, the models trained on this dataset in subsequent recursive steps will also depend on $\mathbf{x}_z$. However, during ancestral sampling in the returned network, a different value $\mathbf{X}_Z = \mathbf{x}'_z$ might come from other c-components (ex: $M_Y(W_2, .)$ in Fig. 3). Thus, to make our trained models suitable for any values, we pick $\mathbf{X}_Z$ from a uniform distribution or from $P(\mathbf{X}_Z)$ and generate $\mathcal{D}'$ accordingly. We save $\mathbf{X}_Z$ in $\hat{\mathbf{X}}$, its values in $\mathcal{D}[\hat{\mathbf{X}}, S']$ and in $\hat{G}_{\{S', \overline{\hat{\mathbf{X}}}\}}$ with incoming edges removed, to be considered during training in the next recursive calls. Whenever ID-GEN visits Step 7 again, $\hat{\mathbf{X}}$ will be applied along with the new $\mathbf{X}_Z$. Finally, a recursive call is performed with these updated parameters (line 4) which will return a network trained on dataset $\mathcal{D}' \sim P_{\mathbf{x}_z}(\mathbf{v})$. It can sample from $P_{\mathbf{x} \cap S'}(\mathbf{y})$ and equivalently from the original $P_{\mathbf{x}}(\mathbf{y})$.

**Algorithm simulation:** We apply ID-GEN to sample from $P_{w_1}(y)$ for the causal graph $G$ in Fig. 3. Since $G \setminus \{W_1\}$ has three c-components $\{W_2\}, \{X\}, \{Y\}$, we first call (i) step 4 of ID-GEN. $P_{w_1}(y)$ is factorized as: $\sum_{x, w_2} P_{w_1, x, y}(w_2) P_{w_1, w_2, y}(x) P_{w_1, w_2, x}(y)$. Thus, step 4 will return the sampling networks $\{H_{W_2}, H_X, H_Y\}$ that can sample from each of these factors. Here, we focus only on $H_Y$. ID-GEN reaches (ii) step 7 for the query: $P_{w_1, w_2, x}(y)$ since we have $S = G \setminus \{W_1, W_2, X\} = \{Y\}, S' = \{W_1, X, Y\}$ and $S \subset S'$. Here, sampling from $P_{x, w_1, w_2}(y)$ in $G$, with observational training dataset is equivalent to sampling from $P_{x, w_1}(y)$ in $\hat{G} = G_{\overline{W_2}}$ with $\text{do}(W_2)$ interventional data. With $W_2 \sim P(w_2)$, we generate $\mathcal{D}' \sim P_{w_2}(\mathbf{v})$ by calling step 6 (base case). We pass $\mathcal{D}'$ as the dataset parameter for the next recursive call. This step implies that if the recursive call returns a network that is trained on $\mathcal{D}' \sim P_{w_2}(\mathbf{v})$ and can sample from $P_{x, w_1}(y)$, it can also be used to sample

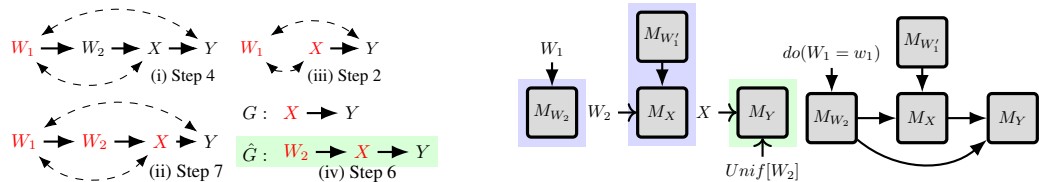

Figure 3: (Left: top-down) $P_{w_1}(y)$ is factorized into $P_{w_1,x,y}(w_2)$, $P_{w_1,w_2,y}(x)$ and $P_{w_1,w_2,x}(y)$ (Step 4). Steps 7, 2, 6 is shown for $P_{w_1,w_2,x}(y)$ only. (Right: bottom-up) we combine the sampling networks of each c-factor. For any $\text{do}(W_1 = w_1)$, we use $\mathcal{H}$ to get samples from $P_{w_1}(y)$.

from $P_{w_1,w_2,x}(y)$. Next, since $W_1 \notin An(Y)_G$, at (iii) step 2, we drop $W_1$ from all parameters before the next recursive call. We are at the base case (iv) step 6 with $\hat{G} : W_2 \to X \to Y$. Thus, we train a conditional model $M_Y(W_2, X)$ on $\mathcal{D}'[W_2, X, Y]$ that can sample from $P(y|w_2, x)$. This would be returned as $H_Y$ at step 4 (Fig. 3:green). Similarly, we can obtain sampling network $H_{W_2}$ and $H_X$ to sample from $P_{w_1,x,y}(w_2) = P(w_2|w_1)$ and $P_{w_1,w_2,y}(x) = \sum_{w'_1} P(x|w'_1, w_2)P(w'_1)$ (Fig. 3:blue). We connect these networks and perform ancestral sampling with fixed $w_1$ for $\text{do}(W_1 = w_1)$.

**(Un)conditional sampling and complexity:** ID-GEN returns a sampling network $\mathcal{H}$ when recursion ends. For unconditional query $P_\mathbf{x}(\mathbf{y})$, we fix $\mathbf{X} = \mathbf{x}$ in $\mathcal{H}$ and perform ancestral sampling to generate joint samples. We pick the $\mathbf{Y}$ values in these joint samples (equivalent to marginalization in ID) and report as interventional samples. For a conditional query $P_\mathbf{x}(\mathbf{y}|\mathbf{z})$, ID-GEN uses the sampling network to first generate samples $\mathcal{D}[\mathbf{X}, \mathbf{Z}, \mathbf{Y}] \sim P_\mathbf{x}(\mathbf{y}, \mathbf{z})$ and then train a new conditional model $M_\mathbf{Y}(\mathbf{X}, \mathbf{Z})$ on $\mathcal{D}$ to sample from $\mathbf{Y} \sim P_\mathbf{x}(\mathbf{y}|\mathbf{z})$ (Alg.5:IDC-GEN). The sampling network has $\mathcal{O}(|An(\mathbf{Y})_G|)$ number of models and requires $\mathcal{O}(|An(\mathbf{Y})_G|)$ time to sample from it. Please, see our complexity details in Appendix C.5, Appendix C.6.

**Theorem 3.2.** *Under Assumptions: i) the SCM is semi-Markovian, ii) we have access to the ADMG, iii) $P(\mathbf{V})$ is strictly positive and iv) trained generative models sample from correct distributions, ID-GEN and IDC-GEN are sound and complete to sample from any identifiable $P_x(y)$ and $P_x(y|z)$.*

Note that although existing work can sample from (low-dimensional) distributions, Theorem 3.2, makes ID-GEN, to our knowledge, the first method to use **only feed-forward models** to provably sample from identifiable high-dimensional interventional distributions.

# 4 Experiments

To illustrate ID-GEN's capabilities with high-dimensional image and text variables, we evaluate it on semi-synthetic: Colored MNIST and real-world: CelebA and MIMIC-CXR datasets. We provide additional details in Appendix F. Codes are available at github.com/musfiqshohan/idgen.

## 4.1 ID-GEN performance on napkin-MNIST dataset and baseline comparison

**Setup:** We consider a semi-synthetic Colored-MNIST dataset for the napkin graph [39] in Fig. 4 with image variables $W_1, X, Y$ and paired discrete variable $W_2$. Here, $X$ and $Y$ inherit the same digit value as image $W_1$ which is propagated through discrete $W_2.d \in [0-9]$. $X$ is either red or green which is also inherited from $W_1$ through discrete $W_2.c$, i.e., $W_1.color : \{r, g, b, y, m, cy\} \to W_2.c : \{0, 1\} \to X.color : \{r, g\}$. Unobserved $Ucolor$ makes $W_1$ and $Y$ correlated with the same color, and unobserved $Uthickness$ makes $W_1$ and $X$ correlated with the same thickness. Even though image $X$ (takes r and g) is a direct ancestor of image $Y$ (takes all 6 colors), $Y$ only inherits the digit property from $X$ but correlates the color property with image $W_1$. This color correlation between $W_1$ and $Y$ is created by Ucolor. All mechanisms include $10\%$ noise. Our target is to sample from $P_x(y)$.

**Training and evaluation:** We follow ID-GEN steps: [3, 7, 2, 6]. Step 3 implies $P_x(y) = P_{x,w_1,w_2}(y)$, i.e., intervention set $=\{X, W_1, W_2\}$. Step 7 suggests to generate $\text{do}(W_2)$ interventional dataset $\mathcal{D}'[W_1, W_2, X, Y] \sim P_{w_2}(w_1, x, y) = P(w_1)P(x, y|w_1, w_2)$. To obtain $\mathcal{D}'$, we i) sample $W_1 \sim P(w_1)$, and ii) train a conditional diffusion model to sampling from $P(x, y|w_1, w_2)$ with arbitrary $W_2$ values. Next, Step 2 drops non-ancestor $W_1$ and Step 6 trains a diffusion model $M_Y(x, w_2)$ on the new dataset $\mathcal{D}'$ to sample from $P'(y|x, w_2)$. $M_Y(x, w_2)$ is returned as output that can sample from $P_x(y), \forall w_2$. We compare our performance with three baselines: i) a classifier-free diffusion

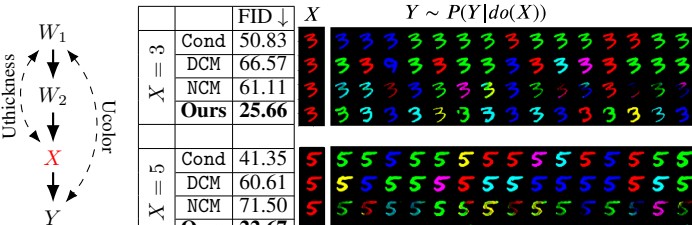

Center table (FID scores):

| | | FID ↓ |
|---|---|---|
| X = 3 | Cond | 50.83 |
| | DCM | 66.57 |
| | NCM | 61.11 |
| | **Ours** | **25.66** |
| X = 5 | Cond | 41.35 |
| | DCM | 60.61 |
| | NCM | 71.50 |
| | **Ours** | **22.67** |

Right table:

| Color | True $P_x(y)$ | Cond | DCM | **Ours** |
|---|---|---|---|---|
| R | 0.17 | 0.17 | 0.24 | **0.13** |
| G | 0.17 | 0.45 | 0.27 | **0.24** |
| B | 0.17 | 0.15 | 0.28 | **0.21** |
| Yw | 0.17 | 0.19 | 0.05 | **0.14** |
| Mg | 0.17 | 0.02 | 0.07 | **0.11** |
| Cy | 0.17 | 0.10 | 0.09 | **0.18** |
| TVD | 0 | 0.54 | 0.58 | **0.25** |

Figure 4: (Left:) Causal graph with color and thickness as unobserved. (Center:) FID scores (lower the better) of each algorithm and images generated from them. (Right:) Likelihood calculated from the $P_x(y)$ images generated by each algorithm. We closely reflect the true $P_x(y)$ with low TVD.

model that samples from the (Cond)itional distribution $P(y|x)$, ii) the DCM algorithm [11] that uses diffusion models to samples from $P_x(y)$ but without confounders, and iii) the NCM algorithm [58] that uses GANs and considers confounders. We performed $do(x)$ intervention with two images, i) digit 3 and ii) digit 5, both colored red. In Fig. 4, we show interventional samples for each method alongside their FID scores representing the image quality (lower:better). The (Cond) model (row 1, 5), DCM (row 2, 6) and our algorithm (row 5, 8) all generate good quality images of digit 3 and digit 5 with a specific color. However, the NCM algorithm (row 3, 7) generates images with blended colors (such as green + red). We observe that ID-GEN achieves the lowest FID scores (25.66 and 22.67), showing the ability to generate high-quality images consistent with the dataset. Whereas, Cond and DCM generate almost the same structure for all digits lacking variety, which explains their high FID. Note that $do(x)$ removes the color bias between $X$ and $Y$ along the backdoor path. Thus, interventional samples should show all colors with uniform probability. Since Cond and DCM can not deal with confounders they show bias towards R, G, B colors of $Y$ for red $X$. ID-GEN removes such bias and balances different colors (Fig. 4). For a more rigorous evaluation, we use the effectiveness metric proposed in [34] and employ a classifier to map all generated images to discrete analogues $(Digit, Color, Thickness)$ and compute exact likelihoods. We compare them with our ground truth $P(Y.color|do(x))$ (uniform) and display these results for the color attribute in Fig. 4(right). We emulate the interventional distribution more closely with a low total variation distance: 0.25 compared to the baselines Cond (0.54) and DCM (0.58). We skip classifying colors of NCM as they are blended.

## 4.2 Evaluating CelebA image translation models with ID-GEN

**Setup:** We apply ID-GEN to evaluate multidomain image translation of some existing generative models (ex: Male to Female domain translation). We examine whether they apply causal changes in (facial) attributes or add unnecessary changes due to the spurious correlations among different attributes they picked up in the training data. Our application is motivated by Goyal et al. [15], who generate counterfactual images to explain a pre-trained classifier while we examine pre-trained image generative models. We employ two generative models that are trained on CelebA dataset [29]: i) StarGAN [12] and ii) EGSDE [64] (an approach that utilizes energy-guided stochastic differential equations). We assume the graph in Fig. 5 where the original image $I_1$ causes its own attributes Male and Young. We consider an unobserved confounder between them, as in the dataset, men are more likely to be old (correlation coeff=0.42, [44]) and a classifier might have some bias toward predicting young-male images as old-male. These attributes along with the original image

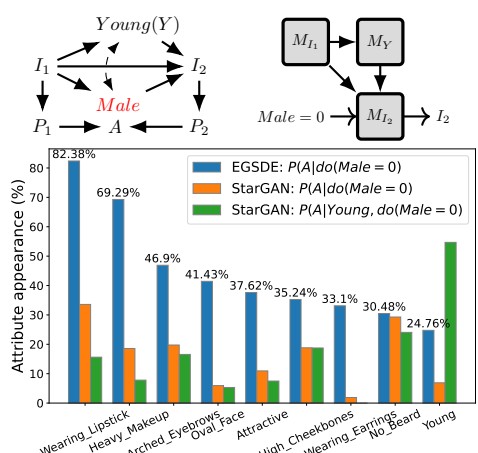

Figure 5: i) Graph and sampling network for $P_{Male}(I_2)$. ii) For both causal and non-causal attributes, EGSDE shows high correlation.

are used to generate a translated image $I_2$. Next, $P_1$ and $P_2$ are 40 CelebA attributes of $I_1$ and $I_2$. $A$ is the difference between $P_1, P_2$, i.e., the additional attributes (ex: makeup) that gets added to $I_2$ but are absent in $I_1$ during translation. We estimate $P_{Male=0}(A)$, i.e., the causal effect of changing the domain from Male to Female on the appearance of a new attribute.

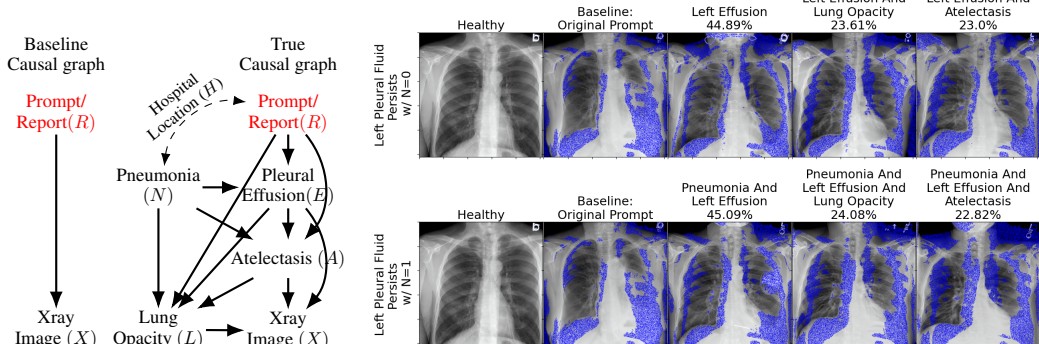

Figure 6: Left: Baseline vs our causal graph. Right: images for specific prompt w/ and w/o pneumonia. Inferred attributes are shown with their likelihood. Blue indicates changes compared to healthy.

**Training and Evaluation**: For $P_{Male}(A)$, We first generate $I_2 \sim P_{\text{Male}=0}(I_2)$ and then use a classifier on $I_2$. ID-GEN vists steps: $4, 6$ and factorizes as $P_{\text{Male}=0}(I_2) = \int_{Y,I_1} P(I_2|\text{Male} = 0, Y, I_1) * P(Y, I_1)$. To sample from these factors, it first trains $M_Y, M_{I_1}$. Next, instead of training $M_{I_2}$ for $P(I_2|\text{Male} = 0, Y, I_1)$, we plug in the model that we want to evaluate (StarGAN or EGSDE) as $\tilde{M}_{I_2}$. We connect these models to build the sampling network in Fig. 5. We can now perform ancestral sampling in the network with $\text{Male} = 0$ and generate samples of $I_2$. Next, we use a classifier to obtain 40 attributes of $I_1$ and $I_2$ as $P_1$ and $P_2$. Finally, we obtain the added attributes, $A$ by comparing $P_1$ and $P_2$ and report the proportion as the estimate of $P_{\text{Male}=0}(A)$. Similarly, we also estimate the conditional query, $P_{\text{Male}=0}(A|Y)$, using StarGAN with $Y = 1$ fixed.

In Fig. 5, we show top 9 most appeared attributes. We observe that EGSDE introduces both causal (ex: WearingLipstick to $82\%$, HeavyMakeup to $69.28\%$ of all images) and non-causal attributes (ex:Attractive to $37.61\%$ and Young to $24.76\%$) with high probability. The model might assign high-probability to non-causal attributes because they were spuriously correlated in the CelebA training dataset (ex: sex and age). On the other hand, StarGAN and conditional StarGAN introduce new attributes with a low probability ($\leq 30\%$) even if they are causal which is also not preferred. These evaluations enabled by ID-GEN help us to understand the fairness or bias in the prediction of image translation models. Finally, for the conditional query $P_{\text{Male}=0}(A|Y)$, StarGAN translates $54.68\%$ of all images as $\text{Young}, \text{Female}$ which is consistent. In Appendix F.3, we discuss how baselines deal poorly with these queries.

### 4.3 Invariant prediction with foundation models for chest X-ray generation

**Setup:** In this section, we demonstrate ID-GEN's utility to intervene with a text variable and generate images from the corresponding interventional distribution. Due to the lack of high-quality, annotated medical imaging datasets, the task of generating X-ray images $(X)$ from the prescription reports $(R)$ is recently being investigated. Consider the report-to-Xray generation task shown in Fig. 6. The shown causal graph is designed based on the MIMIC-CXR dataset [24] (see Appendix F.4 for details). Existing works [10] solve this task by using vision-language foundation models to directly generate CXR images conditioned on text prompts (or report): $P(X|R)$(Fig. 6 left). These models can generate images satisfying the prompt attributes (ex: effusion). However, they ignore the status of other attributes caused by prompt attributes (effusion→atelectasis) that might implicitly contribute to generate the image. Moreover, these models are trained on multiple datasets collected from different locations, which could introduce different types of bias [9]. For example, in many scenarios, patients with severe pneumonia are transferred to a different hospital and receives critical reports. Thus, there exists a potential confounding bias between report $(R)$ writing style and pneumonia prevalence $(N)$ in a specific location $(H)$ i.e. $R \leftarrow H \rightarrow N$. The direct $R \rightarrow X$ prediction of the above mentioned conditional models might absorb the backdoor bias: $R \leftarrow H \rightarrow N \rightarrow ... \rightarrow X$ and get affected if $P(R, N)$ shifts in a new location. We aim to make the image generation task of such models, interpretable and invariant in the test domain. Therefore, we consider the same input (report variable) and output (X-ray image) as such these models. For this purpose, ID-GEN performs $\text{do}(R)$ intervention to remove the backdoor bias and infer attributes from the interventional distribution. Now these attributes are used to generate the final images (motivated by [48, 27]).

**Training and Evaluation:** We indicate attribute extraction with outgoing edges from report $R$ and image generation with incoming edges to X-ray $X$. We assume pneumonia infection $N$ affects other attributes. For our target query $P_R(N, E, A, L, X)$, ID-GEN first visits step 4 and factorizes it as $P_R(N, E, A, L, X) = P(N)P_{R,N}(E)P_{R,N,E}(A)P_{R,N,E,A}(L)P_{E,A,L}(X)$. ID-GEN goes to step 6 for each, converts them to conditional distributions and trains corresponding models or use pre-trained ones for $M_N, M_E, M_A, M_L, M_X$. We utilize an LLM labeler [16] that can only extract attributes $E, A, L$ from the report $R$ (thus $R \not\rightarrow N$). If $R$ explicitly mentions $E, A, L$ then the models $M_E, M_A, M_L$ output 1. Otherwise, they probabilistically infer their output conditioned on parent attributes in the causal graph. The models we train are able to converge to their conditional distribution with a low total variation distance ($\approx 0.05$). We utilize a stable diffusion-based model [10] as $M_X$ to sample $X \sim P_{E,A,L}(X)$. After ID-GEN connects all models, we intervene with a prompt, infer all attributes along with their empirical distribution and generate corresponding X-rays. In Fig. 6 (right), we show our results for the prompt intervention do($R =$ left pleural fluid persists). For better visualization, we also condition on $N = 0$ and $N = 1$ separately. First, the LLM labeler extracts effusion ($E = 1$) from the prompt. Next, ancestral sampling with $E = 1$ in the sampling network provides us with the top 3 most probable attribute combination ($N = 0/1, E = 1, A, L$) and their corresponding images. In Fig. 6, we provide (i) a healthy image, (ii) image generated by baseline [10], and (iii)-(v) images generated with ID-GEN and their attributes with empirical probabilities. ID-GEN makes these predictions interpretable while facilitating them to be invariant with domain shifts.

## 5 Related work

Shalit et al. [43], Louizos et al. [30], Zhang et al. [63], Vo et al. [55] propose novel approaches to solve the causal effect estimation problem using variational autoencoders. Their proposed solution and theoretical guarantees are tailored for specific causal graphs containing treatment, effect, and covariates (or observed proxy variables), where they can apply the backdoor adjustment formula. Sanchez and Tsaftaris [42] employ DDPMs [22] to generate high-dimensional interventional samples, but only for bivariate models. Kocaoglu et al. [26] perform adversarial training on a collection of conditional generative models following the topological order to sample from interventional distributions. Pawlowski et al. [35] employ a conditional normalizing flow-based approach to offer high-dimensional interventional sampling as part of their solution. Chao et al. [11] designs diffusion-based causal models for arbitrary causal graphs with classifier-free guidance [21]. However, these works have limited applications due to their strong assumption of no latent confounders in the system.

Xia et al. [58] propose a similar training process as Kocaoglu et al. [26] in the presence of hidden confounders. They show explicit connections with the Causal Hierarchy Theorem [3] and formalize the identification problem with neural models. However, it is difficult to match an arbitrary high-dimensional distribution, and their joint GAN training approach may suffer from convergence issues. Rahman and Kocaoglu [40] utilizes a modular algorithm to relax the joint training restriction for specific structures, but might face the convergence issue when the number of high-dimensional variables increases. Note that these methods are not suitable for efficient conditional sampling. Jung et al. [25] convert the expression returned by identification algorithm [51] into a form where it can be computed through a re-weighting function to allow sample-efficient estimation. Similarly Bhattacharyya et al. [4, 5] utilize bounded number of samples from the observational distribution to construct an interventional sampler. However, computing these reweighting functions/conditional distributions from data is still highly nontrivial with high-dimensional variables. Zečević et al. [61] models each probabilistic term in the expression obtained for $P(y|\text{do}(x))$ with (i)SPNs. However, they require access to the interventional data and do not address identification from only observations. Wang and Kwiatkowska [56] design a novel probabilistic circuit architecture to encode the target causal query and estimate causal effects but do not offer high-dimensional sampling.

## 6 Conclusion

We propose a sound and complete algorithm to sample from conditional or unconditional high-dimensional interventional distributions. Our approach is able to leverage the state-of-the-art conditional generative models by showing that any identifiable causal effect estimand can be sampled from efficiently, only via feed-forward models. In future, we aim to relax our assumption on access to the ADMG and adapt our algorithm to deal with soft interventions.

## Acknowledgments and Disclosure of Funding

This research has been supported in part by NSF CAREER 2239375, IIS 2348717, Amazon Research Award and Adobe Research.

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

## A    Limitations and future work

In this section, we list some future directions of our work. i) We assume that we have access to the fully specified causal graph (ADMG) and the causal model is semi-MarkoviaN. Although these are common assumptions in causal inference, we aim to extend our algorithm for structures with more uncertainty (ex: PAGs). ii) In certain cases, the identification algorithm acts on particular variables, yet the expression for the causal effect does not depend on these variables. However, given that our sample size is limited, distinct values of these variables could affect the accuracy of the causal effect [19]. Since our algorithm follows the same recursive trace as the identification algorithm, it might suffer from the same problem. In addition, some model training might be unnecessarily performed due to this issue. To avoid this scenario, the pruning algorithm proposed in [53] might be merged with us to reduce the redundant cost of model training. iii) Since we target a specific causal query, if the query is substantially changed (causal effect on new variables), we might have to re-train some models. iv) For causal effect estimation in high-dimension, we follow the ID algorithm [45] to consider a hard intervention $\text{do}(X = x)$ i.e., fixing a specific value of the intervened variable, and assume that the rest of the conditional distribution stays unchanged. An interesting future direction would be to consider imperfect/soft intervention where the underlying mechanism of intervention changes instead of being fixed to a specific value.

## B    Broader impact

During the last few years, researchers have proposed many deep learning-based approaches to learn the unknown structural causal model from available data and employ the learned model to estimate the causal effect or sample from the high-dimensional interventional distribution. However, all these methods propose solutions tailored to specific neural architectures. As a result, when better generative models appear (such as diffusion models), existing methods lack the flexibility to utilize them, and thus we are in need of a new causal sampling method to get benefits of the new architecture. ID-GEN proposes a generic method that is independent of any model architecture and can generate high-dimensional interventional samples with any model architecture (GANs, VAE, Normalizing flow, diffusion models, etc.) as long as that model has the ability to generate conditional samples. Thus, our algorithm would allow the causal community to always use the latest generative model for high-dimensional causal sampling-based applications.

On a different note, the ability to sample from interventional distributions may enhance deep-generative models to obtain fake data, which can be exploited similarly to fake image generation. Since our algorithm considers causal relations among different variables, it has the ability to generate comparatively sensible and realistic fake images. Fake images can be a source of distress or might influence public actions and opinion. Thus, careful deployment of should be considered for deep causal generative models and the same procedures to detect fake image should be taken.

## C    ID-GEN additional discussion

In this section, we provide more details about our algorithm ID-GEN.

### C.1    Sampling from any interventional distribution with ID-GEN

Here we provide the full form of the causal query $P(\mathbf{v}|\text{do}(r))$ for the graph in Figure 7. Each term is expressed in terms of observational distribution. Note that it is nonintuitive and nontrivial for existing algorithms to sample from this complicated expression. Our algorithm ID-GEN solves this problem by training a specific set of models in a recursive manner and building a sampling network combining them. Finally, ID-GEN uses this network to sample from the following interventional distribution.

$$
\begin{aligned}
P(\mathbf{v}|\text{do}(r)) &= P(w_2, w_3|\text{do}(r))P(w_4|\text{do}(w_3))P(w_1|\text{do}(\mathbf{v} \setminus \{w_1, x\}))P(x|\text{do}(\mathbf{v} \setminus \{x\})) \\
&= P(w_2, w_3|r) * P(w_4|w_3) * \sum_{r'} P(w_1|r', w_2, w_3, w_4)P(r'|w_3, w_4) \\
&\quad * \frac{\sum_{r'} P(w_1, x|r', w_2, w_3, w_4)P(r'|w_3, w_4)}{\sum_{r'} P(w_1|r', w_2, w_3, w_4)P(r'|w_3, w_4)}
\end{aligned}
\tag{1}
$$

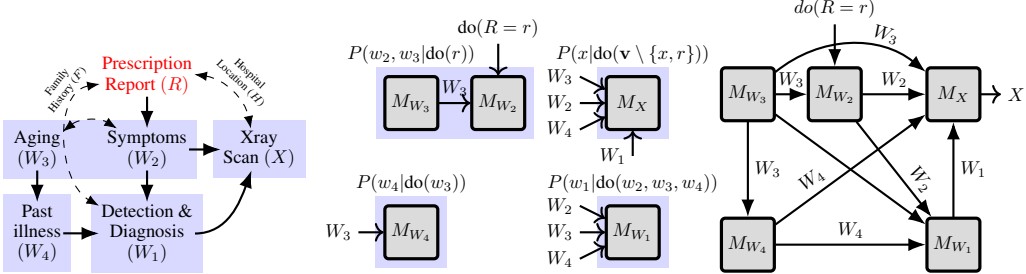

(a) ML model failure scenario    (b) Train and build sampling network.    (c) Merge and sample $P(y|\text{do}(x))$.

Figure 7: $\text{do}(R = r)$ removes the $R \leftrightarrow X$ bias and makes prediction of $X$ domain invariant. ID-GEN factorizes $P(\mathbf{v}|\text{do}(r))$ into four factors and trains conditional models ($\{M_{V_i}\}_i$) for each (blue shades). The intervened value $R = r$ is propagated through the merged network to generate all other variables.

## C.2   Cyclic dependency dealt with ID-GEN

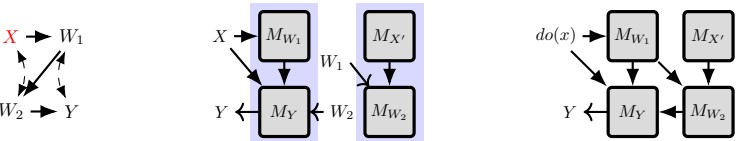

Figure 8: $\leftrightarrow$:Confounding. $M_{W_1}$, $M_Y$ sample from $P_{x,w_2}(w_1, y) = P(w_1|x) \, P(y|x, w_1, w_2)$. $M_{X'}$, $M_{W_2}$ sample from $P_{x,w_1}(w_2) = \sum_{x'} P(x') \, P(w_2|x', w_1)$. Joint network samples from $P_x(y)$.

**Example C.1** (Cyclic dependency). For the graph in Fig. 8, $P_x(y)$ does not fit any familiar criterion such as backdoor or front door. ID algorithm factorizes $P_x(y)$ into c-factors at its step 4, as $P_x(y) = \sum_{w_1, w_2} P_{x,w_1}(w_2) P_{x,w_2}(w_1, y)$ and estimates each of them recursively. Suppose we naively design an algorithm that follow ID step-by-step and trains a generative model and sample from every factor it encounters. This algorithm would not know which of these factors to learn and sample first with the generative models. To sample $W_2 \sim P_{x,w_1}(w_2)$ we need $W_1$ as input, which has to be sampled from $P_{x,w_2}(w_1, y)$. But to sample $\{W_1, Y\} \sim P_{x,w_2}(w_1, y)$, we need $W_2$ as input which has to be sampled from $P_{x,w_1}(w_2)$. Therefore, no order helps to sample all $W_1, W_2, Y$ consistently.

ID-GEN follows ID's recursive trace to reach at the factorization: $P_x(y) = \sum_{w_1, w_2} P_{x,w_1}(w_2) P_{x,w_2}(w_1, y)$ but solves the deadlock issue by avoiding direct sampling from them. Rather, it first trains the required models for c-components $\{W_1, Y\}, \{W_2\}$ individually, considering all possible input values, and then connects them to perform sampling. Thus, ID-GEN first obtains $P_{x,w_2}(w_1, y) = P(w_1|x)P(y|x, w_1, w_2)$, and trains a conditional model for each of the conditional distributions (Fig. 8 left-blue). Similarly, for $P_{x,w_1}(w_2) = \sum_{x'} P(x')P(w_2|x', w_1)$, we train models for each conditional distribution (right-blue). Note that $X$ and $X'$ are sampled from the same $P(x)$ but considered as different variables. Thus, we train $M_{X'}$ to sample from $P(x')$ which is different from intervention $\text{do}(X)$. Finally, we merge these networks trained for each c-component to build a single sampling network and can perform ancestral sampling on this network to sample from $P_x(y), \forall x$.

## C.3   Related works

In Figure 9, we show a visual comparison among different implementations of existing works. Researchers have proposed deep neural networks for causal inference problems [43, 30] and neural causal methods Xia et al. [58], Kocaoglu et al. [26] to deal with high-dimensional data. However, the proposed approaches are not applicable to any general query due to the restrictions they employ or their algorithmic design. For example, some approaches perform well only for low-dimensional discrete and continuous data (group 2 in Figure 9: [58, 1]) while some propose modular training to partially deal with high-dimensional variables (group 3: [40]). Other works with better generative

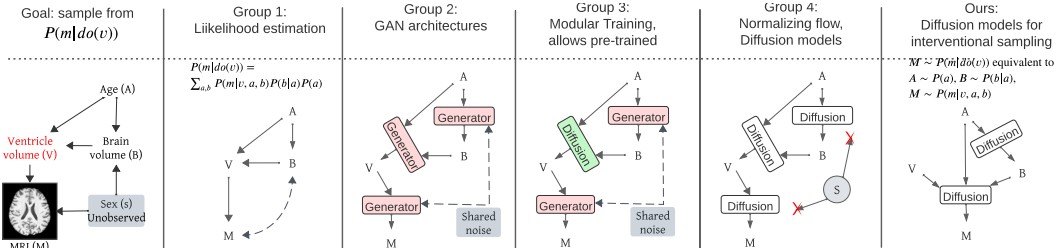

Figure 9: Suppose we aim to sample from $P(m|\text{do}(v))$, with $M$(MRI) as high-dimensional. Group 1 includes algorithms (ex: ID) that depend on likelihood estimation such as $P(m|v, a, b)$. Group 2 algorithms with GAN architectures have issues with GAN convergence while Group 3 improves convergence with modularization but both struggle to match the joint distribution. Group 4 utilizes normalizing flow, diffusion models, etc but cannot deal with confounders. Groups 2-4 only do unconditional interventions or costly rejection sampling. Finally, our method can employ classifier-free diffusion models to sample from $P(m|\text{do}(v))$ or conditional $P(m|a, \text{do}(v))$. The causal graph is adapted from Ribeiro et al. [41].

performance depend on a strong structural assumption: no latent variable in the causal graph (group 4: Kocaoglu et al. [26], Pawlowski et al. [35], Chao et al. [11]).

### C.4  ID-GEN recursion tree example

In Figure 10, we show a possible recursive route of ID-GEN for a causal query $P(\mathbf{y}|\text{do}(\mathbf{x}))$. At any recursion level, we check condition for 7 steps (S1-S7) and enter into one step based on the satisfied conditions. The red edges indicate the top-down phase, and the green edges indicate the bottom-up phase. The rectangular gray boxes (`ConditionalGMs(.)`, `MergeNetwork(.)`, `Update(.)`) represent the functions that allow ID-GEN to sample from the high-dimensional interventional distribution. $P_{\mathbf{v}\backslash s_1}(s_1)P_{\mathbf{v}\backslash s_2}(s_2)$ are obtained after performing c-factorization at step 4. We also indicate the recursive route with increasing indices. We hope that this figure helps the readers understand the recursion route in a better way.

### C.5  IDC-GEN: conditional interventional sampling

Existing works that use causal graph-based feedforward models [59, 40] need to update the posterior of $Z$'s upstream variables which is not efficiently feasible in their architecture. We, given a causal query $P_x(y|z)$, sample from this conditional interventional query by calling Algorithm 5: IDC-GEN. This function finds the maximal set $\alpha \subset Z$ such that we can apply do-calculus rule-2 and move $\alpha$ from conditioning set $Z$ and add it to intervention set $X$. Precisely, $P_x(y|z) = P_{x\cup\alpha}(y|z \setminus \alpha) = \frac{P_{x\cup\alpha}(y,z\backslash\alpha)}{P_{x\cup\alpha}(z\backslash\alpha)}$. Next, Algorithm 1: ID-GEN (.) is called to obtain the sampling network that can sample from the interventional joint distribution $P_{x\cup\alpha}(y, z \setminus \alpha)$. We use the sampling network to generate samples $\mathcal{D}[\mathbf{X} \cup \alpha, \mathbf{Y}, \mathbf{Z} \setminus \alpha]$. We train a new conditional model $M_Y$ on $\mathcal{D}$ that takes $Z \setminus \alpha, X \cup \alpha$ as input and samples $Y \sim P_{x\cup\alpha}(y, z \setminus \alpha)$ i.e, $Y \sim P_x(y|z)$.

### C.6  ID-GEN computational complexity

Bhattacharyya et al. [5] in their work discusses the sample complexity and time complexity of Shpitser and Pearl [46]'s ID algorithm. Since our algorithm ID-GEN is build upon the recursive structure of ID, we follow their approach to determine the computational complexity of our algorithm.

Suppose, in step 4 ID-GEN factorizes $P_{\mathbf{x}}(\mathbf{y})$ as

$$\sum_{\mathbf{v}\backslash(\mathbf{y}\cup\mathbf{x})}P_{\mathbf{v}\backslash s_1}(s_1)\dots P_{\mathbf{v}\backslash s_l}(s_l)\dots P_{\mathbf{v}\backslash s_n}(s_n)$$

where each $S_i$ is the c-factor of each c-component $C_i$. Let the number of variables located in each c-component be $k$. Suppose, the intervention $\mathbf{X}$ can be partitioned into multiple c-components and the c-components can be arranged in a way such that $\mathbf{X} = \cup_{i=1}^{l}\mathbf{X}_i$ and $\mathbf{X}_i \subseteq C_i$, i.e., all interventions are located in the first $l$ c-components. Following [5], we define two sets $\mathbf{C}_{>l} = \cup_{i>l}C_i$ and $\mathbf{C}_{\leq l} = \cup_{i\leq l}C_i$.

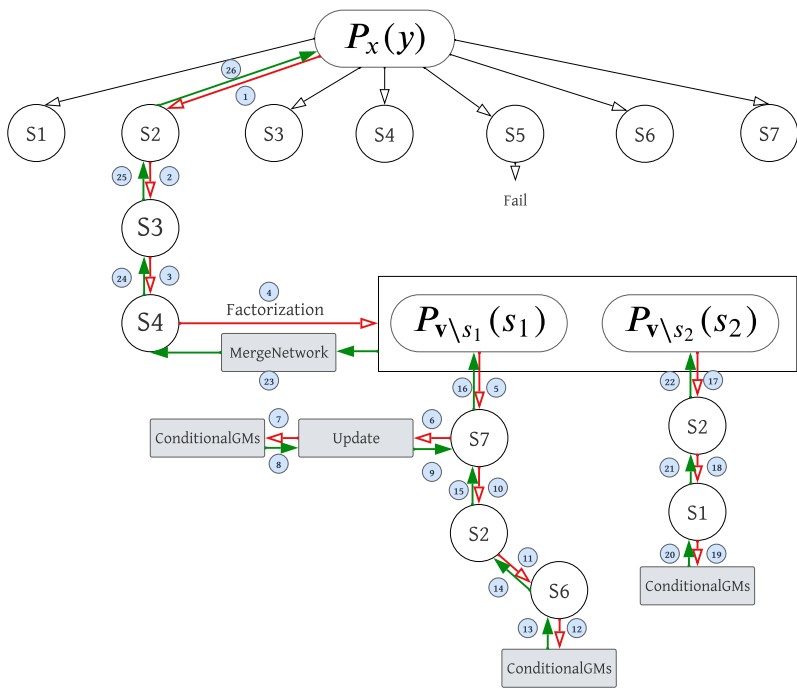

Figure 10: ID-GEN Recursion Tree Example

For $|\mathbf{C}_{>l}| = n - l$, c-components that do not contain any $\mathbf{X}$, ID-GEN will either go to step1 or step 6. In the base cases, they will train $k$ models for each c-components. Thus, the number of models trained for these c-components will be $\mathcal{O}((n-l)k)$.

For the first $|\mathbf{C}_{\leq l}| = l$, c-components, we assume that each c-component contains intervention subset $\mathbf{X}_i$ of size $\mathcal{X}$. Then the size of the remaining c-component is $k - \mathcal{X}$. For each $P_{\mathbf{v}\backslash s_i}(s_i)$, ID-GEN will eventually reach the base cases: Step 1 or 6 and will train $k - \mathcal{X}$ models. Now, consider recursive steps. We assume that, in the worst case, $\mathbf{X}_i$ will reduce one at a time, by visiting step 7 and step 2 alternately. Thus, ID-GEN will visit step 7, $\mathcal{O}((\mathcal{X}/2 - 1))$ times (except base case). Whenever ID-GEN visits step 7, it will apply a subset of the intervention $\mathbf{X}_i$ and atmost $k$ models will be trained to sample the updated training dataset. Thus, till the base case, the total number of models trained in step 7s will be $\mathcal{O}(k(\mathcal{X}/2 - 1))$. If we consider the whole recursive route for each c-component $C_i \in \mathbf{C}_{\leq l}$, we will train $\mathcal{O}(k(\mathcal{X}/2 - 1)) + \mathcal{O}((k - \mathcal{X})) = \mathcal{O}(k\mathcal{X}/2)$ number of models. For all c-components in $\mathbf{C}_{\leq l}$, we will train $\mathcal{O}(lk\mathcal{X}/2)$ number of models.

Finally, if we consider all c-components, we will train in total $\mathcal{O}((n-l)k + lk\mathcal{X}/2)$ models. If the cost of training a diffusion model to learn a specific conditional distribution is $\mathcal{O}(T)$, then the total training cost is $\mathcal{O}(T(n-l)k + Tlk\mathcal{X}/2)$. Note that, when there exists no confounders, we have $k = 1$ and the trining cost is $\mathcal{O}(T(n-l) + Tl) = \mathcal{O}(Tn)$. Here $|An(\mathbf{Y})_G| = n$. Existing algorithms train $|\mathbf{V}|$ number of models for a causal graph of $\mathbf{V}$ variables. Thus, their training cost is $\mathcal{O}(T|\mathbf{V}|)$.

## C.7 ID-GEN for Non-Markovian Causal Models

We assumed the underlying true causal model to be semi-Markovian in our paper. This is not a limitation as [52] showed that causal models with arbitrary latent variables can always be converted into semi-Markovian causal models i.e., models in which latent variables have no parent and only two children, while preserving the same independence relations between the observed variables.

# D Theoretical analysis

Here we provide formal proofs for all theoretical claims made in the main paper, along with accompanying definitions and lemmas.

**Definition D.1.** A **conditional generative model** for a random variable $X \in \mathbf{V}$ relative to the distribution $P(\mathbf{v})$ is a function $M_X : \mathcal{P}a_X \rightarrow |X|$ such that $M_X(pa_X) \sim P(x|pa_X), \forall pa \in \mathcal{P}a$, where $\mathcal{P}a$ is a subset of observed variables in $V$.

**Definition D.2** (Recursive call). For a function $f()$, when a subprocedure with the same name is called within $f()$ itself, we define it as a recursive call. At Steps 2, 3, 4, 7 of Algorithm 1:ID-GEN, the sub-procdedure ID-GEN(.) with updated parameters are recursive calls, but the sub-procedure `ConditionalGMs(.)`, `Update(.)` are not recursive calls.

Here, we restate the assumptions that are mentioned in the main paper.

**Assumption D.3.** The causal model is semi-Markovian.

**Assumption D.4.** We have access to the true acyclic-directed mixed graph (ADMG) induced by the causal model.

**Assumption D.5.** Each conditional generative model trained by ID-GEN correctly samples from the corresponding conditional distribution.

**Assumption D.6.** The observational joint distribution is strictly positive and Markov relative to the causal graph.

**Lemma D.7** (c-component factorization [51]). *Let $\mathcal{M}$ be an SCM that entails the causal graph $G$ and $P_x(y)$ be the interventional distribution for arbitrary variables $X$ and $Y$. Let $C(G \setminus X) = \{S_1, \ldots, S_n\}$. Then we have $P_x(y) = \sum\limits_{v \setminus (y \cup x)} P_{v \setminus s_1}(s_1) P_{v \setminus s_2}(s_2) \ldots P_{v \setminus s_n}(s_n)$.*

**Definition D.8.** We say that a sampling network $\mathcal{H}$ is valid for an interventional distribution $P_\mathbf{x}(\mathbf{y})$ if the following conditions hold:

- Every node $V \in \mathcal{H}$ has an associated conditional generative model $M_V(.)$ except the variables in $\mathbf{X}$.

- If the values $\mathbf{X} = \mathbf{x}$ are specified in $\mathcal{H}$, then the samples of $\mathbf{Y}$ obtained after dropping $\mathcal{H} \setminus \{\mathbf{X} \cup \mathbf{Y}\}$ from all generated samples are equivalent to samples from $P_\mathbf{x}(\mathbf{Y})$.

**Proposition D.9.** *The `ConditionalGMs(.)` $(S', \mathbf{X}_Z, G, \mathcal{D}, \hat{\mathbf{X}}, \hat{G})$ called inside `Update(.)`, returns $\mathcal{D}[\mathbf{X}_Z, S'] \sim P_{\mathbf{X}_Z}(S')$.*

*Proof.* Suppose that our goal is to generate samples from $P_{\mathbf{X}_Z}(\mathbf{Y} = S')$. $S'$ is a single c-component and the intervention set $\mathbf{X} = \mathbf{X}_Z$ located outside of $S'$. This is the entering condition for step 6 of the ID-GEN algorithm. Thus, we can directly apply it as Algorithm 2: `ConditionalGMs(.)`, instead of another recursive ID-GEN $(\mathbf{Y} = S', \mathbf{X} = \mathbf{X}_Z, G, \mathcal{D}, \hat{\mathbf{X}}, \hat{G})$ call. $\square$

**Proposition D.10.** *At any recursion level of Algorithm 1: ID-GEN $(\mathbf{Y}, \mathbf{X}, G, \mathcal{D}, \hat{\mathbf{X}}, \hat{G})$ and Algorithm 6: ID$(\mathbf{Y}, \mathbf{X}, P, G)$ , the execution step is determined by the values of the set of observed variables $\mathbf{Y}$, the set of intervened variables $\mathbf{X}$ and the causal graph $G$, at that recursion level.*

*Proof.* To enter in steps [1-7] of ID-GEN and steps [1-7] of ID, specific graphical conditions are checked, which depend only on the values of the parameters $\mathbf{Y}, \mathbf{X}$ and $G$. If that graphical condition is satisfied, both algorithms enter in their corresponding steps. Thus, the execution step of each algorithm at any recursive level is determined by only $\mathbf{Y}, \mathbf{X}$ and $G$. $\square$

**Lemma D.11.** *(Recursive trace)* *At any recursion level $R$, recursive call ID-GEN $(\mathbf{Y}, \mathbf{X}, \mathcal{D}, G, \hat{\mathbf{X}}, \hat{G})$ enters Step $i$ if and only if recursive call ID$(\mathbf{Y}, \mathbf{X}, P, G)$ enters Step $i$, for any $i \in [7]$*

*Proof.* Suppose, both algorithms start with an input causal query $P_\mathbf{x}(\mathbf{y})$ for a given graph $G$. For this query, the parameters $\mathbf{Y}, \mathbf{X}, G$ of ID$(\mathbf{Y}, \mathbf{X}, P, G)$ and ID-GEN $(\mathbf{Y}, \mathbf{X}, \hat{X}', \mathcal{D}, G)$ represent the same objects: the set of observed variables $\mathbf{Y}$, the set of intervened variables $\mathbf{X}$ and the causal graph $G$. According to Proposition D.10, Algorithm 1: ID-GEN and Algorithm 6: ID only check these 3

parameters to enter into any steps and decide the next recursive step. Thus, we use proof by induction based on these 3 parameters to prove our statement.

**Induction base case (recursion level $R = 0$):** Both algorithms start with the same input causal query $P_\mathbf{x}(\mathbf{y})$ in $G$, i.e., the same parameter set $\mathbf{Y}, \mathbf{X}, G$. We show that at recursion level $R = 0$, ID-GEN enters step $i$ if and only if ID enters step $i$ for any $i \in [7]$.

**Step 1:** Both ID and ID-GEN check if the intervention set $\mathbf{X}$ is empty and go to step 1. Thus, ID-GEN enters step 1 iff ID enters step 1.

**Step 2:** If the condition for step 1 is not satisfied, then both ID and ID-GEN check if there exist any non-ancestor variables of $\mathbf{Y}$: $\mathbf{V} \setminus An(\mathbf{Y})_G$ in the graph to enter in their corresponding step 2. Thus, with the same $\mathbf{Y}$ and $G$, ID-GEN enters step 2 iff ID enters step 2.

**Step 3:** If conditions for Steps [1-2] are not satisfied, both ID and ID-GEN check if $\mathbf{W}$ is an empty set for $\mathbf{W} = (\mathbf{V} \setminus \mathbf{X}) \setminus An(\mathbf{Y})_{G_{\overline{\mathbf{X}}}}$ to enter in their corresponding step 3. Since both algorithms have the same $\mathbf{Y}, \mathbf{X}$ and $G$, ID-GEN enters step 3 iff ID enters step 3.

**Step 4:** If conditions for Steps [1-3] are not satisfied, both ID and ID-GEN check if they can partition the variables set $C(G \setminus \mathbf{X})$ into $k$ (multiple) c-components. Since these condition checks depend on $\mathbf{X}$ and $G$ and both have the same input $\mathbf{X}$ and $G$, ID-GEN enters step 4 iff ID enters step 4.

**Step 5:** If conditions for Steps [1-4] are not satisfied, both algorithms check if $C(G \setminus \mathbf{X}) = \{S\}$ and $C(G) = \{G\}$ to enter their corresponding step 5. Since both have the same $\mathbf{X}$ and $G$, ID-GEN enters step 5 iff ID enters step 5.

**Step 6:** If the conditions of Steps [1-5] are not satisfied, ID and ID-GEN check if $C(G \setminus \mathbf{X}) = \{S\}$ and $S \in C(G)$, to enter their corresponding step 6. Since both have the same $\mathbf{X}$ and $G$, ID-GEN enters step 6 iff ID enters step 6.

**Step 7:** If conditions for Steps [1-6] are not satisfied, both ID and ID-GEN check if $C(G \setminus \mathbf{X}) = \{S\}$ and $(\exists S')$ such that $S \subset S' \in C(G)$ to enter their corresponding step 7. Since both have the same $\mathbf{X}$ and $G$, both will satisfy this condition and enter this step. Thus, ID-GEN enters step 7 iff ID enters step 7.

**Induction Hypothesis:** We assume that for recursion levels $R = 1, \ldots, r$, ID-GEN and ID follow the same steps at each recursion level and maintain the same values for the parameter set $\mathbf{Y}, \mathbf{X}$ and $G$.

**Inductive steps:** Both algorithms have the same set of parameters $\mathbf{Y}, \mathbf{X}, G$ and they are at the same step $i$ at the current recursion level $R = r$. As inductive step, we show that with the same values of the parameter set $\mathbf{Y}, \mathbf{X}$ and $G$ at recursion level $R = r$, ID-GEN and ID will visit the same step at recursion level $R = r + 1$.

**Step 1**: Step 1 is a base case of both algorithms. After entering into step 1 with $\mathbf{X} = \emptyset$, ID estimates $\sum_{\mathbf{v} \setminus y} P(\mathbf{v})$ and ends the recursion. ID-GEN also ends the recursion after training a set of conditional models $M_{V_i}(V_\pi^{(i-1)}) \sim P(v_i | v_\pi^{(i-1)})$. Thus, ID-GEN goes to the same step at recursion level $R = r + 1$ iff ID goes to the same step which in this case is `Return`.

**Step 2**: At step 2, both ID-GEN and ID update their intervention set $\mathbf{X}$ as $\mathbf{X} \cap An(\mathbf{Y})_G$ and the causal graph $G$ as $G_{An(\mathbf{Y})}$. The same values of the parameters $\mathbf{X}, \mathbf{Y}$ and $G$ will lead both algorithms to the same step at the recursion level $R = r + 1$. Thus, at recursion level $R = r + 1$, ID-GEN visits step $i$ iff ID visits step $i, \forall i \in [7]$.

**Step 3**: At step 3, both algorithms only update the intervention set parameter $\mathbf{X}$ as $\mathbf{X} = \mathbf{X} \cup \mathbf{W}$ and perform the next recursive call. Since they leave for the next recursive call with the same set of parameters, at recursion level $R = r + 1$, ID-GEN visits step $i$ iff ID visits step $i, \forall i \in [7]$.

**Step 4**: At step 4, both ID-GEN and ID partition the variables set $C(G \setminus \mathbf{X})$ into $k$ (multiple) c-components. Then they perform a recursive call for each c-component with parameter $\mathbf{Y} = S_i$ and $\mathbf{X} = \mathbf{V} \setminus S_i$. For each of these $k$ recursive calls, both algorithms use the same values for the parameter set $\mathbf{Y}, \mathbf{X}$ and $G$. Thus, at recursion level $R = r + 1$, ID-GEN visits step $i$ iff ID visits step $i, \forall i \in [7]$.

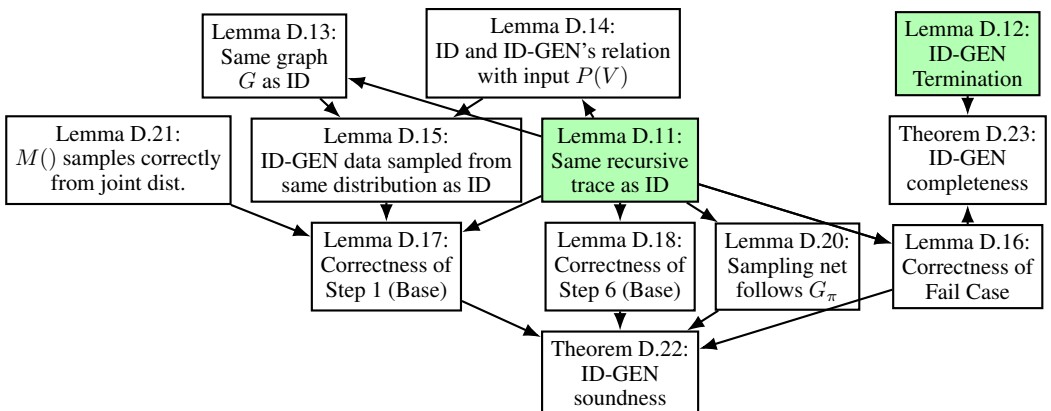

Figure 11: Flow Chart of Proofs

**Step 5**: Step 5 is a base case for both algorithms, and if both algorithms are at step 5, they return `FAIL`. This step represents the non-identifiability case. Thus, ID-GEN goes to the same step at recursion level $R = r + 1$ iff ID goes to the same step, which in this case is `Return`.

**Step 6**: Step 6 is a base case for both algorithms. ID calculates the product of a set of conditional distributions $P(v_i | v_\pi^{i-1})$ and returns it. While ID-GEN trains conditional models $M_{V_i}(V_\pi^{(i-1)})$ to learn those conditional distributions, ID-GEN returns a sampling network after connecting these models. Both algorithms end the recursion here and return different objects but that will not affect their future trace. Thus, ID-GEN goes to the same step at recursion level $R = r + 1$ iff ID goes to the same step which in this case is `Return`.

**Step 7**: ID algorithm at step 7, updates its distribution parameter $P$ with $P_{\mathbf{x_z}}(S')$. On the other hand, ID-GEN generates $do(\mathbf{X_Z})$ interventional samples. Both algorithms update their parameters set $\mathbf{Y}, \mathbf{X}$ and $G$ in the same way as $\mathbf{Y} = \mathbf{Y}, \mathbf{X} = \mathbf{X} \cap S'$ (or $\mathbf{X} \setminus \mathbf{X}_Z$ since $\mathbf{X} \cap S' = \mathbf{X} \setminus \mathbf{X}_Z$) and $G = G_{S'}$. Since they leave for the next recursive call with the same set of parameters $\mathbf{Y}, \mathbf{X}, G$, at recursion level $R = r + 1$, ID-GEN visits step $i$ iff ID visits step $i$, $\forall i \in [7]$.

Therefore, we have proved by induction that ID-GEN $(\mathbf{Y}, \mathbf{X}, G, \mathcal{D}, \hat{\mathbf{X}}, \hat{G})$ enters Step $i$ if and only if ID$(\mathbf{Y}, \mathbf{X}, P, G)$ enters Step $i$, for any $i \in [7]$ for any recursion level $R$.

$\square$

**Lemma D.12.** *Termination: Let $P_{\mathbf{x}}(\mathbf{Y})$ be a query for causal graph $G = (\mathbf{V}, E)$ and $\mathcal{D} \sim P(\mathbf{V})$. Then the recursions induced by ID-GEN $(\mathbf{Y}, \mathbf{X}, G, \mathcal{D}, \hat{\mathbf{X}}, \hat{G})$ terminate in either step 1, 5, or 6.*

*Proof.* The result follows directly from Lemma D.11: Consider any causal query $P_{\mathbf{x}}(\mathbf{Y})$. Since ID is complete, the query – whether it is identifiable or not – terminates in one of the steps 1, 5, or 6. By Lemma D.11, ID-GEN follows the same steps as ID, and hence also terminates in one of these steps. $\square$

**Lemma D.13.** *Consider the recursive calls $ID(*, G_{ID})$ and $ID\text{-}GEN(*, G_{ID\text{-}GEN}, \hat{\mathbf{X}}, \hat{G})$ at any level of the recursion. Then $G_{ID} = G_{ID\text{-}GEN}$ and $G_{ID} = \hat{G} \setminus \hat{\mathbf{X}}$.*

*Proof.* According to Lemma D.11, ID and ID-GEN follow the same recursive trace. Thus, at any recursion level, both algorithms will stay at step $i \in [7]$.

**Base case:** At recursion level $R = 0$, both ID and ID-GEN start with the same input causal graph $G$. Thus, $G_{ID} = G_{ID\text{-}GEN} = \hat{G} = G$. Also at $R = 0$, the set of intervened variables $\hat{\mathbf{X}} = \emptyset$. Thus, $G_{ID} = \hat{G} \setminus \emptyset = G$ holds.

**Induction hypothesis:** At any recursion level $R = r$, let $G_{ID}^r$ be the graph parameter of the ID algorithm and $\{G_{ID\text{-}GEN}^r, \hat{\mathbf{X}}^r, \hat{G}^r\}$ be the parameters for the ID-GEN algorithm. We assume that $G_{ID}^r = G_{ID\text{-}GEN}^r$ and $G_{ID}^r = \hat{G}^r \setminus \hat{\mathbf{X}}^r$.

**Inductive step:** The set of parameters $\{G_{\text{ID-GEN}}, \hat{\mathbf{X}}, \hat{G}\}$ of ID-GEN and the graph parameter $G$ of ID are only updated at step 2 and step 7 for the next recursive calls. Thus, we prove the claim for these two steps separately.

**At step 2**: In both ID and ID-GEN algorithms, we remove the non-ancestor variables $\mathbf{V} \setminus An(Y)$ from the graph. For ID, $G_{ID}^{r+1} = G_{ID}^{r}(An(\mathbf{Y}))$. For the ID-GEN algorithm, we obtain $G_{\text{ID-GEN}}^{r+1} = G_{\text{ID-GEN}}^{r}(An(\mathbf{Y}))$. Thus, $G_{ID}^{r+1} = G_{\text{ID-GEN}}^{r+1}$. In ID-GEN, we also update $\hat{G}^{r+1}$ as $\hat{G}^{r+1} = \hat{G}^{r}(An(\mathbf{Y}))$. Since removing non-ancestor does not affect the intervened variables $\hat{\mathbf{X}}$ anyway, the same relation between $G_{ID}^{r+1}$ and $\hat{G}^{r+1}$ is maintained, i.e., $G_{ID}^{r+1} = \hat{G}^{r+1} \setminus \hat{\mathbf{X}}$.

**At step 7**: Both ID and ID-GEN finds $\mathbf{X}_Z = \mathbf{X} \setminus S'$ and apply $do(\mathbf{X}_Z)$ as intervention. Also, in ID-GEN, $\hat{\mathbf{X}}^{r+1}$ is set to $\hat{\mathbf{X}}^{r} \cup \mathbf{X}_Z$.

*For ID-GEN:*

Since $\mathbf{X}_Z$ is intervened on, $G^r$ for ID-GEN is updated as $G^{r+1} = G_{\overline{\mathbf{X}_Z}}^{r}$ and $\hat{G}^r$ is updated as $\hat{G}^{r+1} = \hat{G}_{\overline{\mathbf{X}_Z}}^{r}$.

Now, $\mathbf{X}_Z$ has all incoming edges are cut off in $G^{r+1}$ and does not have any parents. As a result, removing $\mathbf{X}_Z$ from $G^{r+1}$ is valid. Thus, we obtain $G^{r+1} = G^{r+1} \setminus \mathbf{X}_Z = G_{\overline{\mathbf{X}_Z}}^{r} \setminus \mathbf{X}_Z = G_{S'}^{r}$. However, for the other graph parameter, we keep $\hat{G}^{r+1} = \hat{G}_{\overline{\mathbf{X}_Z}}^{r}$ as it is, since we would follow this structure in the base cases and consider $\mathbf{X}_Z$ while training the conditional models. Note that, in $\hat{G}^r$ the intervened variables $\hat{\mathbf{X}}^r$ (before recursion level $r$) had already incoming edges removed. Thus, after $\hat{\mathbf{X}}^r$ being updated as $\hat{\mathbf{X}}^{r+1} = \hat{\mathbf{X}}^{r} \cup \mathbf{X}_Z$, we can write $\hat{G}^{r+1} = \hat{G}_{\overline{\mathbf{X}_Z}}^{r} = \hat{G}_{\overline{\hat{\mathbf{X}}^{r+1}}}^{r}$. Since by inductive assumption, $G^r = \hat{G}^r \setminus \hat{\mathbf{X}}^r$, thus

$$G^{r+1} = \hat{G}^{r+1} \setminus \{\hat{\mathbf{X}}^r \cup \mathbf{X}_Z\} = \hat{G}^{r+1} \setminus \{\hat{\mathbf{X}}^{r+1}\} \tag{2}$$

*For ID:*

ID algorithm updates its parameters $G$ as $G^{r+1} = G_{S'}^{r} = G^r \setminus \mathbf{X}_Z$. Since ID-GEN and ID have the same graph parameter at recursion level $r$, i.e, $G_{ID}^{r} = G_{\text{ID-GEN}}^{r}$ and they updated the parameter in the same way, $G_{ID}^{r+1} = G_{\text{ID-GEN}}^{r+1}$ holds true. Since $G_{\text{ID-GEN}}^{r+1} = \hat{G}^{r+1} \setminus \{\hat{\mathbf{X}}^{r+1}\}$ according to Equation 2, we also obtain $G_{ID}^{r+1} = \hat{G}^{r+1} \setminus \{\hat{\mathbf{X}}^{r+1}\}$.

Therefore, the lemma holds for any recursion level $R$. $\qquad\square$

**Lemma D.14.** *Let $P(.)$ be the input observational distribution to the ID algorithm and $\mathcal{D} \sim P(.)$ be the input observational dataset to the ID-GEN algorithm. Suppose, $\hat{\mathbf{X}}$ is the set of variables that are intervened at step 7 of both the ID and ID-GEN algorithm from recursion level $R = 0$ to $R = r$. Consider recursive calls $ID(*, P_{ID}(\mathbf{v}))$ and ID-GEN$(*, \mathcal{D}, *, \hat{\mathbf{X}}, \hat{G})$ at recursion level $R = r$. Then $P_{ID} = P_{\hat{\mathbf{x}}}(\mathbf{v})$ and $\mathcal{D}[\hat{\mathbf{X}}, \mathbf{V}] \sim P_{\hat{\mathbf{x}}}(\hat{\mathbf{x}}, \mathbf{v})$.*

*Proof.* According to Lemma D.11, ID and ID-GEN follow the same recursive trace. Thus, at any recursion level, both algorithms will stay at step $i \in [7]$.

**Base case:** At recursion level $R = 0$, ID starts with the observational distribution $P$ and ID-GEN starts with the observational training dataset $\mathcal{D} \sim P$. At $R = 0$, $\hat{\mathbf{X}} = \emptyset$ since no variables have yet been intervened on. Thus, for ID algorithm $P_{ID} = P_{\emptyset}(V)$ holds. For ID-GEN, $\mathcal{D}[\emptyset, \mathbf{V}] \sim P_{\emptyset}(\emptyset, \mathbf{V})$ holds. Thus, the claim is true for $R = 0$.

**Induction hypothesis:** Let the set of variables intervened at step 7s from the recursion level $R = 0$ to $R = r$, be $\hat{\mathbf{X}}^r$. At $R = r$, let the distribution parameter of the ID algorithm be $P_{ID}^r = P_{\hat{\mathbf{x}}^r}(\mathbf{v})$. Let the dataset parameter of the ID-GEN algorithm be sampled from $P_{\hat{\mathbf{x}}^r}(\hat{\mathbf{x}}^r, \mathbf{v})$, i.e, $\mathcal{D}^r \sim P_{\hat{\mathbf{x}}^r}(\hat{\mathbf{x}}^r, \mathbf{v})$.

**Inductive step:** The distribution parameter $P$ of ID and the dataset parameter $\mathcal{D}$ of ID-GEN only change at step 2 and step 7 for the next recursive calls. Thus, we prove the claim for these two steps separately.

**At step 2**: At step 2 of both ID and ID-GEN algorithms, there is no change in $\hat{\mathbf{X}}$, i.e., no additional variables are intervened to change the distribution $P$ or the dataset $\mathcal{D}$.

At this step, we marginalize over $\mathbf{V} \setminus An(\mathbf{Y})$. Since $\hat{\mathbf{X}} \subset An(\mathbf{Y})_{\hat{G}}$, marginalizing non-ancestors out does not impact $\hat{\mathbf{X}}$. Thus, $P_{ID}^{r+1}(An(\mathbf{Y})) = \sum_{\mathbf{V} \setminus An(\mathbf{Y})_G} P_{ID}^r(\mathbf{v}) = \sum_{\mathbf{V} \setminus An(\mathbf{Y})_G} P_{\hat{\mathbf{x}}^r}(\mathbf{v}) = P_{\hat{\mathbf{x}}^r}(An(\mathbf{Y}))$. This holds true since we can marginalize over $\mathbf{V} \setminus An(\mathbf{Y})_G$ in the joint interventional distribution.

In the ID-GEN algorithm, we drop the values of variables $\mathbf{V} \setminus An(\mathbf{Y})_G$ from the data set, that is, $\mathcal{D}^r[\hat{\mathbf{X}}, \mathbf{V}] \rightarrow \mathcal{D}^{r+1}[\hat{\mathbf{X}}, An(\mathbf{Y})_G]$. Since we assumed $\mathcal{D}^r[\hat{\mathbf{X}}, \mathbf{V}] \sim P_{\hat{\mathbf{x}}^r}(\hat{\mathbf{x}}^r, \mathbf{v})$, therefore $\mathcal{D}^{r+1}[\hat{\mathbf{X}}, An(\mathbf{Y})_G] \sim P_{\hat{\mathbf{x}}^r}(An(\mathbf{Y}))$ holds.

**At step 7**: At step 7, both ID and ID-GEN algorithm intervened on variable set $\mathbf{X}_Z$ where $\mathbf{X}_Z = \mathbf{X} \setminus S'$. Thus, at level $R = r + 1$, we have $\hat{\mathbf{X}}^{r+1} = \hat{\mathbf{X}}^r \cup \mathbf{X}_Z$. Note that at this step, $S' = \mathbf{V} \setminus \mathbf{X}_Z$.

ID algorithm updates its current distribution parameter $P_{ID}^r$ by intervening on $\mathbf{X}_Z$. Thus, at level $R = r + 1$, $P_{ID}^{r+1}(\mathbf{v} \setminus \mathbf{x}_Z) = P_{ID}^{r+1}(s') = P_{ID}^r(\mathbf{v}|\mathrm{do}(\mathbf{x}_Z)) = P_{\hat{\mathbf{x}}^r}(\mathbf{v}|\mathrm{do}(\mathbf{x}_Z)) = P_{\hat{\mathbf{x}}^r \cup \mathbf{x}_Z}(s') = P_{\hat{\mathbf{x}}^{r+1}}(s')$. Thus the claim holds for ID algorithm at level $R = r + 1$.

In the ID-GEN algorithm, we generate an interventional dataset $\mathcal{D}^{r+1}[\mathbf{X}_Z, \hat{\mathbf{X}}, S']$ by applying the $\mathrm{do}(\mathbf{X}_Z)$ intervention on dataset $\mathcal{D}^r[\hat{\mathbf{X}}, \mathbf{V}]$. Since $\mathcal{D}^r[\hat{\mathbf{X}}^r, \mathbf{V}] \sim P_{\hat{\mathbf{x}}^r}(\hat{\mathbf{x}}^r, \mathbf{v})$, we obtain $\mathcal{D}^{r+1}[\mathbf{X}_Z, \hat{\mathbf{X}}^r, S'] \sim P_{\hat{\mathbf{x}}^r \cup \mathbf{x}_z}(\hat{\mathbf{x}}^r, \mathbf{x}_z, s') \implies \mathcal{D}^{r+1}[\hat{\mathbf{X}}^{r+1}, S'] \sim P_{\hat{\mathbf{x}}^{r+1}}(\hat{\mathbf{x}}^{r+1}, s')$. Thus, the claim holds true for ID-GEN at the recursion level $R = r + 1$.

Therefore, $P(.)$ being the input distribution for ID and $\mathcal{D} \sim P(.)$ being the input dataset for ID-GEN, we proved by induction that $P_{ID} = P_{\hat{\mathbf{x}}}(\mathbf{v})$ and $\mathcal{D}[\hat{\mathbf{X}}, \mathbf{V}] \sim P_{\hat{\mathbf{x}}}(\hat{\mathbf{x}}, \mathbf{v})$ at any recursion level.

$\square$

**Lemma D.15.** *Consider recursive calls $ID(\mathbf{Y}, \mathbf{X}, G, P_{ID}(\mathbf{v}))$ and ID-GEN $(\mathbf{Y}, \mathbf{X}, G, \mathcal{D}, \hat{\mathbf{X}}, \hat{G})$ at any recursion level R. If $\mathcal{D}[\hat{\mathbf{X}}, V] \sim P_{ID\text{-}GEN}(\hat{\mathbf{x}}, \mathbf{v})$, then $P_{ID}(\mathbf{v}) = P_{ID\text{-}GEN}(\mathbf{v}|\hat{\mathbf{x}})$ for fixed values of $\hat{\mathbf{X}} = \hat{\mathbf{x}}$.*

*Proof.* At any level $R = r$ of the recursion , we have $G_{ID} = G_{\text{ID-GEN}}$ and $G_{ID} = \hat{G} \setminus \hat{\mathbf{X}}$ according to Lemma D.13 and $P_{ID} = P_{\hat{\mathbf{x}}}(\mathbf{v})$ and $\mathcal{D}[\hat{\mathbf{X}}, \mathbf{V}] \sim P_{\hat{\mathbf{x}}}(\hat{\mathbf{x}}, \mathbf{v})$ according to Lemma D.14 where $P(\mathbf{v})$ is the input observational distribution.

Let $\mathcal{D}[\hat{\mathbf{X}}, \mathbf{V}] \sim P_{\hat{\mathbf{x}}}(\hat{\mathbf{x}}, \mathbf{v}) = P_{\text{ID-GEN}}(\hat{\mathbf{x}}, \mathbf{v}), \forall \hat{\mathbf{x}}$.

Now, since in $\hat{G}$ we have already intervened on $\hat{\mathbf{X}}$, for fixed $\hat{\mathbf{X}} = \hat{\mathbf{x}}$, we have $P_{\hat{\mathbf{x}}}(\hat{\mathbf{x}}, \mathbf{v}) = P_{\hat{\mathbf{x}}}(\hat{\mathbf{x}}) * P_{\hat{\mathbf{x}}}(\mathbf{v}|\hat{\mathbf{x}}) = P_{\hat{\mathbf{x}}}(\mathbf{v}|\hat{\mathbf{x}})$, thus $\mathbf{V}$ will be sampled from $P_{\text{ID-GEN}}(\mathbf{v}|\hat{\mathbf{x}})$ for fixed consistent $\hat{\mathbf{X}} = \hat{\mathbf{x}}$.

On the other hand, in the ID algorithm, at any recursion level, $P_{ID}(\mathbf{v}) = P_{\hat{\mathbf{x}}}(\mathbf{v})$ for fixed consistent $\hat{\mathbf{X}} = \hat{\mathbf{x}}$ and $G_{ID} = \hat{G} \setminus \hat{\mathbf{X}}$.

Since according to Lemma D.14, $P_{ID} = P_{\hat{\mathbf{x}}}(\mathbf{v})$ and $\mathcal{D}[\hat{\mathbf{X}}, \mathbf{V}] \sim P_{\hat{\mathbf{x}}}(\hat{\mathbf{x}}, \mathbf{v})$ at any recursion level, i.e., starting from the same $P(\mathbf{V})$, the same set of interventions are performed in the same manner in both algorithms; thus for fixed $\hat{\mathbf{X}} = \hat{\mathbf{x}}$, we obtain $P_{ID}(\mathbf{v}) = P_{\text{ID-GEN}}(\mathbf{v}|\hat{\mathbf{x}})$ .

$\square$

**Lemma D.16.** *If a non-identifiable query is passed to it, ID-GEN will return* `FAIL`.

*Proof.* Suppose ID-GEN is given a non-identifiable query as input. Since the ID algorithm is complete, if ID is given this query, it will reach its step 5 and return `FAIL`. According to the lemma D.11, ID-GEN follows the same trace and the same sequence of steps as ID for a fixed input. Since for the non-identifiable query ID reaches step 5, ID-GEN will also reach step 5 and return `FAIL`. $\square$

**Lemma D.17.** *ID-GEN Base case (step 1):*

*Let, at any recursion level of ID-GEN, the input dataset $\mathcal{D}[\hat{\mathbf{X}}, \mathbf{V}]$ is sampled from a specific joint distribution $P(\hat{\mathbf{X}}, \mathbf{V})$, i.e., $\mathcal{D}[\hat{\mathbf{X}}, \mathbf{V}] \sim P(\hat{\mathbf{X}}, \mathbf{V})$ where $\hat{\mathbf{X}}$ is the set of intervened variables at step 7s. Given a target interventional query $P(\mathbf{y})$ over a causal graph $G = (V, E)$, suppose ID-GEN*

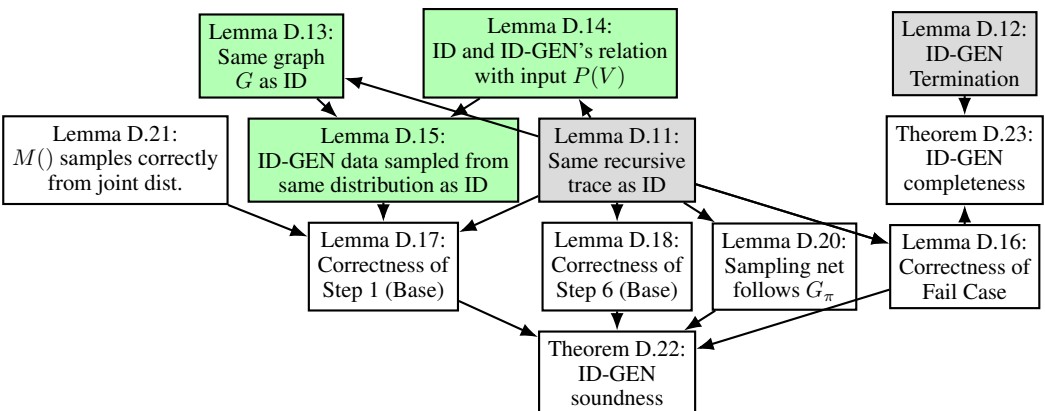

Figure 12: Flow Chart of Proofs

$(\mathbf{Y}, \mathbf{X} = \emptyset, G, \mathcal{D}, \hat{\mathbf{X}}, \hat{G})$ *enters step 1 and returns* $\mathcal{H}$. *Then* $\mathcal{H}$ *is a valid sampling network for* $P(\mathbf{y})$ *with fixed consistent* $\hat{\mathbf{X}} = \hat{\mathbf{x}}$.

*Proof.* Since the intervention set $\mathbf{X}$ is empty, we are at the base case step 1 of ID-GEN. Suppose that for the same query, the ID algorithm reaches step 1. Let $P_{ID}(\mathbf{v})$ be the current distribution of the ID algorithm after performing a series of marginalizations in step 2 and intervention in step 7 on the input observational distribution. $\mathcal{D}[\hat{\mathbf{X}}, \mathbf{V}]$ is the dataset parameter of ID-GEN algorithm that went through the same transformations as the ID algorithm in the sample space according to Lemma D.15. $\hat{\mathbf{X}}$ is the set of interventions that are applied on the dataset at step 7s.

Since ID algorithm is sound it returns correct output for the query $P(\mathbf{y})$. We prove the soundness of ID-GEN step 1 by showing that the sampling network ID-GEN returns, is valid for the output of the ID algorithm.

ID algorithm is sound and factorizes $P_{ID}(\mathbf{v})$ with respect to the graph $G$ as below.

$$P_{ID}(\mathbf{v}) = \prod_{v_i \in V} P(v_i | v_{\pi_G}^{(i-1)})$$

And obtains $P_{ID}(\mathbf{y})$ by:

$$P_{ID}(\mathbf{y}) = \sum_{v \backslash y} \prod_{v_i \in V} P(v_i | v_{\pi_G}^{(i-1)}) \tag{3}$$

Let $\mathcal{D}[\hat{\mathbf{X}}, \mathbf{V}] \sim P_{\text{ID-GEN}}(\hat{\mathbf{x}}, \mathbf{v}) = P(\hat{\mathbf{x}}, \mathbf{v}), \forall(\hat{\mathbf{x}}, \mathbf{v})$. For fixed $\hat{\mathbf{X}} = \hat{\mathbf{x}}$, $P_{\text{ID-GEN}}(\mathbf{v}|\hat{\mathbf{x}})$ can be factorized with respect to $G$ and $\hat{G}$ as following:

$$P_{\text{ID-GEN}}(\mathbf{v}|\hat{\mathbf{x}}) = \prod_{v_i \in V} P(v_i | v_{\pi_G}^{(i-1)}, \hat{\mathbf{x}})$$
$$= \prod_{v_i \in V} P(v_i | v_{\pi_{\hat{G}}}^{(i-1)})$$

[Changed the graph from $G$ to $\hat{G}$ since $\hat{\mathbf{X}} \in \hat{G}$ and only affects the descendants of $\hat{\mathbf{X}}$.]

And $P_{\text{ID-GEN}}(\mathbf{y}|\hat{\mathbf{x}})$ can be obtained by:

$$P_{\text{ID-GEN}}(\mathbf{y}|\hat{\mathbf{x}}) = \sum_{v \backslash y} \prod_{v_i \in V} P(v_i | v_{\pi_{\hat{G}}}^{(i-1)}) \tag{4}$$

Since for fixed $\hat{\mathbf{X}} = \hat{\mathbf{x}}$, $P_{ID}(\mathbf{v}) = P_{\text{ID-GEN}}(\mathbf{v}|\hat{\mathbf{x}})$ according to Lemma D.15. Thus, the corresponding conditional distributions in Equation 3 and 4 are equal, i.e, $P_{ID}(v_i | v_{\pi_G}^{(i-1)}) = P_{\text{ID-GEN}}(v_i | v_{\pi_{\hat{G}}}^{(i-1)}), \forall\{i|V_i \in \mathbf{V}\}$. Therefore, $P_{ID}(\mathbf{y})$ and $P_{\text{ID-GEN}}(\mathbf{y}|\hat{\mathbf{x}})$ having the same factorization and the corresponding conditional distributions being equal implies that $P_{ID}(\mathbf{y}) = P_{\text{ID-GEN}}(\mathbf{y}|\hat{\mathbf{x}})$.

Now, based on Assumption D.5, ID-GEN learns to sample from each conditional distribution $P_{\text{ID-GEN}}(v_i|v_\pi^{(i-1)}, v_{\pi_{\hat{G}}}^{(i-1)})$ of the product in Equation 4, by training a conditional model $M_{V_i}$ on samples from the dataset $\mathcal{D}[\hat{\mathbf{X}}, \mathbf{V}] \sim P(\hat{\mathbf{x}}, \mathbf{v})$.

For each $V_i$, we add a node containing $V_i$ and its associated conditional generative model $M_{V_i}$ to a sampling network. This produces a sampling network $\mathcal{H}$ that is a DAG, where each variable $V_i \in \mathbf{V}$ has a sampling mechanism. By factorization and Assumption D.5, ancestral sampling from this sampling graph produces samples from $P_{\text{ID-GEN}}(\mathbf{v}|\hat{\mathbf{x}})$. We can drop values of $V \setminus Y$ to obtain the samples from $P_{\text{ID-GEN}}(\mathbf{y}|\hat{\mathbf{x}})$. Since $P_{ID}(\mathbf{v}) = P_{\text{ID-GEN}}(\mathbf{v}|\hat{\mathbf{x}})$, both factorized in the same manner and $M_{V_i}$ learned conditional distribution of ID-GEN's factorization, the sampling network returned by ID-GEN, will correctly sample from the distribution $P_{ID}(\mathbf{y})$ returned by the ID algorithm for fixed $\hat{\mathbf{X}} = \hat{\mathbf{x}}$. Thus, the sampling network returned by ID-GEN is valid for $P(\mathbf{y})$.

$\square$

**Lemma D.18.** *ID-GEN Base case (step 6): Let, at any recursion level of ID-GEN, the input dataset $\mathcal{D}[\hat{\mathbf{X}}, \mathbf{V}]$ is sampled from a specific joint distribution $P(\hat{\mathbf{X}}, \mathbf{V})$, i.e., $\mathcal{D}[\hat{\mathbf{X}}, \mathbf{V}] \sim P(\hat{\mathbf{X}}, \mathbf{V})$ where $\hat{\mathbf{X}}$ is the set of intervened variables at step 7s. Given an identifiable interventional query $P_{\mathbf{x}}(\mathbf{y})$ over a causal graph $G = (V, E)$, suppose ID-GEN $(\mathbf{Y}, \mathbf{X}, G, \mathcal{D}, \hat{\mathbf{X}}, \hat{G})$ immediately enters step 6 and returns $\mathcal{H}$. Then $\mathcal{H}$ is a valid sampling network for $P_{\mathbf{x}}(\mathbf{y})$ with fixed consistent $\hat{\mathbf{X}} = \hat{\mathbf{x}}$.*

*Proof.* By Lemma D.11, both ID-GEN $(\mathbf{Y}, \mathbf{X}, G, \mathcal{D}, \hat{\mathbf{X}}, \hat{G})$ and $ID(\mathbf{Y}, \mathbf{X}, P, G)$ enter the same base case step 6. By the condition of step 6, $G \setminus \mathbf{X}$ has only one c-component $\{S\}$, where $S \in C(G)$.

Let $P(\mathbf{v})$ be the current distribution of the ID algorithm after performing a series of marginalizations in step 2 and intervention in step 7 on the input observational distribution. Let $\mathcal{D}[\hat{\mathbf{X}}, \mathbf{V}] \sim P'(\hat{\mathbf{x}}, \mathbf{v})$ be the dataset parameter of ID-GEN algorithm that went through the same transformations as the ID algorithm in the sample space. $\hat{\mathbf{X}}$ is the set of interventions that are applied on the dataset at step 7s. According to Lemma D.15 for fixed values of $\hat{\mathbf{X}} = \hat{\mathbf{x}}$, $P(\mathbf{v}) = P'(\mathbf{v}|\hat{\mathbf{x}})$ holds true at any recursion level.

Since ID algorithm is sound it returns correct output for the query $P_{\mathbf{x}}(\mathbf{y})$. We prove the soundness of ID-GEN step 6 by showing that the sampling network ID-GEN returns, is valid for the output of the ID algorithm.

With the joint distribution $P(\mathbf{v})$, the soundness of ID implies that

$$P_{\mathbf{x}}(\mathbf{y}) = \sum_{S \setminus Y} \prod_{\{i | V_i \in S\}} P(v_i | v_{\pi_G}^{(i-1)}) \tag{5}$$

where $\pi_G$ is a topological ordering for $G$.

With the joint distribution $P'(\mathbf{v}|\hat{\mathbf{x}})$, ID-GEN can factorize $P'_{\mathbf{x}}(\mathbf{y}|\mathbf{x})$ in the same manner:

$$P'_{\mathbf{x}}(\mathbf{y}|\hat{\mathbf{x}}) = \sum_{S \setminus Y} \prod_{\{i | V_i \in S\}} P'(v_i | v_{\pi_G}^{(i-1)}, \hat{\mathbf{x}})$$

$$= \sum_{S \setminus Y} \prod_{\{i | V_i \in S\}} P'(v_i | v_{\pi_{\hat{G}}}^{(i-1)}) \tag{6}$$

[Changed the graph from $G$ to $\hat{G}$ since $\hat{\mathbf{X}} \in \hat{G}$ and only affects the descendants of $\hat{\mathbf{X}}$.]

Since $P(\mathbf{v}) = P'(\mathbf{v}|\hat{\mathbf{x}})$, the corresponding conditional distributions in Equation 5 and 6 are equal, i.e, $P(v_i|v_{\pi_G}^{(i-1)}) = P'(v_i|v_{\pi_{\hat{G}}}^{(i-1)}), \forall\{i | V_i \in S\}$. Therefore, $P_{\mathbf{x}}(\mathbf{y})$ and $P'_{\mathbf{x}}(\mathbf{y}|\hat{\mathbf{x}})$ having the same factorization and the corresponding conditional distributions being equal implies that $P_{\mathbf{x}}(\mathbf{y}) = P'_{\mathbf{x}}(\mathbf{y}|\hat{\mathbf{x}})$.

ID-GEN operates in this case by training, from joint samples $\mathcal{D}[\hat{\mathbf{X}}, V]$, a model to correctly sample each $P'(v_i \mid v_{\pi_{\hat{G}}}^{(i-1)})$ term, i.e., we learn a conditional generative model $M_{V_i}(V_{\pi_{\hat{G}}}^{(i-1)})$ which produces samples from $P'(v_i \mid v_{\pi_{\hat{G}}}^{(i-1)})$, which we can do according to Assumption D.5. Then we construct a sampling network $\mathcal{H}$ by creating a node $V_i$ with a sampling mechanism $M_{V_i}$ for each $V_i \in S$. We add

edges from $V_j \to V_i$ for each $V_j \in V_{\pi_{\hat{G}}}^{(i-1)}$. Since every vertex in $\hat{G}$ is either in $S$ or in $\mathbf{X} \cup \hat{\mathbf{X}}$, every edge either connects to a previously constructed node or a variable in $\mathbf{X} \cup \hat{\mathbf{X}}$. Since we already have fixed values for $\hat{\mathbf{X}} = \hat{\mathbf{x}}$, when we specify values for $\mathbf{X}$ and sample according to topological order $\pi_{\hat{G}}$, this sampling graph provides samples from the distribution $\prod_{\{i | V_i \in S\}} P'(v_i | v_{\pi_{\hat{G}}}^{(i-1)})$, i.e. $P'_{\mathbf{x}}(\mathbf{s}|\hat{\mathbf{x}})$.

Since, as shown earlier, $P'_{\mathbf{x}}(\mathbf{s}|\hat{\mathbf{x}}) = P_{\mathbf{x}}(\mathbf{s})$, samples from the sampling network $\mathcal{H}$ is consistent with $P_{\mathbf{x}}(\mathbf{s})$ as well. We obtain samples from $P_{\mathbf{x}}(\mathbf{y})$ by dropping the values of $S \setminus Y$ from the samples obtained from $P_{\mathbf{x}}(\mathbf{s})$. We assert the remaining conditions to show that this sampling network is correct for $P_{\mathbf{x}}(\mathbf{y})$: certainly this graph is a $DAG$ and every $v \in S$ has a conditional generative model in $\mathcal{H}$. By the conditions to enter step 6, $\hat{G} = S \cup \mathbf{X} \cup \hat{\mathbf{X}}$ and $S \cap \{\mathbf{X} \cup \hat{\mathbf{X}}\} = \emptyset$. Then every node in $\mathcal{H}$ is either in $S$ or is in $\mathbf{X} \cup \hat{\mathbf{X}}$: hence the only nodes without sampling mechanisms are those in $\mathbf{X} \cup \hat{\mathbf{X}}$ as desired. Therefore, when $\hat{\mathbf{X}}$ is fixed as $\hat{\mathbf{x}}$, $\mathcal{H}$ is a valid sampling network for $P_{\mathbf{x}}(\mathbf{y})$. $\square$

**Proposition D.19.** *At any level of the recurison, the graph parameters $G$ and $\hat{G}$ in ID-GEN* $(\mathbf{Y}, \mathbf{X}, G, \mathcal{D}, \hat{\mathbf{X}}, \hat{G})$ *have the same topological order excluding* $\hat{\mathbf{X}}$.

*Proof.* Let $\pi_G$ be the topological order of $G$ and $\pi_{\hat{G}}$ be the topological order of $\hat{G}$. At the beginning of the algorithm $G = \hat{G}$. Thus, $\pi_G = \pi_{\hat{G}}$. According to the lemma D.13, at any level of recursion, $G_{ID} = G_{\text{ID-GEN}}$ and $G_{ID} = \hat{G} \setminus \hat{\mathbf{X}}$. Thus, we have $G = G_{\text{ID-GEN}} = G_{ID} = \hat{G} \setminus \hat{\mathbf{X}}$. Therefore, $\pi_G = \pi_{\hat{G} \setminus \hat{\mathbf{X}}}$.

$\square$

**Lemma D.20.** *Let $\mathcal{H}$ be a sampling network produced by ID-GEN from an identifiable query $P_{\mathbf{x}}(\mathbf{y})$ over a graph $G$. If $G$ has the topological ordering $\pi$, then every edge in the sampling graph of $\mathcal{H}$ adheres to the ordering $\pi$.*

*Proof.* We consider two factors: which edges are added, and with respect to which graphs. Since the only base cases ID-GEN enters are steps 1 and 6, the only edges added are consistent with the topological ordering $\pi$ for the graph that was supplied as an argument to these base case calls. The only graph modifications occur in steps 2 and 7, and these yield subgraphs of $G$. Thus the original topological ordering $\pi$ for graph $G$ is a valid topological ordering for each restriction of $G$. Therefore any edge added to $\mathcal{H}$ is consistent with the global topological ordering $\Pi$. $\square$

**Lemma D.21.** *Let $\mathcal{H}$ be a sampling network for random variables $\{V_1, V_2, \ldots V_n\}$ formed by a collection of conditional generative models $M_{V_i}$ relative to $P(\mathbf{v})$ for all $V_i$. Then the tuple $(V_1, V_2 \ldots V_n)$ obtained by sequentially evaluating each conditional generative model relative to the topological order of the sampling graph is a sample from the joint distribution $\Pi_i P_i(v_i | pa_i)$.*

*Proof.* Without loss of generality, let $(V_1, V_2, \ldots, V_N)$ be a total order that is consistent with the topological ordering over the nodes in $G$. To attain a sample from the joint distribution, sample each $V_i$ in order. When sampling $V_j$, each $V_i$ for all $i < j$ is already sampled, which is a superset of $Pa_j$ (the inputs to $M_{V_j}$) by definition of topological orderings. Thus, all inputs to every conditional generative model $M_{V_j}$ are available during sampling. Since each $V_j$ is conditionally independent of $V_i \notin Pa_j$, the joint distribution factorizes as given in the claim. $\square$

**Theorem D.22.** *ID-GEN Soundness: Let $P_{\mathbf{x}}(\mathbf{y})$ be an identifiable query given the causal graph $G = (V, E)$ and that we have access to joint samples $\mathcal{D} \sim P(\mathbf{v})$. Then the sampling network returned by ID-GEN $(\mathbf{Y}, \mathbf{X}, G, \mathcal{D}, \hat{\mathbf{X}}, \hat{G})$ correctly samples from $P_{\mathbf{x}}(\mathbf{y})$ under Assumption D.5.*

*Proof sketch:* Suppose that $P_{\mathbf{x}}(\mathbf{y})$ is the input causal query and Assumptions D.3, D.4, D.5, D.6 hold. The soundness of ID-GEN implies that if the trained conditional models converge to (near) optimality, ID-GEN returns the correct samples from $P_{\mathbf{x}}(\mathbf{y})$. For each step of the ID algorithm that deals with probabilities of discrete variables, multiple actions are performed in the corresponding step of ID-GEN to correctly train conditional models to sample from the corresponding distributions. ID-GEN merges these conditional models according to the topological order of $G$, to build the final sampling network $\mathcal{H}$. Therefore, according to structural induction, when we intervened on $\mathbf{X}$ and

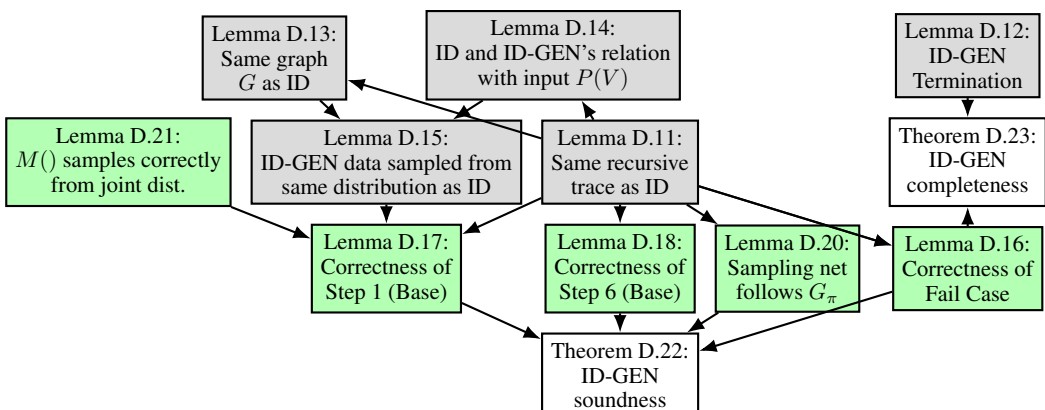

Figure 13: Flow Chart of Proofs

perform ancestral sampling in $\mathcal{H}$ each model in the sampling network will contribute correctly to generate samples from $P_x(y)$.

*Proof.* We proceed by structural induction. We start from the base cases, i.e., the steps that do not call ID-GEN again. ID-GEN only has three base cases: step 1 is the case when no variables are being intervened upon and is covered by Lemma D.17; step 6 is the other base case and is covered by Lemma D.18; step 5 is the non-identifiable case and since we assumed that $P_{\mathbf{x}}(\mathbf{y})$ is identifiable, we can skip ID-GEN's step 5.

The structure of our proof is as follows. By the assumption that $P_{\mathbf{x}}(\mathbf{y})$ is identifiable and due to Lemma D.12, its recursions must terminate in steps 1 or 6. Since we have already proven correctness for these cases, we use these as base cases for a structural induction. We prove that if ID-GEN enters any of step 2, 3, 4 or 7, under the inductive assumption that we have correct sampling network for the recursive calls, we can produce a correct overall sampling network. The general flavor of these inductive steps adheres to the following recipe: i) determine the corresponding recursive call that ID algorithm makes; ii) argue that we can generate the correct dataset to be analogous to the distribution that ID uses in the recursion; iii) rely on the inductive assumption that the generated DAG from ID-GEN's recursion is correct.

We consider each recursive case separately. We start with step 2. Suppose ID-GEN $(\mathbf{Y}, \mathbf{X}, G, \mathcal{D}, \hat{\mathbf{X}}, \hat{G})$ enters step 2, then according to Lemma D.11, $ID(\mathbf{Y}, \mathbf{X}, P, G)$ enters step 2 as well. Hence the correct distribution to sample from is provided by ID step 2:

$$P_{\mathbf{x}}(\mathbf{y}) = ID(\mathbf{Y}, \mathbf{X} \cap An(Y)_G, \sum_{\mathbf{V} \setminus An(\mathbf{Y})_G} P(\mathbf{v}), G_{An(\mathbf{Y})}).$$

Now, according to Lemma D.15, at the current step, the dataset $\mathcal{D}$ of ID-GEN is sampled from a distribution $P'(\mathbf{v}|\hat{\mathbf{x}})$ such that for fixed values of $\hat{\mathbf{X}} = \hat{\mathbf{x}}$, $P(\mathbf{v}) = P'(\mathbf{v}|\hat{\mathbf{x}})$ holds true. Following our recipe, we need to update the dataset $\mathcal{D}$ such that it is sampled from $\sum_{V \setminus An(Y)_G} P(\mathbf{v}) = \sum_{V \setminus An(Y)_G} P'(\mathbf{v}|\hat{\mathbf{x}})$. We do this by dropping all non-ancestor variables of $\mathbf{Y}$ (in the graph $G$) from the dataset $\mathcal{D}$, thereby attaining samples from the joint distribution $\sum_{\mathbf{V} \setminus An(\mathbf{Y})_G} P(\mathbf{v})$. Since $\hat{G}$ is used at the base case, we update it as $\hat{G}_{An(\mathbf{Y})}$, the same as $G_{An(\mathbf{Y})}$ to propagate the correct graph at the next step. Therefore, we can generate the sampling network from ID-GEN($\mathbf{Y}, \mathbf{X} \cap An(\mathbf{Y})_G, G_{An(\mathbf{Y})}, \mathcal{D}[An(Y)], \hat{\mathbf{X}}, \hat{G}_{An(Y)}$) by the inductive assumption and simply return it.

Next, we consider step 3. Suppose ID-GEN $(\mathbf{Y}, \mathbf{X}, G, \mathcal{D}, \hat{\mathbf{X}}, \hat{G})$ enters step 3. Then by Lemma D.11, $ID(\mathbf{Y}, \mathbf{X}, P, G)$ enters step 3, and the correct distribution to sample from is provided from ID step 3 as

$$P_{\mathbf{x}}(\mathbf{y}) = ID(\mathbf{Y}, \mathbf{X} \cup W, P, G)$$

where $W := (\mathbf{V} \setminus \mathbf{X}) \setminus An(\mathbf{Y})_{G_{\overline{\mathbf{X}}}}$. Since the distribution passed to the recursive call is $P$, we can simply return the sampling graph generated by ID-GEN $(\mathbf{Y}, \mathbf{X}, G, \mathcal{D}, \hat{\mathbf{X}}, \hat{G})$, which we know is

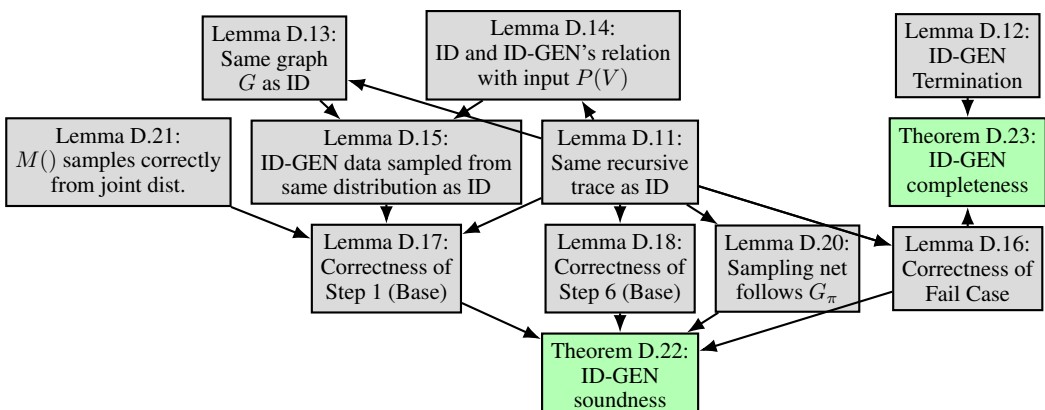

Figure 14: Flow Chart of Proofs

correct for $P_{\mathbf{X} \cup \mathbf{W}}(\mathbf{Y})$ by the inductive assumption. Thus, the returned sampling network by ID-GEN can sample from $P_{\mathbf{x}}(\mathbf{y})$. While we do need to specify a sampling mechanism for $\mathbf{W}$ to satisfy our definition of a valid sampling network, this can be chosen arbitrarily, say $\mathbf{W} \sim P(\mathbf{w})$ or uniform the distribution.

Next we consider step 4. Suppose ID-GEN $(\mathbf{Y}, \mathbf{X}, G, \mathcal{D}, \hat{\mathbf{X}}, \hat{G})$ enters step 4. Then by Lemma D.11, $ID(\mathbf{Y}, \mathbf{X}, P, G)$ enters step 4 and the correct distribution to sample from is provided from ID step 4 as:

$$\sum_{V \setminus (y \cup x)} \prod_i ID(s_i, v \setminus s_i, P, G)$$

where $S_i$ are the c-components of $G \setminus \mathbf{X}$, i.e., elements of $C(G \setminus \mathbf{X})$. By the inductive assumption, we can sample from each term in the product with the sampling network returned by ID-GEN$(S_i, \mathbf{X} = \mathbf{V} \setminus S_i, G, \mathcal{D}, \hat{\mathbf{X}}, \hat{G})$. However, recall the output of ID-GEN: ID-GEN returns a 'headless' (no conditional models for $\mathbf{X}$) sampling network as follows:

ID-GEN $(\mathbf{Y}, \mathbf{X}, G, \mathcal{D}, \hat{\mathbf{X}}, \hat{G})$ returns a sampling network, i.e., a collection of conditional generative models where for each variable in $G$ and every variable except those in $\mathbf{X}$ have a specified conditional generative model. To sample from this sampling network, values for $\mathbf{X}$ must first be specified. In the step 4 case, the values $\mathbf{v} \setminus s_i$ need to be provided to sample values for $S_i$, and similarly for $i \neq j$, values $\mathbf{v} \setminus s_j$ are needed to sample values for $S_j$. Since $S_i \subseteq (\mathbf{V} \setminus S_j)$ and $S_j \subseteq (\mathbf{V} \setminus S_i)$, it might lead to cycles (as shown in Example C.1) if we attempt to generate samples for each c-components sequentially. Thus, it does not suffice to sample from each c-component sequentially or separately.

Note that, $H_i$ is the correct sampling network corresponding to ID-GEN$(S_i, \mathbf{X} = \mathbf{V} \setminus S_i, G, \mathcal{D}, \hat{\mathbf{X}}, \hat{G})$ by definition, for each node $V_i \in S_i$, $V_i$ has a conditional generative model in $H_i$. By Lemma D.20, each edge in $H_i$ adheres to the topological ordering $\pi_G$ (at the current level). Hence, if we apply `MergeNetwork(.)` to construct a graph $\mathcal{H}$ from $\{H_i\}_i$, it will also adhere to the original topological ordering $\pi_G$. Thus, $\mathcal{H}$ is a DAG.

Since every node $V_i$ in $G \setminus \mathbf{X}$ has a conditional generative model in some $H_i$, the only nodes in combined $\mathcal{H}$ without conditional generative models are those in $\mathbf{X}$. Finally, since each node in $\mathcal{H}$ samples the correct conditional distribution by the inductive assumption, $\mathcal{H}$ samples from the product distribution $P_{\mathbf{x}}(\mathbf{y})$ correctly. The sum $\sum_{\mathbf{v} \setminus (\mathbf{y} \cup \mathbf{x})}$ can be safely ignored now and can be applied later since the sample values of the marginalized variables ($\mathbf{v} \setminus (\mathbf{y} \cup \mathbf{x})$) can be dropped from the joint at the end of the algorithm to attain samples values of the remaining variables. Hence $\mathcal{H}$ is correct for $P_{\mathbf{x}}(\mathbf{y})$.

Step 5 can never happen by the assumption that $P_{\mathbf{x}}(\mathbf{y})$ is identifiable, and step 6 has already been covered as a base case. The only step remaining is step 7.

Lemma D.11 says, ID-GEN $(\mathbf{Y}, \mathbf{X}, G, \mathcal{D}, \hat{\mathbf{X}}, \hat{G})$ enters step 7 by the same conditions $ID(\mathbf{Y}, \mathbf{X}, P, G)$ enters step 7. Then by assumption, $C(G \setminus \mathbf{X}) = \{S\}$ and there exists a confounding component $S' \in C(G)$ such that $S \subset S'$. The correct distribution to sample from is

provided from ID step 7 as

$$P_{\mathbf{x}}(\mathbf{y}) = ID(\mathbf{Y}, \mathbf{X} \cap S', P', G_{S'})$$

where

$$P' := \prod_{\{i | V_i \in S'\}} P(V_i | V_\pi^{(i-1)} \cap S', v_\pi^{(i-1)} \setminus S').$$

Examining ID algorithm more closely, if we enter step 7 during ID, the interventional set $\mathbf{X}$ is partitioned into two components: $\mathbf{X} \cap S'$ and $\mathbf{X}_Z := \mathbf{X} \setminus S'$. From Lemmas 33 and 37 of Shpitser and Pearl [46], in the event we enter step 7, $P_{\mathbf{x}}(\mathbf{y})$ is equivalent to $P'_{\mathbf{x} \cap S'}(\mathbf{y})$ where $P'(\mathbf{v}) = P_{\mathbf{x}_Z}(\mathbf{v})$. ID estimates $P_{\mathbf{x}_Z}(\mathbf{v})$ with a similar computation as the step 6 base case.

To sample correctly in ID-GEN, we consider two cases.

i) $\hat{\mathbf{X}} = \emptyset$: When ID-GEN visits step 7 for the first time, we have $\hat{\mathbf{X}} = \emptyset$. In that case, we first update our dataset $\mathcal{D}[\mathbf{V}] \sim P(\mathbf{v})$ to samples from $\mathcal{D}' \sim P_{\mathbf{x}_Z}(\mathbf{v})$, and then we recurse on the query $P'_{\mathbf{x} \cap S'}(\mathbf{y})$ over the graph $G_{S'}$. Also, $\mathbf{X}_Z$ is outside the c-component $S'$. Therefore we generate a dataset from $\mathcal{D}' \sim P_{\mathbf{x}_Z}(S')$ via running directly the `ConditionalGMs(.)` of the ID-GEN algorithm, Algorithm 2: `ConditionalGMs`$(S', X_Z, G, \mathcal{D}, \hat{\mathbf{X}}, \hat{G})$. This is attainable via the inductive assumption and Lemma D.12. The only divergence from ID during the generation of $\mathcal{D}'$ is that ID presumes pre-specified fixed values for $\mathbf{X}_Z$, where we train a sampling mechanism that is agnostic a priori to the specific choice of $\mathbf{X}_Z$. To sidestep this issue, we generate a dataset with all possible values of $\mathbf{X}_Z$ and be sure to record the values of $\mathbf{X}_Z$ in the dataset $\mathcal{D}'[\mathbf{X}_Z, S']$.

ii) $\hat{\mathbf{X}} \neq \emptyset$: When ID-GEN has visited step 7 already once and thus $\hat{\mathbf{X}} \neq \emptyset$. We consider $\hat{\mathbf{X}}$ along with $\mathbf{X}_Z$ when generating $\mathcal{D}'$ this time. More precisely, we update our dataset $\mathcal{D}[\hat{\mathbf{X}}, \mathbf{V}] \sim P(\hat{\mathbf{x}}, \mathbf{v})$ to samples from $\mathcal{D}' \sim P_{\mathbf{x}_Z \cup \hat{\mathbf{x}}}(\mathbf{x}_z, \hat{\mathbf{x}}, \mathbf{v})$. Rest of the steps follow similarly as the above case. We record the new $\mathbf{X}_Z$ in $\hat{\mathbf{X}}$ and carry them in $\mathcal{D}'[\hat{\mathbf{X}}, S']$ and $\hat{G}$.

Next, we need to map the recursive call $ID(\mathbf{Y}, \mathbf{X} \cap S', P', G)$ to ID-GEN. ID-GEN sends the same parameters $\mathbf{Y}, \mathbf{X} \cap S'$ and $G$ as ID. Now, equivalent to passing the distribution $P'$ of ID, we pass the dataset $\mathcal{D}[\hat{\mathbf{X}}, S']$ sampled from this distribution, including the intervened values for $\hat{\mathbf{X}}$ used to obtain this dataset. According to Lemma D.15, this dataset is sampled from $P'$ if we fix to a specific value $\hat{\mathbf{X}} = \hat{\mathbf{x}}$. Finally, ID algorithm uses specific value of $\mathbf{X}_Z$ and then ignores those variables from $\mathbf{X}$ and $G$ for rest of the recursion. On the other hand, ID-GEN saves $\mathbf{X}_Z$ in $\hat{\mathbf{X}}$ as $\hat{\mathbf{X}} = \hat{\mathbf{X}} \cup \mathbf{X}_Z$ and keeps $\hat{\mathbf{X}}$ connected to $\hat{G}$ with incoming edges cut, i.e., $G_{S', \overline{\hat{\mathbf{X}}}}$ for the next recursive calls since ID-GEN utilizes $\hat{\mathbf{X}}$ and the topological order in $\hat{G}$ at the base cases (step 1 and 6). By the inductive assumption, we can generate a correct sampling network from the call ID-GEN$(\mathbf{Y}, \mathbf{X} \setminus \mathbf{X}_Z, G_{S'}, \mathcal{D}[\hat{\mathbf{X}}, S], \hat{\mathbf{X}}, \hat{G}_{\{S', \overline{\hat{\mathbf{X}}}\}})$, and hence the returned sampling graph is correct for $P_{\mathbf{x}}(\mathbf{y})$.

Since we have shown that every recursion of ID-GEN ultimately terminates in a base case, that all the base cases provide correct sampling graphs, and that correct sampling graphs can be constructed in each step assuming the recursive calls are correct, we conclude that ID-GEN returns the correct sampling graph for $P_{\mathbf{x}}(\mathbf{y})$.

$\square$

**Theorem D.23.** *ID-GEN is complete.*

*Proof.* Suppose we are given a causal query $P_{\mathbf{x}}(\mathbf{y})$ as input to sample from. We prove that if ID-GEN fails, then the query $P_{\mathbf{x}}(\mathbf{y})$ is non identifiable, implying that it is not possible to train conditional models on observational data and correctly sample from the interventional distribution $P_{\mathbf{x}}(\mathbf{y})$.

If ID-GEN reaches step 5, it returns `FAIL`. According to the lemma D.11, ID reaches at step 5 if and only if ID-GEN reaches at step 5. Step 5 is also the `FAIL` case of the ID algorithm. Since ID is complete and returns `FAIL` at Step 5, the query $P_x(y)$ is not identifiable.

$\square$

# E    Conditional interventional sampling

**Conditional sampling:** Given a conditional causal query $P_x(y|z)$, we sample from this conditional interventional query by calling Algorithm 5: IDC-GEN. This function finds the maximal set $\alpha \subset Z$ such that we can apply rule-2 and move $\alpha$ from conditioning set $Z$ and add it to intervention set $X$. Precisely, $P_x(y|z) = P_{x \cup \alpha}(y|z \setminus \alpha) = \frac{P_{x \cup \alpha}(y, z \setminus \alpha)}{P_{x \cup \alpha}(z \setminus \alpha)}$. Next, Algorithm 1: ID-GEN(.) is called to obtain the sampling network that can sample from the interventional joint distribution $P_{x \cup \alpha}(y, z \setminus \alpha)$. We use the sampling network to generate samples $\mathcal{D}'$ through feed-forward. A new conditional model $M_Y$ is trained on $\mathcal{D}'$ that takes $Z \setminus \alpha$ and $X \cup \alpha$ as input and outputs $Y$. Finally, we generate new samples with $M_Y$ by feeding input values such that $Y \sim P_{x \cup \alpha}(y, z \setminus \alpha)$ i.e, $Y \sim P_x(y|z)$.

**Theorem E.1** (Shpitser and Pearl [46])**.** *For any $G$ and any conditional effect $P_X(Y|W)$ there exists a unique maximal set $Z = \{Z \in W | P_X(Y|W) = P_{X,Z}(Y|W \setminus Z)\}$ such that rule 2 applies to $Z$ in $G$ for $P_X(Y|W)$. In other words, $P_X(Y|W) = P_{X,Z}(Y|W \setminus Z)$.*

**Theorem E.2** (Shpitser and Pearl [46])**.** *Let $P_X(Y|W)$ be such that every $W \in W$ has a back-door path to $Y$ in $G \setminus X$ given $W \setminus \{W\}$. Then $P_X(Y|W)$ is identifiable in $G$ if and only if $P_X(Y,W)$ is identifiable in $G$.*

**Theorem E.3.  *IDC-GEN Soundness:*** *Let $P_X(Y|Z)$ be an identifiable query given the causal graph $G = (V, E)$ and that we have access to joint samples $\mathcal{D} \sim P(v)$. Then the sampling network returned by IDC-GEN $(Y, X, Z, \mathcal{D}, G)$ correctly samples from $P_X(Y|Z)$ under Assumptions D.3, D.4, D.5, D.6.*

*Proof.* The IDC algorithm is sound and complete based on Theorem E.1 and Theorem E.2. For sampling from the conditional interventional query, we follow the same steps as the IDC algorithm in Algorithm 5: IDC-GEN and call the sound and complete Algorithm 1:ID-GEN as sub-procedure. Therefore, IDC-GEN is sound and complete.  □

# F    Experimental details

## F.1    Training details and compute

We performed some of our experiments on a machine with an RTX-3090 GPU. We also performed some training on 2 A100 GPU's which took roughly 9 hours for 1000 epochs. Training for baseline NCM took more than 50 hours to complete 1000 epochs. The baseline DCM took around 10 hours. When variables were low-dimensional discrete, it was quite fast and took 10-20 minutes to finish all training and get convergence. We discuss more specifics about each experiment in their individual sections.

### F.1.1    Reproducibility

For reproducibility purposes, we provide our anonimized source codes with instructions. Besides, we provided explanations of each experiment along with model settings and hyperparameters. We provide the code to generate the Colored MNIST dataset.

## F.2    Napkin-MNIST dataset

**Data Generation:** First we consider a synthetic dataset imbued over the napkin graph. We consider variables $W_1, W_2, X, Y$, where $W_1, X, Y$ are images derived from MNIST and $W_2$ is a paired discrete variable. We introduce latent confounders $C, T$, denoting color and thickness, where $C$ can be any of $\{red, greed, blue, yellow, magenta, cyan\}$, and $T$ can be any of $\{thin, regular, thick\}$. Data generation proceeds as follows: first we sample latent $C, T$ from the uniform distribution. We color and reweight a random digit from MNIST to form $W_1$. $W_2$ only keeps the digit value in $\{0 \ldots 9\}$ of $W_1$ and a restriction of its color: if the color of $W_1$ is $red$, $green$, or $blue$, $W_2$'s color value is 0, and it is 1 otherwise. $X$ then picks a random MNIST image of the same digit as $W_2$'s digit value (0-9), is colored according to $W_2$'s color value (0-1), and is reweighted according to the latent $T$. Then $Y$ is the same original MNIST image as $X$, with the same thickness but colored according to the latent $C$. Further, for every edge in the graph, we include a random noising process: with

probability 0.1, the information passed along is chosen in a uniformly random manner from the valid range.

Here we describe the data-generation procedure, and training setup for the Napkin-MNIST experiment in full detail.

### F.2.1 Data generation procedure: discrete case

As a warm-up, we outline the generation for the Napkin-MNIST dataset in a low-dimensional setting. When we consider in the next section the high-dimensional case, we simply replace some of these discrete variables with MNIST images which can be mapped back into this low-dimensional case.

We start by enumerating the joint distribution and the support of each marginal variable. First lets define the sets

- COLORS := {red, green, blue, yellow, magenta, cyan}.
- RG_COLORS := {red, green}.
- THICKNESSES := {thin, regular, thick}.
- DIGITS := $\{0, \ldots, 9\}$.

And then the definitions and support of each of the variables in our distribution:

- (Latent) Color $\in$ COLORS.
- (Latent) Thickness $\in$ THICKNESSES.
- $W_1 \in$ DIGITS $\times$ COLORS $\times$ THICKNESSES
- $W_2 \in$ DIGITS $\times$ RG_COLORS.
- $X \in$ DIGITS $\times$ COLORS $\times$ THICKNESSES
- $Y \in$ DIGITS $\times$ COLORS $\times$ THICKNESSES

Now we describe the full data generation procedure. A key hyperparameter is a noise-probability $p$. This defines the probability that any variable flips to a uniform probability. To ease notation, we define the function $\eta_p(v, S)$ defined as

$$\eta_p(v, S) := \begin{cases} v & \text{with probability } 1 - p \\ U(S) & \text{otherwise} \end{cases}$$

and we define the mapping $R :$ COLORS $\to$ RESTRICTED_COLORS as

$$R(c) := \begin{cases} \text{red} & \text{if } c \in \{\text{red, green, blue}\} \\ \text{green} & \text{otherwise} \end{cases}$$

Where $U(S)$ means a uniformly random choice of $S$. Then our data generation procedure follows the following steps:

- Color := $U($COLORS$)$
- Thickness := $U($THICKNESSES$)$
- $W_1 := \big(U(\text{DIGITS}), \quad \eta_p(\text{Color}, \text{COLORS}), \quad \eta_p(\text{Thickness}, \text{THICKNESSES})\big)$
- $W_2 := \big(\eta_p(W_{1 \cdot digit}, \text{DIGITS}), \quad \eta_p(R(W_{1 \cdot color}, \text{RG\_COLORS}))$
- $X := \big(\eta_p(W_{2 \cdot digit}, \text{DIGITS}), \quad \eta_p(W_{2 \cdot color}, \text{RG\_COLORS}), \quad \eta_p(\text{Thickness}, \text{THICKNESSES})\big)$
- $Y := \big(\eta_p(X_{\cdot digit}, \text{DIGITS}), \quad \eta_p(\text{Color}, \text{COLORS}), \quad \eta_p(X_{\cdot thickness}, \text{THICKNESSES})\big)$

It is easy to verify that this describes the Napkin graph, as each only Color, Thickness are latent and each variable only depends on its parents in the SCM.

Secondly, observe that this structural causal model is separable with respect to digits, colors, and thicknesses. Since each digit only depends on parent *digits*, each color only depends on parent *colors*, and each thickness depends only on parent *thicknesses*, these can all be considered separately.

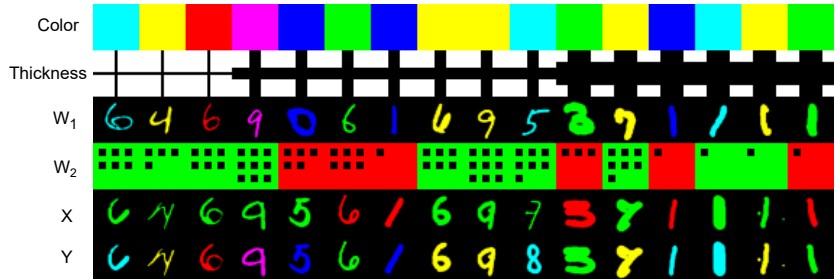

Figure 15: **Joint samples from the Napkin-MNIST dataset:** Samples from the Napkin-MNIST dataset are visualized as columns above. The first row indicates the latent variable `color`, the second row indicates the latent variable `thickness`, and the row labeled $W_2$ is a discrete variable holding a (`color`, `digit`), where digit is represented as the number of dots. Notice that the noising process sometimes causes information to not be passed to children.

Further, because this distribution is only supported over discrete variables, exact likelihoods can be computed for any conditional query. This is much more easily done programmatically, however, and we provide code in the attached codebase to do just that. We will claim without proof that in the case of thicknesses and digits, $P_Y(X) = P(Y|X)$. However in the case of colors, $P_Y(X) \neq P(Y|X)$. Hence we consider this case in the evaluations in the experiments section.

### F.2.2 Data generation procedure: high-dimensional case

The high-dimensional case follows the discrete case of the Napkin-MNIST dataset, with a few key changes. Namely, $W_1, X$, and $Y$ are MNIST images that have been colored and thickened. We explicitly outline these changes:

- $W_1$: A random MNIST image of the provided digit is used, then colored and thickened accordingly (noisy from latents).

- $W_2$ : This is a discrete variable, only encoding the (noised) digit and (noised) restricted color of $W_1$.

- $X$: This is a random MNIST image of the (noised) digit obtained from $W_2$, then colored with the (noised) restricted color from $W_2$ and thickened according to the (noised) latent thickness.

- $Y$: This is the *same* base image of $X$, unless the noising procedure calls for a change in digit, then a random MNIST image of the specified image is used. The (noisy) color is obtained from the latent distribution, and the (noisy) thickness is obtained from $X$.

To color the images, we convert each 1-channel MNIST image into a 3-channel MNIST image, and populate the necessary channels to generate these colors. Note that in RGB images: if only the RG channels are active, the image is yellow; if only the RB channels are active, the image is magenta; if only the BG channels are active, the image is cyan. To thicken the images, we use the MorphoMNIST Castro et al. [8] package[2]. Operationally, we generate a base dataset for our experiments of size equivalent to the original MNIST dataset. That is, the training set has a size of 60K, and the test set has a size of 10K. Because we have access to the latents during the data generation procedure, we are able to train classifiers for each variable to identify their digit, color and thickness. We use a simple convolutional network architecture for each of these cases and achieve accuracy upwards of 95% in each case.

### F.2.3 Diffusion training details

We train two diffusion models during our sampling procedure, and we discuss each of them in turn.

To train a model to sample from $P(X, Y|W_1, W_2)$, we train a *single* diffusion model over the joint $(X, Y)$ distribution, i.e., 6 channels. We train a standard UNet architecture where we follow the

---

[2]https://github.com/dccastro/Morpho-MNIST/

conditioning scheme of classifier-free guidance. That is, we insert at every layer an embedding of the $W_1$ (image) and $W_2$ (2-dimensional discrete variable). To embed the $W_1$ image, we use the base of a 2-layer convolutional neural network for MNIST images, and to embed the $W_1$ we use a standard one-hot embedding for each of the variables. All three embeddings are concatenated and mixed through a 2-layer fully connected network to reach a final embedding dimension of 64. Batch sizes of 256 are used everywhere. Training is performed for 1000 epochs, which takes roughly 9 hours on 2 A100 GPU's. Sampling is performed using DDIM over 100 timesteps, with a conditioning weight of $w = 1$ (true conditional sampling) and noise $\sigma = 0.3$.

To train a model to sample $Y$ from the generated dataset $(W_2, X, Y)$, we follow an identical scheme. An option is to train a single diffusion model for each choice of $W_2$ in our synthetic dataset, however, we argue our schema still produces correct samples because: 1) $W_2$ can be arbitrarily chosen, and thus should not affect $Y$, 2) we argue that the model fidelity benefits from weight sharing across multiple choices of $W_2$, 3) the model is only ever called with a specified value of $W_2$ so we always condition on this $W_2$.

### F.2.4 Extra evaluations

As such, our evaluations include verifying the quality of the trained component neural networks and, in the case of MNIST, a surrogate ground truth for a discrete version of the dataset. In each experiment, for conditional sampling with high-dimensional data, we train diffusion models using classifier-free guidance [21]. For conditional sampling of categorical data, we train a classifier using cross-entropy loss. In addition to the evaluations presented in the main paper, we can further perform evaluations on the component models necessary to sample $P_X(Y)$.

$P(X, Y|W_1, W_2)$: We can evaluate the model approximating samples from $P(X, Y|W_1, W_2)$ on a deeper level than just visual inspection as provided in the main paper. In particular, assuming access to good classifiers that can predict the digit, color, and thickness of an MNIST image, we can compare properties of the generated images with respect to the ground truth in the discrete case. For example, assuming we have hyperparameter of random noise equal to $p$, we can compute the following quantities analytically on the discrete dataset as:

- $P[X_d = W_{2d}] = 1 - p + \frac{p}{10}$
- $P[X_{\cdot c} = W_{2c}] = 1 - p + \frac{p}{10}$
- $P[X_t = W_{1t}] = (1 - p + \frac{p}{3})^2 + \left(\frac{p}{3}\right)^2 * 2$
- $P[Y_d = X_d] = 1 - p + \frac{p}{10}$
- $P[Y_c = W_{1c}] = (1 - p + \frac{p}{6})^2 + big(\frac{p}{6})^2 * 5$
- $P[Y_t = X_t] = 1 - p + \frac{p}{3}$

where $V_d, V_c, V_t$ refer to the digit, color, and thickness attributes respectively. These calculations follow from two formulas. In a discrete distribution with support $S$ and $|S| = K$:

- $P[\eta_p(z, S) = z] = 1 - p + \frac{p}{K}$
- $P[\eta_p(z, S) = \eta_p(z, S)] = (1 - p + \frac{p}{K})^2 + \left(\frac{p}{K}\right)^2 * (K - 1)$

where in the second equation, it is assumed that $\eta_p(\cdot, \cdot)$ are two independent noising procedures.

Then to evaluate, we can 1) consider a large corpus of joint data, 2) run each of $W_1, X, Y$ through a classifier for digit, color, and thickness, 3) evaluate the empirical estimate of each desired probability. We present these results for the synthetic dataset $D_{synth}$ sampled from the diffusion model approximating $P(X, Y|W_1, W_2)$, a dataset $D_{orig}$ generated according to the data generation procedure, and $P_{true}$ the true analytical probabilities. These results are displayed in the Table 1.

Table 2: Color probability distribution of the sampled images. $P(y.c|x.c = red)$ is biased towards [R, G, B] while $P(y.c|\text{do}(x.c = red))$ is not.

| Predicted color probabilities | Red | Green | Blue | Yellow | Magenta | Cyan |
|---|---|---|---|---|---|---|
| Conditional model: $P(y.c|x.c = red)$ | 0.1889 | 0.4448 | 0.1612 | 0.1021 | 0.0232 | 0.0798 |
| Ours: $P(y.c|\text{do}(x.c = red))$ | 0.1278 | 0.2288 | 0.2097 | 0.1445 | 0.1177 | 0.1715 |

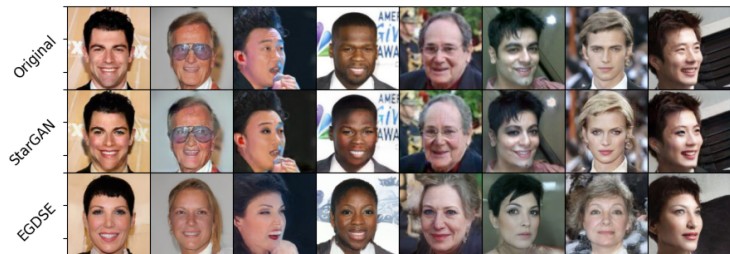

Figure 16: Samples from $P(I_2|\text{do}(Male = 0))$. Row 1: original $I_1$, Row 2: translated $I_2$ by StarGAN and Row3: translated $I_2$ by EGSDE.

Table 1: **Evaluations on the Napkin-MNIST generated dataset**. $V.d, V.c, V.t$ refer to the digit, color and thickness respectively of variable $V$. The first column is with respect to samples generated from diffusion model $\hat{P}(X, Y \mid W_1, W_2)$, Image Data is the dataset used to train $\hat{P}$, Discrete Data is the empirical distribution according to a discrete Napkin-MNIST, and the ground truth is analytically computed. Ideally all values should be equal across a row. While our synthetic dataset generated from $\hat{P}$ is not a perfect representation, it is quite close in all attributes except thickness. This is because the classifier for thickness has some inherent error in it, as evidenced by the mismatch between the base data and ground truth in the thickness rows.

| | $\hat{P}(Y, X \mid W_1, W_2)$ | Image Data | Discrete Data | Ground Truth |
|---|---|---|---|---|
| $P[X.d = W_2.d]$ | 0.931 | 0.895 | 0.909 | 0.910 |
| $P[X.c = W_2.c]$ | 0.964 | 0.950 | 0.950 | 0.950 |
| $P[X.t = W_1.t]$ | 0.683 | 0.776 | 0.879 | 0.873 |
| $P[Y.d = X.d]$ | 0.927 | 0.895 | 0.909 | 0.910 |
| $P[Y.c = W_1.c]$ | 0.847 | 0.841 | 0.841 | 0.842 |
| $P[Y.t = X.t]$ | 0.830 | 0.851 | 0.933 | 0.933 |

### F.2.5    MNIST baseline comparison

Here we provide some additional information about the baseline comparison we showed in Section 4.1. Although the implementation of the baseline methods is different, we considered the performance of each method after running them for 300 epochs (until good image quality is obtained). The NCM algorithm took around 50 hours to complete while our algorithm took approximately 16 hours to complete. We observe the probabilities in Table 2.

In the generated samples from the conditional model, the color of digit image $X$ and digit image $Y$ are correlated due to confounding through backdoor paths. For example, for a digit image $X$ with color as red, $Y$ takes a value from [Red, Green, Blue] with high probability. Thus, the conditional model does not have the ability to generate interventional samples. On the other hand, our algorithm generates samples from the interventional distribution $P(y|\text{do}(x))$ and the generated samples choose different colors:[Red, Green, Blue, Yellow, Magenta, Cyan] with almost the same probability. Therefore, our algorithm shows superior performance compared to baselines and illustrates the high-dimensional interventional sampling capability.

### F.3    CelebA experiment

Here we provide some additional information about the CelebA image to image translation experiment we showed in Section 4.2. We used a pre-trained classifier from this repository: `https://github.`

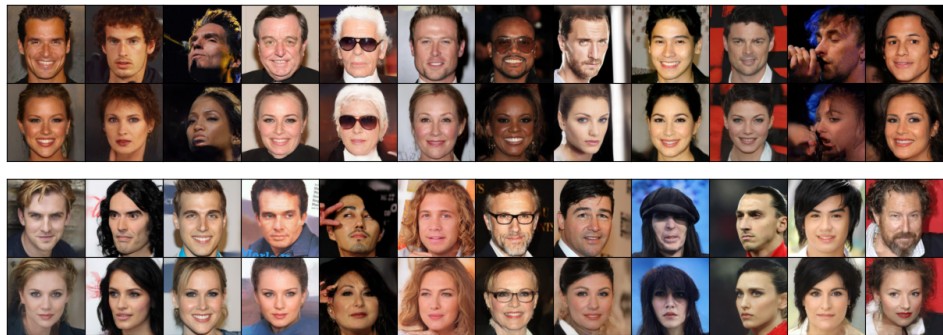

Figure 17: Multi-domain image translation by EGSDE. Images in rows 1 and 3 are the original images (male domain) and images in rows 2 and 4 are translated images (female domain). Table 3 shows, 29.16% of the total images are translated as a young person.

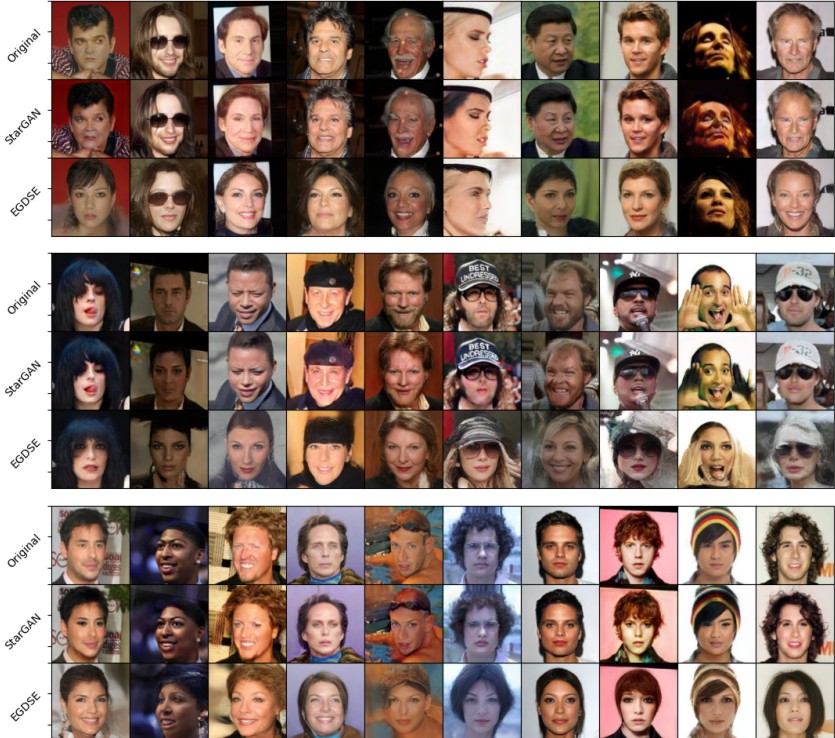

Figure 18: Performance comparison of multi-domain image translation between StarGAN and EGSDE.

com/clementapa/CelebFaces_Attributes_Classification. We use the pre-trained model of EGSDE from this repository:https://github.com/ML-GSAI/EGSDE We use the pre-trained model of StarGAN from this repository:https://github.com/yunjey/stargan. We generated 850 samples withe these models. Note that although EGSDE generates better quality (blur-free) images compared to StarGAN (blurry), it makes its samples more realistic by adding different non-causal attributes such as $Young$ or $Attractive$. As a result, it will generate $Female$ samples from a specific part of the image manifold. As a result, it might lack the ability to generate a different variety of images (for example: old females). The reason behind this is possibly the correlation in the dataset it is trained on. For example, in the CelebA dataset, we observe a high correlation between $Female$ and $Young$ while also a high correlation between $Male$ and $Old$. As a result, when EGSDE converts the images whether the Male is young or old, it converts them to a young female. On the other

Table 3: Additional attributes added (in percentage) in the translated images.

|  | WearingLipstick | HeavyMakeup | ArchedEyebrows | OvalFace | Attractive | Young |
|---|---|---|---|---|---|---|
| Added Attribute (%) | 87.5 | 79.16 | 66.66 | 54.16 | 37.5 | 29.16 |

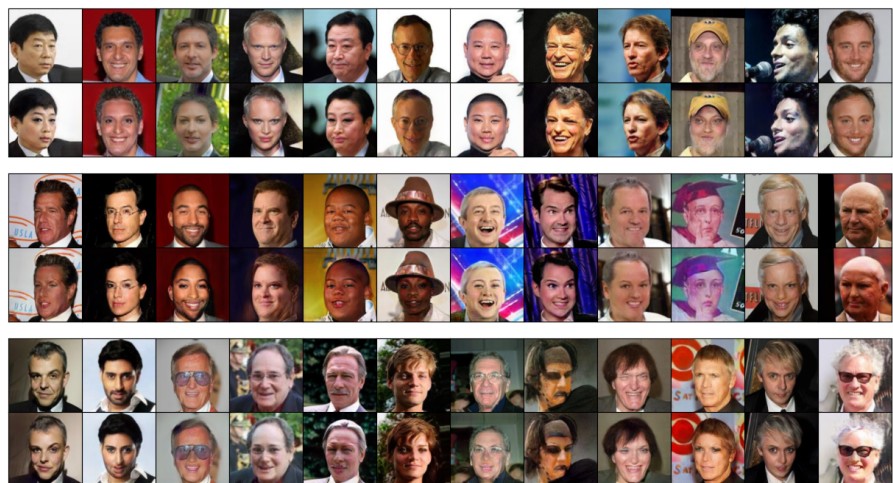

Figure 19: Samples from $P(I_2|Young = 1, \mathrm{do}(Male = 0))$ generated by StarGAN.

| Models | Distribution Matched | TVD ↓ |
|---|---|---|
| $M_N$ | $P(N)$ | 0.0595 |
| $M_E$ | $P(E|N, R)$ | 0.0512 |
| $M_A$ | $P(A|N, E, R)$ | 0.0451 |
| $M_L$ | $P(L|N, E, A)$ | 0.042 |
| $M_X$ | $P(X|N, E, A, L)$ | Pre-trained |

Figure 20: CXR conditional models in the sampling network.

hand, starGAN does not change noncausal attributes, but it also does not properly change the causal attributes as well.

**Baselines**: Existing algorithms such as Xia et al. [59], Chao et al. [11] do not utilize the causal effect expression of $P(A|\mathrm{do}(Male = 0))$, rather they train neural models for each variable in the causal graphs which will be costly in this scenario. Even if they utilize pre-trained models, they have to train models for the remaining variables to match the whole joint and perform sampling-based methods to evaluate $P(A|Young = 1, \mathrm{do}(Male = 0))$. Note that the ground truth if our considered attributes are causally related with an image of a Female is unknown. We assume that they are causal in the CelebA dataset based on expert knowledge [26].

#### F.3.1 Conditional image generation

In Figure 19, we show images generated from conditional interventional distribution.

### F.4 Explaining foundation model output for chest X-ray generation

#### F.4.1 Causal graph

Below we shortly describe the reasoning for our causal graph assumption.

**Nodes and Edges:**

- Prompt/Report ($R$) is variable that represents textual input. In CXR datasets such as MIMIC-CXR, it is generally doctor written text report indicating the existence of different attributes. In our experiment, we consider it as an input prompt feed by some user who is interested to understand how the prompt affects other attributes and the CXR image generation. We

assume that the LLM labeler can extract effusion ($E$) and atelectasis ($A$). Thus, we consider $R \to E$ and $R \to A$.

- Pneumonia($N$) is an infection that inflames the air sacs in one or both lungs [31].

- Pleural effusion ($E$) happens when fluid builds up in the space between the lung and the chest wall. This can occur for reasons such as pneumonia or complications from heart, liver, or kidney disease. Pleural effusion also involves fluid in the lung area. It is common in patients who develop pneumonia. At least 40-60% of patients with bacterial pneumonia will develop a pleural effusion of varying severity. Thus, we consider $N \to E$ .

- Atelectasis ($A$) is one of the most common breathing complications after surgery. Here, complication is a medical problem that occurs during a disease, or after a procedure or treatment [13]. External pulmonary compression by pleural fluid or air (i.e, pleural effusion, pneumothorax, etc.) may cause atelectasis [33]. Thus, we consider the edge $E \to A$. Also, various types of pneumonia, which is a lung infection, can cause atelectasis [32]. Therefore, we consider the edge $N \to A$.

- Lung opacity ($L$) is a lack of transparency, i.e., an opaque or non-transparent area on an x-ray/radiograph. [18]. We consider pneumonia, effusion and atelectasis to be causes for lung opacity. Thus we consider edges $N \to L, E \to L, A \to L$.

- Xray Image ($X$) represents different attributes such as atelectasis, effusion and lung opacity in visible radiography form. Thus, we consider edges $E \to X, A \to X, L \to X$.

- $N \leftrightarrow R$ represents the presence of a hidden confounder between pneumonia and the input prompt/report. We followed [9] to consider the hospital location as the unobserved variable from where the chest x-ray datasets where collected. They discuss that when we merge/mix different datasets and train models on that different types of biases are introduced. For example, in many scenarios, most patients are screened in certain health services and highly suspicious patients are derived to a different area. Another example of introducing bias is, when we aim to increase the number of controls or cases, we expand the dataset with samples coming from significantly different origins and labeled with unbalanced class identifiers. Since we utilized foundation models such as [16, 10] which are trained in multiple chest radiograph datasets, they are prone to above kind of bias. In some cases, to discriminate between cases and controls, these models might depend on the origins/location based details rather than finding actual features related to the disease. Thus, we consider that hospital locations have some effect on the text report on which the models were trained on and also how likely people were affected by pneumonia infection in those areas. Therefore, we consider the presence of a confounder between the text report ($R$) and pneumonia ($N$).

Note that all our results are based on the MIMIC-CXR dataset and our assumption on the causal graph. Thus, our results should not be used to make medical inferences without an expert opinion. In the real world, the domain-specific causal graph might change. Our algorithm can be applied on the domain specific causal graph and the dataset to obtain correct results.

## F.5   Covid X-Ray dataset

### F.5.1   Data preprocessing

Note that a full pipeline of data preparation is contained in `cxray/prep_cxray_dataset.sh` in the provided codebase. We start by downloading the corpus approximately 30K Covid X-Ray images[3]. Then we download Covid-19 labels and Pneumonia labels[4] and attach labels to each image. Then we convert each image to black-white one-channel images and rescale each to a size of $(128 \times 128)$ pixels. Finally, a random split of the 30K images is performed: keeping 20K to be used during training, and 10K to be used as validation images. Note that the labels come with a set of 400 test images, but 400 images is too small to be an effective test set (say for FID computations). We will be explicit about where we use each data set.

---

[3]https://www.kaggle.com/datasets/andyczhao/covidx-cxr2
[4]https://github.com/giocoal/CXR-ACGAN-chest-xray-generator-covid19-pneumonia

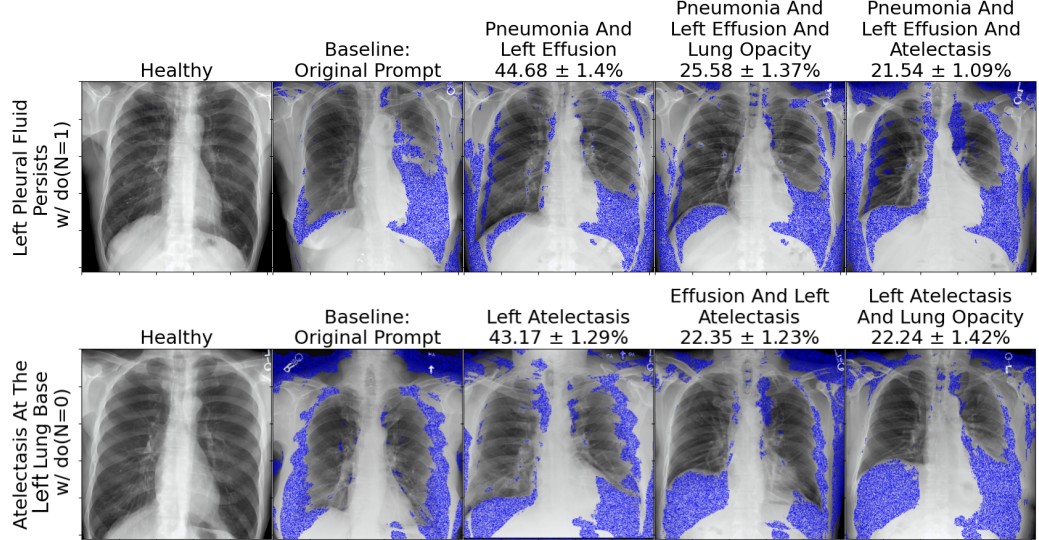

Figure 21: Two prompts, their CXR images, and inferred attributes with the likelihood to appear. Blue regions indicate changes compared to the healthy X-ray.

### F.5.2 Diffusion training details

We train a diffusion model to approximate $P(X|C)$. To do this, we use a standard UNet and classifier-free guidance scheme. We train for 100K training steps over the 20K-sized training set, using a batch size of 16. This takes roughly 10 hours on a single A100. The same classifier-free guidance parameters are used as in the NapkinMNIST diffusion training.

### F.5.3 Calibrated classifier training details

To train a classifier to sample from $P(N|C, X)$, we note that our inputs are a $(1, 128, 128)$ image and a binary variable. Our architecture is as follows: we create an embedding of $X$ by modifying the final linear layer of a ResNet18 to have output dimension 64 (vs 1000), and modify the input channels to be 1. We create an embedding for $N$ by using a standard embedding layer for binary variables, with embedding dimension 64. These embeddings are then concatenated and pushed through 3 fully connected layers with ReLU nonlinearities and dimension 128. A final fully-connected layer with 2 outputs is used to generate logits.

Training follows a standard supervised learning setup using cross entropy loss. We train for 100 epochs using a batch size of 256 and a standard warmup to decaying learning rate (see code for full details).

We note the deep literature suggesting that even though classifiers seek to learn $P(N|X, C)$, neural networks trained using cross entropy loss do not actually do a very good job of estimating this distribution. Training attains an accuracy of $91.2\%$ on the test set. Calibrated classification seeks to solve this problem by modifying the network in a way such that it more accurately reflects this distribution. We follow the standard approach using temperature scaling of the logits, where a temperature is learned over the validation set, using LBFGS with a learning rate of 0.0001 and a maximum of 10K iterations. This does not affect the test accuracy at all, but drastically improves the ECE and MCE reliability metrics. See Guo et al. [17] for a further discussion of temperature scaling.

### F.6 Covid X-Ray dataset

**Data generation:** Next we apply our algorithm to a real dataset using chest X-rays on COVID-19 patients. Specifically, we download a collection of chest X-rays (X) where each image has binary labels for the presence/absence of COVID-19 (C), and pneumonia (N) [57] [5]. We imbue the causal

---

[5]Labels are from https://github.com/giocoal/CXR-ACGAN-chest-xray-generator-covid19-pneumonia/

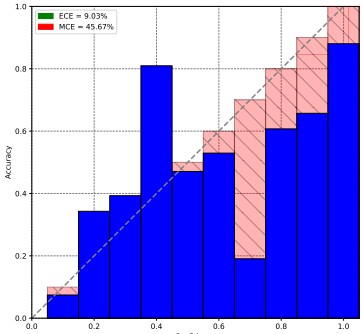 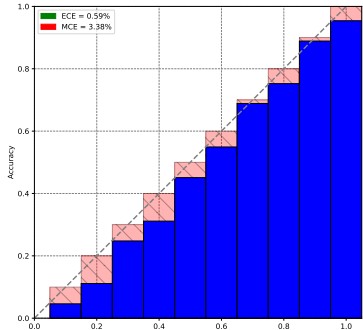

Figure 22: **Reliability plots for $P(N|C, X)$:**. Reliability plots which overlay accuracy versus classifier confidence. (Left) Reliability plot for $P(N|C, X)$ without calibration. (Right) Reliability plot for $P(N|C, X)$ with temperature scaling calibration applied.

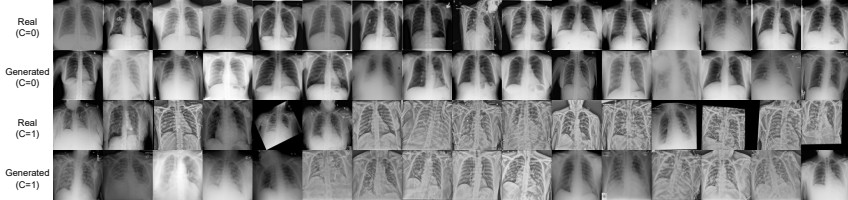

Figure 23: **Generated Covid XRay Images:** Generated chest XRay images from our diffusion model, separated by class and compared against real data.

Table 4: **(Left) Class-conditional FID scores for generated Covid XRAY images** (lower is better). Generated $C = c$, means we sample from the diffusion model conditioned on $c$. Real $(C = c)$ refers to a held out test set of approximately 5k images, partitioned based on $C$-value. Low values on the diagonal and high values on the off-diagonal imply we are sampling correctly from conditional distributions. **(Right) Evalution of Interventional Distribution $P_c(n)$.** We evaluate the distributions $P_c(n = 1)$ for three cases for the Covid-XRAY dataset. Diffusion uses a learned diffusion model for $P(x|c)$, No Diffusion samples $P(x|c)$ empirically from a held out validation set, and no latent evaluates the conditional query assuming no latent confounders in the causal model.

| FID ($\downarrow$) | Real: $C = 0$ | Real: $C = 1$ |
|---|---|---|
| Generated: $C = 0$ | 15.77 | 61.29 |
| Generated $C = 1$ | 101.76 | 23.34 |

| $P_c(n = 1)$ | $c = 0$ | $c = 1$ |
|---|---|---|
| Diffusion | 0.622 | 0.834 |
| No Diffusion | 0.623 | 0.860 |
| No Latent | 0.406 | 0.951 |

structure of the backdoor graph, where $C \to X$, $X \to N$, and there is a latent confounder affecting both $C$ and $N$ but not $X$. This may capture patient location, which might affect the chance of getting COVID-19, and quality of healthcare affecting the right diagnosis. Since medical devices are standardized, location would not affect the X-ray image given COVID-19. We are interested in the interventional query $P_c(n)$: the treatment effect of COVID-19 on the presence of pneumonia.

**Component Models:** Applying ID-GEN to this graph requires access to two conditional distributions: $P(x|c)$ and $P(n|x, c)$. Since $X$ is a high-dimensional image, we train a conditional diffusion model to approximate the former. Since $N$ is a binary variable, we train a classifier that accepts $X, C$ and returns a Bernoulli distribution for $N$. The generated sampling network operates by sampling an $X$ given the interventional $C$, and then sampling an auxiliary $C' \sim P(c')$ and feeding $X, C'$ to the classifier for $P(n|x, c)$, finally sampling from this distribution.

**Evaluation:** Again we do not have access to the ground truth. Instead, we focus on the evaluation of each component model, and we also perform an ablation on our diffusion model. We first evaluate the image quality of the diffusion model approximating $P(x|c)$. We evaluate the FID of generated

samples versus a held-out validation set of 10K X-ray images. When samples are generated with $C$ taken from the training distribution, we attain an FID score of 16.17. We then evaluate the conditional generation by comparing class-separated FID evaluations and display these results in Table 4 (left). The classifier estimating $P(n|x, c)$ has an accuracy of 91.9% over validation set. We note that we apply temperature scaling [17] to calibrate our classifier, where the temperature parameter is trained over a random half of the validation set. Temperature scaling does not change the accuracy, but it does vastly improve the reliability metrics; see Appendix.

Finally we evaluate the query of interest $P_c(n)$. Since we cannot evaluate the ground truth, we consider our evaluated $P_c(n)$ versus an ablated version where we replace the diffusion sampling mechanism with $\hat{P}(x|c)$, where we randomly select an X-ray image from the held-out validation set. We also consider the query $P_c(n)$ if there were no latent confounders in the graph, in which case, the interventional query $P_c(n)$ is equal to $P(n|c)$. We display the results in Table 4 (right).

# G  Pseudo-codes

---

**Algorithm 5** IDC-GEN $(\mathbf{Y}, \mathbf{X}, \mathbf{Z}, \mathcal{D}, G)$

---

1: **Input:** target $Y$, intervention $X$, conditioning set $Z$. training data $\mathcal{D}$, $G$.
2: **Output:** A DAG of trained model to sample from $P_{\mathbf{X}}(\mathbf{Y}|\mathbf{Z})$.
3: **if** $\exists \alpha \in Z$ such that $(Y \perp\!\!\!\perp \alpha | X, Z \setminus \{\alpha\})_{G_{\overline{X},\alpha}}$ **then**
4:     **return** IDC-GEN $(Y, X \cup \{\alpha\}, Z \setminus \{\alpha\}, \mathcal{D}, G)$
5: **else**
6:     $H_1=$ ID-GEN $(Y \cup Z, X, \mathcal{D}, \hat{\mathbf{X}} = \emptyset, \hat{G} = G)$
7:     $\mathcal{D}' \sim H_1(X)$
8:     $H_2 = \emptyset$
9:     Add node $(X, \emptyset)$ and $(Z, \emptyset)$ to $H_2$
10:     Let $M_Y$ be a model trained on $\{Y, X, Z\} \sim \mathcal{D}'$ such that $M_Y(X, Z) \sim P'(Y|X, Z)$ i.e., $M_Y(X, Z) \sim P_X(Y|Z)$
11:     Add node $(Y, M_Y)$ to $H_2$
12:     Add edge $X \rightarrow Y$ and $Z \rightarrow Y$ to $H_2$
13:     **return** $H_2$

---

---

**Algorithm 6** ID$(\mathbf{y}, \mathbf{x}, P, G)$

---

1: **Input:** $\mathbf{y}$, $\mathbf{x}$ value assignments , distribution $P$, $G$.
2: **Output:** Expression for $P_{\mathbf{x}}(\mathbf{y})$ in terms of $P$ or Fail$(F, F')$.
3: **if** $\mathbf{x} = \emptyset$ **then** {Step:1}
4:     Return $\sum_{\mathbf{v} \setminus \mathbf{y}} P(\mathbf{v})$
5: **if** $V \setminus An(\mathbf{Y})_G \neq \emptyset$ **then** {Step:2}
6:     Return ID$(\mathbf{y}, \mathbf{x} \cap An(\mathbf{Y})_G, \sum_{\mathbf{V} \setminus An(\mathbf{Y})_G} P, G_{An(\mathbf{Y})})$
7: Let $\mathbf{W} = (\mathbf{V} \setminus \mathbf{X}) \setminus An(\mathbf{Y})_{G_{\overline{\mathbf{x}}}}$ {Step:3}
8: **if** $\mathbf{W} \neq \emptyset$ **then**
9:     Return ID$(\mathbf{y}, \mathbf{x} \cup \mathbf{w}, P, G)$
10: **if** $C(G \setminus \mathbf{X}) = \{S_1, \ldots, S_k\}$ **then** {Step:4}
11:     Return $\sum_{\mathbf{v} \setminus (\mathbf{y} \cup \mathbf{x})} \prod_i ID(s_i, \mathbf{v} \setminus s_i, P, G)$
12: **if** $C(G \setminus \mathbf{X}) = \{S\}$ **then**
13:     **if** $C(G) = \{G\}$ **then** {Step:5}
14:         Return **FAIL**$(G, G \cap S)$
15:     **if** $S \in C(G)$ **then** {Step:6}
16:         Return $\sum_{s \setminus \mathbf{y}} \prod_{\{i|V_i \in S\}} P(v_i | v_\pi^{i-1})$
17:     **if** $\exists S'$ s.t. $S \subset S' \in C(G)$ **then** {Step:7}
18:         Return ID$(\mathbf{y}, \mathbf{x} \cap S', P = \prod_{\{i|V_i \in S'\}} P(V_i | V_\pi^{(i-1)} \cap S', v_\pi^{(i-1)} \setminus S'), G_{S'})$

---

---

**Algorithm 7** IDC($Y, X, Z, P, G$)

---

1: **Input:** $x, y, z$ value assignments, $P$ a probability distribution, $G$ a causal diagram (an I-map of $P$).
2: **Output:** Expression for $P_X(Y|Z)$ in terms of $P$ or Fail(F, F').
3: **if** $\exists \alpha \in Z$ such that $(Y \perp\!\!\!\perp \alpha | X, Z \setminus \{\alpha\})_{G_{\overline{X}, \underline{\alpha}}}$ **then**
4:     **return** IDC($Y, X \cup \{\alpha\}, Z \setminus \{\alpha\}, \mathcal{D}, G$)
5: **else**
6:     let $P' = $ ID($\mathbf{y} \cup \mathbf{z}, \mathbf{x}, P, G$)
7:     return $P' / \sum_{\mathbf{y}} P'$

---

