# OpenReview forum: "Conditional Generative Models are Sufficient to Sample from Any Causal  Effect Estimand"
_NeurIPS.cc/2024/Conference — NeurIPS 2024 poster_

### Official Review · Reviewer_xD7P · 2024-06-25

**Soundness:** 3
**Presentation:** 1
**Contribution:** 2
**Rating:** 5
**Confidence:** 2

**Summary:**

The paper leverages state-of-the-art conditional generative models and algorithms from causal do calculus to perform "approximately correct" high-dimensional interventional sampling. Their contribution is ID-GEN, a recursive algorithm that uses diffusion models (among other generative models) to sample from any identifiable causal effect estimand in high-dimensional settings. The efficacy of the method is demonstrated on three diverse datasets: Colored MNIST, CelebA, and MIMIC-CXR.

**Strengths:**

- The paper introduces ID-GEN, a novel algorithm that integrates diffusion models with causal inference for high-dimensional interventional sampling.
- ID-GEN creatively exploits the causal structure via the known recursive algorithm for sampling in complex causal scenarios, particularly with latent confounders.
- The algorithm is applied in three applications across diverse datasets (Colored MNIST, CelebA, MIMIC-CXR).

**Weaknesses:**

- The paper is objectively hard to read. Many important graphical elements that should be paired with the text in the main paper are delayed to supplements. This is notably problematic for Example 4.1, where one would expect the full example to be self-contained in the main text.
- The contributions are stated in the introduction, but it still seems hard to understand if the proposed method is "just" an implementation of the ID algorithm, replacing probability tables by samples from diffusion models. I appreciate that this is hard already, but it has much lower novelty then proposing a new recursive algorithm. This should be well explained in the manuscript.
- The paper does not discuss the implications of the proposed algorithm. Is there any way to extend this to symbolic calculus, or to probabilistic programming? What are the obstacles for moving towards automatic causal inference with images (which would be a super exciting prospect).

**Questions:**

- Can you clarify whether the proposed ID-GEN algorithm is primarily an implementation of the existing ID algorithm with diffusion models substituting for probability tables, or does it introduce fundamentally new recursive methodologies? How does this distinction affect the perceived novelty of your contribution?
- What are the potential extensions of your algorithm to areas like symbolic calculus or probabilistic programming?
- Are there any specific obstacles that need to be overcome to advance towards automatic causal inference using high-dimensional data such as images? How feasible is this prospect in the near future?

---

> ### Author Rebuttal · Authors · 2024-08-07
>
> We thank the reviewer for their valuable effort. We are happy that they acknowledged our algorithm as novel and sound. Below we
> address their concerns.
>
>
> > ..., Obstacles for moving towards automatic causal inference with images?.
> > How feasible is this prospect in the near future?
>
> We interpret the reviewer's mention of "automatic causal inference" as the process of
> automating every step of causal inference where the inputs are raw data and assumptions/constraints and the outputs are
> high-dimensional
> interventional samples/images. Below we share the steps and challenges for each step.
>
> Step 1, causal variables: We detect the causal variables in the raw data (challenging in high-dimension).
>
> Step 2, causal graph: We discover the causal relationships among the causal variables obtained from Step 1.
> It is hard to obtain fully specified causal graphs specially when high-dim variables are involved.
> Generally, we obtain partial graphs with uncertain edges and/or uncertain directions.
>
> Step 3, causal inference: We learn the underlying causal mechanisms that generate each variable from
> the causal graph parents obtained from Step 2. Finally, we generate interventional samples using the learned mechanisms
> and report samples as output. Even though some promising results have been shown for fully-specified graphs,
> high-dimensional causal inference is
> still under-explored in presence of unobserved shared parents (confounders). Our proposed ID-GEN offers a solution for
> this problem.
>
> We hope our contribution becomes a useful piece in this challenging process.
>
>
> > ... extend this to symbolic calculus
>
> In Pearl's causality, we found do-calculus [1] closest to symbolic calculus [2].
> In [2], Pearl mentioned that "Polynomial tests are now available for deciding when $P(x_i|do(x_j)$ is identifiable,...,
> in terms of observed quantities.
> These tests and derivations are based on a **symbolic calculus**,..., in which interventions, side by side with
> observations, are given explicit notation,
> and are permitted to transform probability expressions."
>
> The 3 do-calculus rules express an interventional distribution in terms of observational distribution. The ID
> algorithm (Shpitser et al [45]) we considered
> actually applies these 3 rules systematically along its recursive steps. Since we follow ID's recursive steps, we are
> implicitly following the do-calculus rules.
>
> [1] Pearl, J. 2000. Causality. Cambridge University Press, 2nd edition\
> [2] Pearl, Judea. "A causal calculus for statistical research." AISTAT (1996): 23-33
>
>
> > or to probabilistic programming?
>
> We found a relevant work [3] that implements Bayesian causal inference based on probabilistic programming using MiniStan
> and Gen programming languages.
> According to [3], one can use these programs to estimate the parameters of a model from data that represents the
> conditional distributions of variables. As ID-GEN is model agnostic,
> one could explore the connection between ID-GEN and the probabilistic programming approach.
>
> [3] Witty, Sam, et al. "Bayesian causal inference via probabilistic program synthesis." arXiv (2019).
>
>
> > ..., for Example 4.1, ... ,expect the full example to be self-contained in the main text.
>
> We will to utilize the additional space of the camera-ready version to accommodate Example 4.1 if our paper gets accepted.
>
>
> > ...if the proposed method is "just" an implementation of the ID algorithm, replacing probability tables by
> > samples from diffusion models.
>
> We apologize for such an impression. Our algorithm definitely offers novel methodologies compared to the mentioned
> implementation. Let us imagine an algorithm, AlgX that follows ID step-by-step and trains a
> diffusion model for every conditional probability table it encounters.
> Below we discuss the challenges it faces.
>
> $X \rightarrow W_1 $\
> $\updownarrow~$ $\swarrow$ $~~\updownarrow$\
> $W_2 \rightarrow Y$
>
> Given the query, $P_{x}(y)$, ID factorizes as:
> $$P_{x}(y) = \sum_{w_1,w_2} P_{x,w_1}(w_2) \cdot P_{x,w_2}(w_1,y)$$
> where each factor of this product is an interventional distribution.
>
> i) We have only observational samples as training data. If we train the diffusion models on our dataset, it will learn
> conditional distributions. However, if we want to sample from the above factors we need interventional samples for model
> training.
> AlgX doesn't know how to obtain such interventional samples. \
> -We recursively solve each factor while generating interventional training data during the top-down recursion (**Alg 1:
> step 4+Alg 1: step 7+Alg 4:Update**).
>
> ii) Suppose, AlgX somehow can sample from each factor. In which order should it
> sample?
> If it samples $\\{W_2\\}
> \sim P_{x,w_1}(w_2)$ it needs $W_1$ as input, which has to be sampled from $P_{x,w_2}(w_1,y)$. But to sample $\\{W_1,
> Y\\}
> \sim P_{x,w_2}(w_1,y)$, it needs $W_2$ as input which has to be sampled from $P_{x,w_1}(w_2)$.
> Therefore, it will reach a deadlock situation and fail to sample all $W_1, W_2, Y$ consistently.\
> -We disintegrate the factors after the recursion ends, into single variable models and build a sampling network
> connecting them (**Alg 3: MergeNetwork**).
>
> iii) It is unclear how AlgX will mimic the product and the marginalization of the factors for the generated
> high-dimensional samples (ex: image generation).\
> -We perform ancestral sampling on our built sampling network and drop the marginalized ones at the end.
>
> iv) ID goes to step 7 for specific queries such as $P_{w_1, w_2, x}(y)$. It iterates over all values of $X, W_2$ and
> performs $do(X, W_2)$ to update its probability distribution parameter $P(V)$ as
> $$P'(V)= P(W_1|X)P(Y|W_1, x, w_2)$$ for the next recursive calls.
> AlgX equipped with diffusion models does not give any hint about how to mimic this step for high-dimensional sampling.\
> -We generate $do(X, W_2)$ interventional training data with **Alg 4:Update**.
>
> We are more than willing to have further discussion. We would highly appreciate it if the reviewer reconsiders a stronger
> the score for our paper.

---

> > ### Comment · Reviewer_xD7P · 2024-08-13
> >
> > I would like to thank the authors for their detailed response. I will consider changing my score while discussing with other reviewers.

---

> > > ### Author Response · Authors · 2024-08-13
> > >
> > > We cordially thank you for reading our responses. We are also happy to know that  you would consider changing your score after the discussion with other reviewers.

---

### Official Review · Reviewer_fUEc · 2024-06-30

**Soundness:** 4
**Presentation:** 3
**Contribution:** 2
**Rating:** 5
**Confidence:** 3

**Summary:**

This paper proposes an algorithm for sampling from intervention distributions under known causal models with hidden confounders, using conditional generative models.
This allows nonparametric distributional estimation of high-dimensional outcomes such as X-ray image generation, unlike existing methods.
The proposed method combines the existing ID algorithm with a generative model and inherits the ID algorithm's theoretical guarantees. That is, non-identifiable quantities are indicated to be non-identifiable, while all identifiable quantities can be estimated.

**Strengths:**

* They shed light on the new problem setting of causal simulation for outcomes with high-dimensional and multimodal distributions, such as image generation. This could open up new applications if well justified. Such a problem setting requires a very different approach to point estimation of expectations for low-dimensional, unimodal distributions.
* The theoretical background is solid and well-explained. The method is proved to be able to estimate all identifiable quantities and otherwise outputs "unidentifiable."

**Weaknesses:**

[W1] The motivation for high-dimensional distribution estimation is weak. For example, it does not seem very meaningful to me to generate synthetic X-ray images.

[W2] In particular, is it important in cases where there are bidirectional edges due to the presence of hidden confounding factors, but where the causal orientations are all identified among variables? A clear comparison with similar methods would be beneficial for readers, e.g., comparison in assumptions and targets (e.g., parametric/nonparametric, latent confounder, distributional estimation, etc.).

[W3] The base procedure seems to come from existing methods, such as the ID algorithm, and they just combine it with a generative model.

**Questions:**

[Q1] Related to W2, is the proposed method a non-trivial novelty in situations where there are no unobserved confounding variables, i.e. no bidirectional edges?

[Q2] Related to W3, are there any non-trivial points in theoretical guarantees when the generative model is combined with the ID algorithm?

**Limitations:**

The method is limited to the cases where the causal direction is known.

---

> ### Author Rebuttal · Authors · 2024-08-07
>
> We thank the reviewer for their efforts and useful feedback. We are happy that they found our work important and our
> theoretical background solid. Below we address their concerns.
>
>
> >Is it important in cases where there are bidirectional edges ..., but the causal orientations are all
> > identified ...?
>
> Our considered setup is well known as the semi-Markovian causal model in the literature.
> However, all causal orientations can not be identified in general, and we obtain Markov equivalent graphs. There exist
> some recent works [1] that can estimate specific causal effects in such scenarios.
> Thus, one could draw connections between [1] and ID-GEN to perform high-dimensional causal inference in the presence of
> uncertain causal orientation.
>
> [1] Jaber et al. "Identification of conditional causal effects under Markov equivalence." NeurIPS 2019.
>
> > Is the proposed method a non-trivial novelty in situations where there are no ... bidirectional edges?
>
> Even though ID-GEN will work for graphs with no bidirectional edges, it is a much easier problem and can be solved by existing works such as Kocaoglu et al [26]. The problem becomes non-trivial when bi-directed edges are considered.
>
>
> >The motivation for high-dimensional distribution estimation is weak. ... not seem very meaningful to generate
> > synthetic X-ray images.
>
> We would like to humbly point out that X-ray image generation is not the main goal of our algorithm but part of
> answering a causal question.
> We borrowed this application from Chambon et [10] who developed the generative imaging models to deal with the lack of
> high-quality, annotated medical imaging datasets.
> Our main purpose is to illustrate the type of high-dimensional causal query we can answer.
>
>
>
> > The base procedure seems to come from existing methods, such as the ID algorithm, and they just combine it with a
> > generative model.
>
> We respectfully disagree with the reviewer that we just combine the ID algorithm with generative models.
> To explain this point, let us imagine an algorithm, AlgX that follows ID step-by-step and trains a
> generative model for every conditional probability table it encounters.
> Below we discuss the challenges it faces.
>
> $X \rightarrow W_1 $\
> $\updownarrow~$ $\swarrow$ $~~\updownarrow$\
> $W_2 \rightarrow Y$
>
> Given the query, $P_{x}(y)$, ID factorizes as:
> $$P_{x}(y) = \sum_{w_1,w_2} P_{x,w_1}(w_2) \cdot P_{x,w_2}(w_1,y)$$
>
> where each factor of this product is an interventional distribution.
>
> i) We have only observational samples as training data. If we train the generative models on our dataset, it will learn
> conditional distributions. However, if we want to sample from the above factors we need interventional samples for model
> training.
> AlgX doesn't know how to obtain such interventional samples. \
> -We recursively solve each factor while generating interventional training data during the top-down recursion (**Alg 1:
> step 4+Alg 1: step 7+Alg 4:Update**).
>
> ii) Suppose, AlgX somehow can sample from each factor. In which order should it
> sample?
> If it samples $\\{W_2\\}
> \sim P_{x,w_1}(w_2)$ it needs $W_1$ as input, which has to be sampled from $P_{x,w_2}(w_1,y)$. But to sample $\\{W_1,
> Y\\}
> \sim P_{x,w_2}(w_1,y)$, it needs $W_2$ as input which has to be sampled from $P_{x,w_1}(w_2)$.
> Therefore, it will reach a deadlock situation and fail to sample all $W_1, W_2, Y$ consistently.\
> -We disintegrate the factors after the recursion ends, into single variable models and build a sampling network
> connecting them (**Alg 3: MergeNetwork**).
>
> iii) It is unclear how AlgX will mimic the product and the marginalization of the factors for the generated
> high-dimensional samples (ex: image generation).\
> -We perform ancestral sampling on our built sampling network and drop the marginalized ones at the end.
>
> iv) ID goes to step 7 for specific queries such as $P_{w_1, w_2, x}(y)$. It iterates over all values of $X, W_2$ and
> performs $do(X, W_2)$ to update its probability distribution parameter $P(V)$ as
> $$P'(V)= P(W_1|X)P(Y|W_1, x, w_2)$$ for the next recursive calls.
> AlgX equipped with generative models does not give any hint about how to mimic this step for high-dimensional sampling.\
> -We generate $do(X, W_2)$ interventional training data with **Alg 4:Update**.
>
>
>
>
> > ...Any non-trivial points in theoretical guarantees when the generative model is combined with the ID
> > algorithm?
>
>
> We believe that the following theoretical contributions are nontrivial, and allow us to establish our novelty.
>
> **Lemma D.11**: ID carries a probability table as a recursion parameter whereas we carry an interventional dataset.
> Even though we follow ID's recursive trace, to deal with the interventional dataset, we have two extra parameters:
> intervened variables at step 7s: $\hat{X}$
> and causal graph $\hat{G}$ containing $\hat{X}$. This lemma ensures that our interaction with these extra parameters
> will not deviate us from the trace of ID.
>
> **Lemma D.14**: We perform on our training dataset and update them to the interventional dataset at step 7s whereas ID
> manipulates its probability table from observational distribution to interventional distribution.
> Thus, we show a guarantee that our training dataset represents the corresponding probability distribution in
> the ID algorithm at each recursion level.
>
> **Lemma D.20**. shows that every edge in the sampling graph adheres to the topological ordering $\pi$ of the original
> graph. This ensures that our proposed concept of
> a sampling network containing the trained models is not invalid and samples consistently **in an acyclic manner.**
>
> **Lemma D.21**: After training the conditional models from our recursive steps, we connect them to build a sampling
> network before performing the sampling. Here, we prove
> that the sampling network consists of trained models can sample from the expected joint distribution.
>
>
> We are more than willing to have further discussion. We would highly appreciate it if the reviewer reconsiders a stronger
> the score for our paper.

---

> > ### Author Response · Authors · 2024-08-12
> >
> > Dear Reviewer fUEc,
> >
> > We thank you for your invested valuable time to provide constructive feedback for our paper.
> > We were wondering if we have addressed all your concerns including the theoretical and technical novelties of our algorithm.
> > If you have any additional questions in your mind, please let us know. We would be more than happy to answer them.
> >
> > Also, we have used some latex notations in our responses. Please refresh the page if they do not render properly.
> >
> > Thank you.

---

> ### Comment · Reviewer_fUEc · 2024-08-14
>
> Thank you for your response.
>
> For [W1], the lack of motivating examples has not been resolved.
>
> For [W2], I understand the problem of unidentified causal direction would be another direction, and the proposed method can be extended.
>
> For [W3], I understand that there would be some issues in a straightforward extension of the ID algorithm to generative models.
>
> Based on the resolution of these concerns, the score is slightly increased.

---

> > ### Author Response · Authors · 2024-08-14
> >
> > Dear Reviewer fUEc,
> >
> > We cordially thank you for your response and highly appreciate that you raised your score.
> >
> > >For [W1], the lack of motivating examples has not been resolved.
> >
> >
> > We apologize that we misunderstood your concern about the motivational example. To address this concern, we will illustrate the scope of high-dimensional causal inference in another real-world scenario involving brain MRI [1] in our introduction section. Here the goal would be to generate samples from the interventional distribution involving MRI images with attributes such as age, sex, brain volume and ventricle volume [2].
> >
> > Our algorithm, ID-GEN would play an important role in estimating causal effects when we consider the presence of confounders among variables, as existing works such as [2] does not consider confounders in the system. We will explain this motivational example in detail.
> >
> >
> > [1] Sudlow, Cathie, et al. "UK biobank: an open access resource for identifying the causes of a wide range of complex diseases of middle and old age." PLoS medicine 12.3 (2015): e1001779.\
> > [2] Ribeiro, Fabio De Sousa, et al. "High fidelity image counterfactuals with probabilistic causal models." arXiv preprint arXiv:2306.15764 (2023).

---

### Official Review · Reviewer_5UTo · 2024-07-11

**Soundness:** 3
**Presentation:** 3
**Contribution:** 3
**Rating:** 7
**Confidence:** 3

**Summary:**

This paper studies the problem of sampling from an interventional distribution of high-dimensional data, where the causal relationships are described by a acyclic directed mixed graph (ADMG). Motivated by the ID algorithm that provides a recursive way of identifying any causal effect from a conditional probability table, the authors propose ID-GEN that follows the recursive steps of ID but instead trains deep generative models to fit the conditional distributions. The final sampling model is then obtained by connecting all the trained networks together in some suitable way. The authors prove that ID-GEN is sound and complete, and run extensive experiments on both synthetic and real dataset to demonstrate the effectiveness of their approach.

**Strengths:**

1. This paper studies the problem of sampling from an interventional distribution, an arguably important problem with broad applications. Current approaches, as the authors point out, are either restricted to simple causal graphs or face computational challenges.

2. Most parts of the paper are clearly written, and sufficient explanations are provided for the key steps in the ID-GEN algorithm. Also, simple examples are provided that help with the understanding of the paper. The paper is also well-organized and the authors put most complicated details into the Appendix.

3. The authors conduct extensive experiments to demonstrate the superior performance of their model, by comparing with other sampling models proposed by previous works.

**Weaknesses:**

I don't think this paper has obvious weaknesses. One thing that the authors may wish to improve is that the notations are a litle bit complex; and it would be better to more often remind the authors of their meanings.

**Questions:**

1. In some identification formula e.g. Eq.(1) in the paper, the probability on the denominator might be small. To what extent would this affect the stablity of the proposed algorithm.

2. Can your algorithm be straightforwardly adapted to the case of imperfect interventions i.e. some conditional probabilities in the structural causal model are modified, but no causal edges are removed?

---

> ### Author Rebuttal · Authors · 2024-08-07
>
> We thank the reviewer for their effort on our paper. We are really happy that they found our work broadly applicable,
> our paper well-written and our experiments extensive. Below, we address their concerns.
>
>
> >In some identification formulas e.g. Eq.(1) in the paper, the probability on the denominator might be small. To what
> > extent would this affect the stability of the proposed algorithm.
>
>
> A fraction in ID-GEN is generally created when we traverse the recursion and reach any point where we have to estimate
> a conditional distribution $P(Y|Z)$. This $P(Y|Z)$ can be written as $\frac{P(Y,Z)}{P(Z)}$ and estimated from the joint distribution.
> Thus, in such an expression if the denominator $P(Z=z)\sim 0$, it implies that we have a small number of samples for $Z=z$. This will result in a high variance for the estimation of $P(Y|Z=z)$.
>
> ID-GEN trains generative models to learn and sample from $P(Y|Z)$.
> Thus, for $Z=z$, the trained models might have low prediction accuracy or low image generation quality when sampled from the final interventional distribution.
> However, for other values $Z=z'$ where $P(z')$ is not close to zero, the trained models in ID-GEN shows good empirical performance (for example: Figure 2)
> and does not affect the over all stability of the algorithm.
>
>
>
>
> > Can your algorithm be straightforwardly adapted to the case of imperfect interventions i.e. some conditional
> > probabilities in the structural causal model are modified, but no causal edges are removed?
>
> We thank the reviewer for such an interesting question.
> For causal effect estimation in high-dimension, we follow the ID algorithm [1] to consider hard intervention $do(X=x)$, i.e., fixing a specific value to the intervened variable,
> and assume that the rest of the conditional distribution stays unchanged.
>
> However, there are recent algorithms such as [2] that
> deals with **imperfect/soft intervention** where the underlying mechanism of intervention $X$ changes instead of fixing to a specific value.
> Note that, these algorithms deal with low-dimensional variables.
>
> Since [2] also utilizes the identification algorithm, in principle our algorithm could be adapted for such cases. However, it might require
> more careful handling, and we will address it as our future works.
>
>
> [1] Shpitser et al. Complete identification methods for the causal hierarchy. Journal of Machine Learning Research, 9:1941–1979, 2008.\
> [2] Correa et al "A calculus for stochastic interventions: Causal effect identification and surrogate experiments." AAAI, 2020.

---

### Official Review · Reviewer_6o9W · 2024-07-15

**Soundness:** 3
**Presentation:** 3
**Contribution:** 4
**Rating:** 5
**Confidence:** 2

**Summary:**

This paper provides an algorithm for sampling from a causal interventional distribution using conditional generative models, building on Shpitser and Pearl's ID algorithm. They discuss how their algorithm, ID-GEN, can sample from any identifiable distribution given a causal graph, and handles the presence of unobserved confounders (when identifiable). Empirically, they demonstrate their method can work for measurement, evaluation and interpretability purposes in the challenging setting where both the target and treatment are high dimensional e.g. images.

**Strengths:**

- interesting work, the case of high-dimensional variables in a causal graph is a super important and under-discussed one
- thorough theoretical treatment of the extension of the ID algorithm
- experiments show a nice range of usages of the suggested ID-GEN approach

**Weaknesses:**

- I frankly got a bit lost in a few key parts of Section 3. I got the main ideas (I think) but missed a bunch of nuance. Some spots were: Example 4.1 (I don’t understand why ID fails in this case but ID-GEN succeeds - specifically the importance of merging is a bit lost on me), Step 4 of ID-GEN (again, merging), and Step 7 of ID-GEN (I think the logic around how training data is sampled, used and modified wrt the graph needs to be explained more clearly)
- Step 1 of ID-GEN confuses me - I don’t see why we can’t just learn a model of P(y) directly in this case? Also the 2nd equality in 203 doesn’t make sense to me - how is the sum over values of v equal to P(y)?
- in each experimental section I find I have at least a medium-sized point of confusion around the setup or evaluation - more care should be taken to explain empirical setup + results overall
- In 5.1, the authors state that U_color makes W1 and Y correlated by color - however, X contains color information and is a direct ancestor of Y, so this unobserved confounding seems trivial
- in 5.1, it seems like a better metric than a classifier (which may be unreliable and as you note isn’t useful for all possible images) would be something based specifically on the RGB values of the pixels themselves
- in 5.2, I don’t quite see why Young & Male have unobserved confounding - are they not fully determined by the shared + observed parent I_1?
- in 5.3, I don’t understand why the report is a causal ancestor in the graph - isn’t it generated upon viewing the X-ray?
- in 5.3, I think the setup with the labeller can be made clearer - how good is this labeller? How is it structured? Additionally, is the bottom row intended to be a success or failure examples? (label says it should be right lung base but all inferences name the left lung)


Smaller points:
- L115: are unobserved confounders only allowed to affect 2 variables in this framework? Is that more limiting than general SCMs?

**Questions:**

- would be great to see clarification throughout Section 3, particularly in the highlighted areas and around merging and training data sampling
- experimental setups all need clarifications
- generally assuming I understood what's happening better I'd be happy to increase my score

---

> ### Author Rebuttal · Authors · 2024-08-01
>
> We thank the reviewer for their valuable efforts on our paper.
> We are happy to receive their appreciation for our theoretical contribution and experimental setup.
> Below we address their concerns.
>
> > Step 1 of ID-GEN ... why we can't just learn a model of P(y) directly? ..., how is the sum over values of v equal to
> > P(y)?
>
> The reviewer is correct that if the intervention set is empty, we can learn a single model $M$ that can sample the
> variable set $\mathbf{Y}$ from
> $P(\mathbf{Y})$, specially if all variables in $\mathbf{Y}$ are low dimensional. However, we consider that $\mathbf{Y}$
> is allowed to contain both low and high-dimensional variables.
> Rahman and Kocaoglu [40] show that it is non-trivial to achieve convergence if we directly attempt to match such joint
> distributions $P(\mathbf{Y})$.
>
> In line 203, we stated that $P(y)= \sum_{\mathbf{v} \setminus y} P(\mathbf{v})$. Here, $\mathbf{V}$ is the set of all
> variables and $Y$ is the target variable.
> Please note that we do not sum over $\mathbf{V}$ but rather over $\mathbf{V} \setminus Y$.
> Intuitively, we first consider all variables and then drop (sum) the unnecessary ones.
>
>
> > more care should be taken to explain empirical setup + results overall
>
> We apologize for the confusion and will bring the Napkin-MNIST, CelebA, and Xray generation empirical details from the
> appendix to the main paper.
>
>
> > In 5.1, the authors state that U\_makes W1 and Y correlated by color - however, X contains color information and
> > is a direct ancestor of Y, so this unobserved confounding seems trivial.
>
>
> We humbly point out that even though image X (takes R and G) is a direct ancestor of image Y (takes all 6 colors),
> $Y$ **only inherits the digit property** from $X$ but correlates the color property with image $W_1$. This color
> correlation between $W_1$ and $Y$ is created by U\_color.
>
>
> > It seems like a better metric than a classifier, ..., would be something based specifically on the RGB values of the
> > pixels themselves
>
> As we are not doing counterfactual generation, we should not match against any specific image.
> Our goal is to generate images that are correct at the population level.
> For example, in Fig 2 (napkin-MNIST), if we generate 1000 images from the interventional distribution,
> there should be around 167 images for each of the 6 colors (Pr=0.17). Since the output is not deterministic, we can not
> evaluate them with pixel-wise RGB values.
>
>
> > in 5.2, ..., why Young & Male have unobserved confounding - are they not fully determined by the shared + observed
> > parent I_1?
>
> In the CelebA dataset, men are more likely to be old (correlation coeff=0.4, (Shen et al [44])). As a result, any
> classifier trained on the CelebA, will
> not have 100% accuracy at the population level and might have some bias toward predicting young-male images
> as [Old, Male].
>
>
> > in 5.3, ..., why is the report a causal ancestor in the graph - isn't it generated upon viewing the X-ray?
>
> We followed the setup discussed by Chambon et al [10] where they generate synthetic X-ray images from prescription
> reports
> with a vision-language foundation model. We aim to make the model's predictions interpretable and invariant with domain
> shifts.
> Thus, we follow the causal mechanisms of the environment the model will be deployed in to build the causal graph.
> Therefore, we consider the same initial input and final output as the model, having the report variable as an ancestor
> of the X-ray image.
>
>
> > ..., the setup with the labeller can be made clearer - how good is this labeller?
>
> We skipped details on the X-ray report labeler as it is taken from Gu et al [16] and not our contribution.
> However, we will add more details. In short, it is a BERT architecture trained on pseudo-labels created by GPT-4
> to classify findings in CXR reports. It achieves the best or the second-best F1 score compared to its baselines.
>
>
> > The label says it should be the right lung base but all inferences name the left lung.
>
> We apologize for the mistake and have fixed it. The label should have said, "Atelectasis at the **left** lung base". Top
> and bottom, both rows are simulations of how ID-GEN is more useful compared
> to the direct application of baseline LLM.
>
> > L115: are unobserved confounders only allowed to affect 2 variables in this framework? Is that more limiting than
> > general SCMs?
>
> We consider the commonly used semi-Markovian SCM where unobserved confounders only affect two variables. A more relaxed
> assumption is when hidden confounders
> can affect any number of variables, known as the non-Markovian SCM.
>
>
> > Step 4 of ID-GEN (again, merging), and Step 7 of ID-GEN,...,needs to be explained more clearly
>
> We apologize some parts of our paper were hard to understand. We will elaborate the definitions in Section 3 and add
> more discussion on the merging in Step 4 and the parameter
> updates in Step 7.
>
>
>
> > Example 4.1 ... I don’t understand why ID fails ... but ID-GEN succeeds. ...
>
> ID can't deal with high-dimensional variables. However, if we imagine an algorithm that follows ID step-by-step while
> training a generative model for every factor it encounters,
> that approach might fail at step 4. Step 4 factorizes the query into multiple factors and this algorithm would not know
> which factor to learn and sample first with the generative models.
> In some cases, there exists no such sampling order of the factors at all (cyclic issue).
>
> ID-GEN avoids sampling factor by factor, and first trains required models for each of them, considering all possible
> input values.
> After the recursion ends, it disintegrates the variables in the factors and treats their model as individual nodes.
> A sampling network is built connecting these model nodes considering their input-outputs. This network now enables
> consistent interventional sampling.
>
> We will be happy to provide further clarification for any concerns of the reviewer. We would highly appreciate if the reviewer reconsiders a stronger score for our paper.

---

> > ### Comment · Reviewer_6o9W · 2024-08-12
> > **Thanks**
> >
> > Thanks for the rebuttal - I appreciate the corrections on some of the points. I think that given that my main issues were around general methodological confusion I probably will lean against increasing my score, but happy to consider this given discussion with other reviewers.

---

> > > ### Author Response · Authors · 2024-08-12
> > >
> > > Dear Reviewer 6o9W,
> > >
> > > We thank you for your reply. We agree that some steps of our algorithm might seem a little hard to understand. We aim to add more details on those steps according to your feedback. If you prefer any further clarification, please let us know and we would be happy to elaborate.
> > >
> > > Thank you.

---

### Decision · Program_Chairs · 2024-09-25

**Decision:**

Accept (poster)

**Comment:**

The submission provides an algorithm for sampling from a causal interventional distribution using conditional generative models, building on Shpitser and Pearl's ID algorithm.

While most reviewers consider the contribution to be sound and valuable for the field, there has been concerns that the manuscript is hard to parse. Overall, the contribution seems technically sound, but highly inaccessible to a majority of the audience, in addition to the significance of the contribution being highly challenging to assess given the complexity of the assumptions and statement of the results. The AC recommends to make an effort towards the general readership and recommends acceptance.